# On the Learning and Learnability of Quasimetrics

**Tongzhou Wang**
MIT CSAIL

**Phillip Isola**
MIT CSAIL

## Abstract

Our world is full of asymmetries. Gravity and wind can make reaching a place easier than coming back. Social artifacts such as genealogy charts and citation graphs are inherently directed. In reinforcement learning and control, optimal goal-reaching strategies are rarely reversible (symmetrical). Distance functions supported on these asymmetrical structures are called *quasimetrics*. Despite their common appearance, little research has been done on the learning of quasimetrics. Our theoretical analysis reveals that a common class of learning algorithms, including unconstrained multilayer perceptrons (MLPs), provably fails to learn a quasimetric consistent with training data. In contrast, our proposed Poisson Quasimetric Embedding (PQE) is the first quasimetric learning formulation that both is learnable with gradient-based optimization and enjoys strong performance guarantees. Experiments on random graphs, social graphs, and offline Q-learning demonstrate its effectiveness over many common baselines.

Project Page: ssnl.github.io/quasimetric.
Code: github.com/SsnL/poisson_quasimetric_embedding.

## 1 Introduction

Learned *symmetrical* metrics have been proven useful for innumerable tasks including dimensionality reduction (Tenenbaum et al., 2000), clustering (Xing et al., 2002), classification (Weinberger et al., 2006; Hoffer & Ailon, 2015), and information retrieval (Wang et al., 2014). However, the real world is largely *asymmetrical*, and *symmetrical* metrics can only capture a small fraction of it.

Generalizing metrics, *quasimetrics* (Defn. 2.1) allow for *asymmetrical* distances and can be found in a wide range of domains (see Fig. 1). Ubiquitous physical forces, such as gravity and wind, as well as human-defined rules, such as one-way roads, make the traveling time between places a quasimetric. Furthermore, many of our social artifacts are directed graphs— genealogy charts, follow-relation on Twitter (Leskovec & Krevl, 2014), citation graphs (Price, 2011), hyperlinks over the Internet, etc. Shortest paths on these graphs naturally induce quasimetric spaces. In fact, we can generalize to Markov Decision Processes (MDPs) and observe that optimal goal-reaching plan costs (i.e., universal value/Q-functions (Schaul et al., 2015; Sutton et al., 2011)) always form a quasimetric (Bertsekas & Tsitsiklis, 1991; Tian et al., 2020). Moving onto more abstract structures, quasimetrics can also be found as expected hitting times in Markov chains, and as conditional Shannon entropy $H(\cdot \mid \cdot)$ in information theory. (See the appendix for proofs and discussions of these quasimetrics.)

In this work, we study the task of *quasimetric learning*. Given a sampled training set of pairs and their quasimetric distances, we ask: how well can we learn a quasimetric that fits the training data? We define *quasimetric learning* in analogy to metric learning: whereas metric learning is the problem of learning a metric function, quasimetric learning is the problem of learning a quasimetric function. This may involve searching over a hypothesis space constrained to only include quasimetric functions (which is what our method does) or it could involve searching for approximately quasimetric functions (we compare to and analyze such approaches). Successful formulations have many potential applications, such as structural priors in reinforcement learning (Schaul et al., 2015; Tian et al., 2020), graph learning (Rizi et al., 2018) and causal relation learning (Balashankar & Subramanian, 2021).

Towards this goal, our contributions are

- We study the quasimetric learning task with two goals: (1) fitting training data well and (2) respecting quasimetric constraints (Sec. 3);

Figure 1: Examples of quasimetric spaces. The car drawing is borrowed from Sutton & Barto (2018).

- We prove that a large family of algorithms, including unconstrained networks trained in the Neural Tangent Kernel (NTK) regime (Jacot et al., 2018), fail at this task, while a learned embedding into a latent quasimetric space can potentially succeed (Sec. 4);
- We propose Poisson Quasimetric Embeddings (PQEs), the first quasimetric embedding formulation learnable with gradient-based optimization that also enjoys strong theoretical guarantees on approximating arbitrary quasimetrics (Sec. 5);
- Our experiments complement the theory and demonstrate the benefits of PQEs on random graphs, social graphs and offline Q-learning (Sec. 6).

## 2 PRELIMINARIES ON QUASIMETRICS AND POISSON PROCESSES

**Quasimetric space** is a generalization of metric space where all requirements of metrics are satisfied, except that the distances can be asymmetrical.

**Definition 2.1 (Quasimetric Space).** A *quasimetric space* is a pair $(\mathcal{X}, d)$, where $\mathcal{X}$ is a set of points and $d\colon \mathcal{X} \times \mathcal{X} \to [0, \infty]$ is the quasimetric, satisfying the following conditions:

$$\forall x, y \in \mathcal{X}, \qquad x = y \iff d(x, y) = 0, \qquad \text{(Identity of Indiscernibles)}$$
$$\forall x, y, z \in \mathcal{X}, \qquad d(x, y) + d(y, z) \geq d(x, z). \qquad \text{(Triangle Inequality)}$$

Being asymmetric, quasimetrics are often thought of as (shortest-path) distances of some (possibly infinite) weighted directed graph. A natural way to quantify the complexity of a quasimetric is to consider that of its underlying graph. *Quasimetric treewidth* is an instantiation of this idea.

**Definition 2.2 (Treewidth of Quasimetric Spaces (Mémoli et al., 2018)).** Consider a quasimetric space $M$ as shortest-path distances on a positively-weighted directed graph. *Treewidth* of $M$ is the minimum over all such graphs' treewidths.

**Poisson processes** are commonly used to model events (or points) randomly occurring across a set $A$ (Kingman, 2005), e.g., raindrops hitting a windshield, photons captured by a camera. The number of such events within a subset of $A$ is modeled as a Poisson distribution, whose mean is given by a measure $\mu$ of $A$ that determines how "frequently the events happen at each location".

**Definition 2.3 (Poisson Process).** For nonatomic measure $\mu$ on set $A$, a *Poisson process* on $A$ with *mean measure* $\mu$ is a random countable subset $P \subset A$ (i.e., the random events / points) such that

- for any disjoint measurable subsets $A_1, \ldots, A_n$ of $A$, the random variables $N(A_1), \ldots, N(A_n)$ are independent, where $N(B) \triangleq \#\{P \cap B\}$ is the number of points of $P$ in $B$, and
- $N(B)$ has the Poisson distribution with mean $\mu(B)$, denoted as $\text{Pois}(\mu(B))$.

**Fact 2.4 (Differentiability of $\mathbb{P}[N(A_1) \leq N(A_2)]$).** For two measurable subsets $A_1, A_2$,

$$\mathbb{P}[N(A_1) \leq N(A_2)] = \mathbb{P}\big[\underbrace{\text{Pois}(\mu(A_1 \setminus A_2)) \leq \text{Pois}(\mu(A_2 \setminus A_1))}_{\text{two \textit{independent} Poissons}}\big]. \tag{1}$$

Furthermore, for independent $X \sim \text{Pois}(\mu_1)$, $Y \sim \text{Pois}(\mu_2)$, the probability $\mathbb{P}[X \leq Y]$ is *differentiable w.r.t. $\mu_1$ and $\mu_2$*. In the special case where $\mu_1$ or $\mu_2$ is zero, we can simply compute

$$\mathbb{P}[X \leq Y] = \begin{cases} \mathbb{P}[0 \leq Y] = 1 & \text{if } \mu_1 = 0 \\ \mathbb{P}[X \leq 0] = \mathbb{P}[X = 0] = e^{-\mu_1} & \text{if } \mu_2 = 0 \end{cases} \qquad \text{(Pois(0) is always 0)}$$
$$= \exp\left(-(\mu_1 - \mu_2)^+\right), \tag{2}$$

where $x^+ \triangleq \max(0, x)$. For general $\mu_1, \mu_2$, this probability and its gradients can be obtained via a connection to noncentral $\chi^2$ distribution (Johnson, 1959). We derive the formulas in the appendix.

Therefore, if $A_1$ and $A_2$ are parametrized by some $\theta$ such that $\mu(A_1 \setminus A_2)$ and $\mu(A_2 \setminus A_1)$ are differentiable w.r.t. $\theta$, so is $\mathbb{P}[N(A_1) \leq N(A_2)]$.

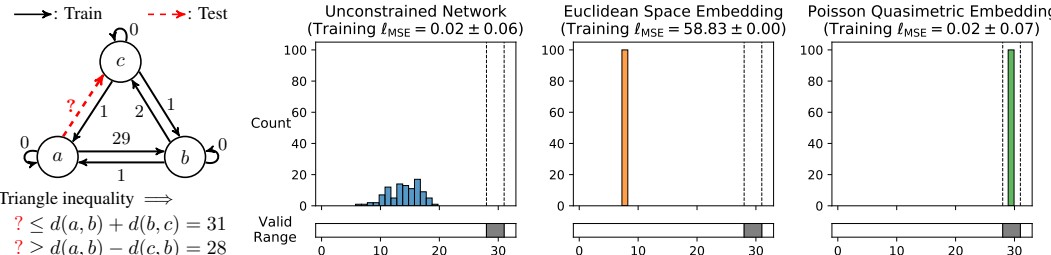

Figure 2: Quasimetric learning on a 3-element space. **Leftmost:** Training set contains all pairs except for $(a, c)$. Arrow labels show quasimetric distances (rather than edge weights). A quasimetric $\hat{d}$ should predict $\hat{d}(a, c) \in [28, 30]$. **Right three:** Different formulations are trained to fit training pairs distances, and then predict on the test pair. Plots show distribution of the prediction over 100 runs.

## 3 QUASIMETRIC LEARNING

Consider a quasimetric space $(\mathcal{X}, d)$. The *quasimetric learning* task aims to infer a quasimetric from observing a training set $\{(x_i, y_i, d(x_i, y_i))\}_i \subset \mathcal{X} \times \mathcal{X} \times [0, \infty]$. Naturally, our goals for a learned predictor $\hat{d} \colon \mathcal{X} \times \mathcal{X} \to \mathbb{R}$ are: respecting the quasimetric constraints and fitting training distances.

Crucially, we are not simply aiming for the usual sense of *generalization*, i.e., low population error. Knowing that true distances have a quasimetric structure, we can better evaluate predictors and desire ones that fit the training data and are (approximately) quasimetrics. These objectives also indirectly capture generalization because a predictor failing either requirement must have large error on some pairs, whose true distances follow quasimetric constraints. We formalize this relation in Thm. 4.3.

### 3.1 LEARNING ALGORITHMS AND HYPOTHESIS SPACES

Ideally, the learning should scale well with data, potentially generalize to unseen samples, and support integration with other deep learning systems (e.g., via differentiation).

**Relaxed hypothesis spaces.** One can simply learn a generic function approximator that maps the (concatenated) input pair to a scalar as the prediction of the pair's distance, or its transformed version (e.g., log distance). This approach has been adopted in learning graph distances (Rizi et al., 2018) and plan costs in MDPs (Tian et al., 2020). When the function approximator is a deep neural network, we refer to such methods as *unconstrained networks*. While they are known to fit training data well (Jacot et al., 2018), in this paper we also investigate whether they learn to be (approximately) quasimetrics.

**Restricted hypothesis spaces.** Alternatively, we can encode each input to a latent space $\mathcal{Z}$, where a latent quasimetric $d_z$ gives the distance prediction. This guarantees learning a quasimetric over data space $\mathcal{X}$. Often $d_z$ is restricted to a subset unable to approximate all quasimetrics, i.e., an **overly restricted hypothesis space**, such as metric embeddings and the recently proposed DeepNorm and WideNorm (Pitis et al., 2020). While our proposed Poisson Quasimetric Embedding (PQE) (specified in Sec. 5) is also a latent quasimetric, it can approximate arbitrary quasimetrics (and is differentiable). PQE thus searches in **a space that approximates all quasimetrics and only quasimetrics**.

### 3.2 A TOY EXAMPLE

To build up intuition on how various algorithms perform according to our two goals, we consider a toy quasimetric space with only 3 elements in Fig. 2. The space has a total of 9 pairs, 8 of which form the training set. Due to quasimetric requirements (esp. triangle inequality), knowing distances of these 8 pairs restricts valid values for the heldout pair to a particular range (which is $[28, 31]$ in this case). If a model approximates 8 training pairs well *and* respects quasimetric constraints well, its prediction on that heldout pair should fall into this range.

We train three models w.r.t. mean squared error (MSE) over the training set using gradient descent:
- Unconstrained deep network that predicts distance,
- Metric embedding into a latent Euclidean space with a deep encoder,
- Quasimetric embedding into a latent PQE space with a deep encoder (our method from Sec. 5).

The three approaches exhibit interesting qualitative differences. Euclidean embedding, unable to model asymmetries in training data, fails to attain a low training error. While both other methods approximate training distances well, unconstrained networks greatly violate quasimetric constraints; only PQEs respect the constraints and consistently predicts within the valid range.

Here, the structural prior of embedding into a quasimetric latent space appears important to successful learning. Without any such prior, unconstrained networks fail badly. In the next section, we present a rigorous theoretical study of the quasimetric learning task, which confirms this intuition.

## 4   THEORETICAL ANALYSIS OF VARIOUS LEARNING ALGORITHMS

In this section, we define concrete metrics for the two quasimetric learning objectives stated above, and present positive and negative theoretical findings for various learning algorithms.

**Overview.**   Our analysis focuses on data-agnostic bounds, which are often of great interests in machine learning (e.g., VC-dimension (Vapnik & Chervonenkis, 2015)). We prove a strong negative result for a general family of learning algorithms (including unconstrained MLPs trained in NTK regime, $k$-nearest neighbor, and min-norm linear regression): they can arbitrarily badly fail to fit training data or respect quasimetric constraints (Thm. 4.6). Our informative construction reveals the core reason of their failure. Quasimetric embeddings, however, enjoy nice properties as long as they can approximate arbitrary quasimetrics, which motivates searching for "universal quasimetrics". The next section presents PQEs as such universal approximators and states their theoretical guarantees.

**Assumptions.**   We consider quasimetric spaces $(\mathcal{X}, d)$ with $\mathcal{X} \subset \mathbb{R}^d$, finite size $n = |X| < \infty$, and finite distances (i.e., $d$ has range $[0, \infty)$). It allows discussing deep networks which can't handle infinities well. This mild assumption can be satisfied by simply capping max distances in quasimetrics. For training, $m < n^2$ pairs are uniformly sampled as training pairs $S \subset \mathcal{X} \times \mathcal{X}$ without replacement.

In the appendix, we provide all full proofs, further discussions of our assumptions and presented results, as well as additional results concerning specific learning algorithms and settings.

### 4.1   DISTORTION AND VIOLATION METRICS FOR QUASIMETRIC LEARNING

We use *distortion* as a measure of how well the distance is preserved, as is standard in embedding analyses (e.g., Bourgain (1985)). In this work, we especially consider *distortion over a subset of pairs*, to quantify how well a predictor $\hat{d}$ approximates distances over the training subset $S$.

**Definition 4.1 (Distortion).** Distortion of $\hat{d}$ over a subset of pairs $S \subset \mathcal{X} \times \mathcal{X}$ is $\mathsf{dis}_S(\hat{d}) \triangleq \left( \max_{(x,y) \in S, x \neq y} \frac{\hat{d}(x,y)}{d(x,y)} \right) \left( \max_{(x,y) \in S, x \neq y} \frac{d(x,y)}{\hat{d}(x,y)} \right)$, and its overall distortion is $\mathsf{dis}(\hat{d}) \triangleq \mathsf{dis}_{\mathcal{X} \times \mathcal{X}}(\hat{d})$.

For measuring consistency w.r.t. quasimetric constraints, we define the *(quasimetric) violation* metric. Violation focuses on *triangle inequality*, which can often be more complex (e.g., in Fig. 2), compared to the relatively simple *non-negativity* and *Identity of Indiscernibles*.

**Definition 4.2 (Quasimetric Violation).** *Quasimetric violation* (*violation* for short) of $\hat{d}$ is $\mathsf{vio}(\hat{d}) \triangleq \max_{A_1, A_2, A_3 \in \mathcal{X}} \frac{\hat{d}(A_1, A_3)}{\hat{d}(A_1, A_2) + \hat{d}(A_2, A_3)}$, where we define $\frac{0}{0} = 1$ for notation simplicity.

Both distortion and violation are nicely agnostic to scaling. Furthermore, assuming *non-negativity* and *Identity of Indiscernibles*, $\mathsf{vio}(\hat{d}) \geq 1$ always, with equality iff $\hat{d}$ is a quasimetric.

Distortion and violation also capture generalization. Because the true distance $d$ has optimal training distortion (on $S$) and violation, a predictor $\hat{d}$ that does badly on either must also be far from truth.

**Theorem 4.3 (Distortion and Violation Lower-Bound Generalization Error).** For non-negative $\hat{d}$, $\mathsf{dis}(\hat{d}) \geq \max(\mathsf{dis}_S(\hat{d}), \sqrt{\mathsf{vio}(\hat{d})})$, where $\mathsf{dis}(\hat{d})$ captures generalization over the entire $\mathcal{X}$ space.

### 4.2   LEARNING ALGORITHMS EQUIVARIANT TO ORTHOGONAL TRANSFORMS

For quasimetric space $(\mathcal{X}, d)$, $\mathcal{X} \subset \mathbb{R}^d$, we consider applying general learning algorithms by concatenating pairs to form inputs $\in \mathbb{R}^{2d}$ (e.g., unconstrained networks). While straightforward, this approach means that the algorithms are generally *unable to relate the same element appearing as 1st or 2nd input*. As we will show, this is sufficient for a wide family of learning algorithms to fail badly– ones equivariant to orthogonal transforms, which we refer to as OrEq algorithms (Defn. 4.4).

For an OrEq algorithm, training on orthogonally transformed data does not affect its prediction, as long as test data is identically transformed. Many standard learning algorithms are OrEq (Lemma 4.5).

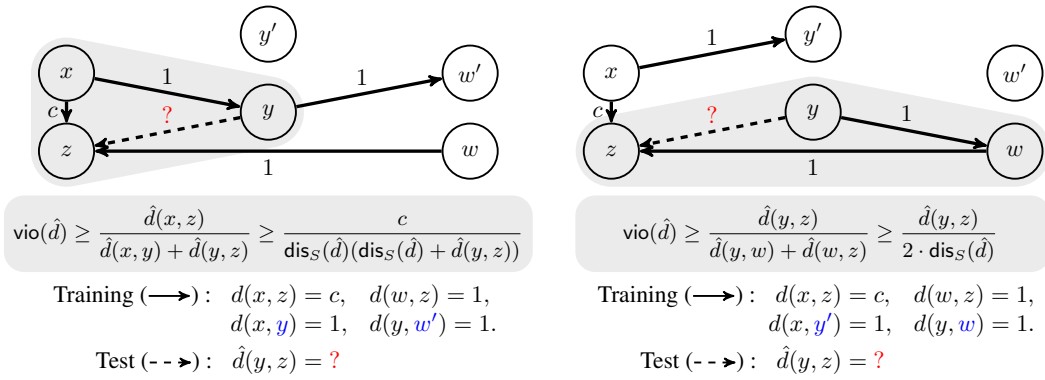

$$\mathsf{vio}(\hat{d}) \geq \frac{\hat{d}(x,z)}{\hat{d}(x,y) + \hat{d}(y,z)} \geq \frac{c}{\mathsf{dis}_S(\hat{d})(\mathsf{dis}_S(\hat{d}) + \hat{d}(y,z))}$$

Training ($\longrightarrow$): $\quad d(x,z) = c, \quad d(w,z) = 1,$
$\qquad\qquad\qquad\quad d(x,y) = 1, \quad d(y,w') = 1.$

Test ($\dashrightarrow$): $\quad \hat{d}(y,z) = ?$

$$\mathsf{vio}(\hat{d}) \geq \frac{\hat{d}(y,z)}{\hat{d}(y,w) + \hat{d}(w,z)} \geq \frac{\hat{d}(y,z)}{2 \cdot \mathsf{dis}_S(\hat{d})}$$

Training ($\longrightarrow$): $\quad d(x,z) = c, \quad d(w,z) = 1,$
$\qquad\qquad\qquad\quad d(x,y') = 1, \quad d(y,w) = 1.$

Test ($\dashrightarrow$): $\quad \hat{d}(y,z) = ?$

Figure 3: Two training sets pose incompatible constraints (⬤) for the test pair distance $d(y,z)$. With one-hot features, an orthogonal transform can exchange $(*,y) \leftrightarrow (*,y')$ and $(*,w) \leftrightarrow (*,w')$, leaving the test pair $(y,z)$ unchanged, but transforming the training pairs from one scenario to the other. Given either training set, an OrEq algorithm must attain same training distortion and predict identically on $(y,z)$. For appropriate $c$, this implies large distortion or violation in one of these cases.

**Definition 4.4 (Equivariant Learning Algorithms).** Given training set $\mathcal{D} = \{(z_i, y_i)\}_i \subset \mathcal{Z} \times \mathcal{Y}$, where $z_i$ are inputs and $y_i$ are targets, a learning algorithm Alg produces a function $\mathsf{Alg}(\mathcal{D})\colon \mathcal{Z} \to Y$ such that $\mathsf{Alg}(\mathcal{D})(z')$ is the function's prediction on sample $z'$. Consider $\mathcal{T}$ a set of transformations $\mathcal{Z} \to \mathcal{Z}$. Alg is equivariant to $\mathcal{T}$ iff for all transform $T \in \mathcal{T}$, training set $\mathcal{D}$, $\mathsf{Alg}(\mathcal{D}) = \mathsf{Alg}(T\mathcal{D}) \circ T$, where $T\mathcal{D} = \{(Tz, y)\colon (z,y) \in \mathcal{D}\}$ is the training set with transformed inputs.

**Lemma 4.5 (Examples of OrEq Algorithms).** $k$-nearest-neighbor with Euclidean distance, MLP trained with squared loss in NTK regime, and min-norm least-squares linear regression are OrEq.

**Failure case.** The algorithms treats the concatenated inputs as generic vectors. If a transform fundamentally changes the quasimetric structure but is not fully reflected in the learned function (e.g., due to equivariance), learning must fail. The two training sets in Fig. 3 are sampled from two different quasimetrics over the same 6 elements An orthogonal transform links both training sets *without affecting the test pair*, which is constrained differently in two quasimetrics. An OrEq algorithm, necessarily predicting the test pair identically seeing either training set, must thus fail on one. In the appendix, we empirically verify that unconstrained MLPs indeed *do fail* on this construction.

Extending to larger quasimetric spaces, we consider graphs containing many copies of *both* patterns in Fig. 3. With high probability, our sampled training set fails in the same way—the learning algorithm can not distinguish it from another training set with different quasimetric constraints.

**Theorem 4.6 (Failure of OrEq Algorithms).** Let $(f_n)_n$ be an *arbitrary* sequence of large values. There is an infinite sequence of quasimetric spaces $((\mathcal{X}_n, d_n))_n$ with $|\mathcal{X}_n| = n$, $\mathcal{X}_n \subset \mathbb{R}^n$ such that, over the random training set $S$ of size $m$, any OrEq algorithm must output a predictor $\hat{d}$ that satisfies

- $\hat{d}$ fails *non-negativity*, or
- $\max(\mathsf{dis}_S(\hat{d}), \mathsf{vio}(\hat{d})) \geq f_n$ (i.e., $\hat{d}$ approximates training $S$ badly or is far from a quasimetric),

with probability $1/2 - o(1)$, as long as $S$ does not contain almost all pairs $1 - m/n^2 = \omega(n^{-1/3})$, and does not only include few pairs $m/n^2 = \omega(n^{-1/2})$.

Furthermore, standard NTK results show that unconstrained MLPs trained in NTK regime converge to a function with zero training loss. By the above theorem, the limiting function is not a quasimetric with nontrivial probability. In the appendix, we formally state this result. Despite their empirical usages, these results suggest that unconstrained networks are likely not suited for quasimetric learning.

## 4.3 Quasimetric Embeddings

A quasimetric embedding consists of a mapping $f$ from data space $\mathcal{X}$ to a latent quasimetric space $(\mathcal{Z}, d_z)$, and predicts $\hat{d}(x,y) \triangleq d_z(f(x), f(y))$. Therefore, they always respect all quasimetric constraints and attain optimal violation of value 1, *regardless of training data*.

However, unlike deep networks, their distortion (approximation) properties depend on the specific latent quasimetrics. If the latent quasimetric can generally approximate *any* quasimetric (with flexible learned encoders such as deep networks), we have nice guarantees for both distortion and violation.

In the section below, we present Poisson Quasimetric Embedding (PQE) as such a latent quasimetric, along with its theoretical distortion and violation guarantees.

## 5 POISSON QUASIMETRIC EMBEDDINGS (PQEs)

Motivated by above theoretical findings, we aim to find a latent quasimetric space $(\mathbb{R}^d, d_z)$ with a deep network encoder $f \colon \mathcal{X} \to \mathbb{R}^d$, and a quasimetric $d_z$ that is both *universal* and *differentiable*:

- for any data quasimetric $(\mathcal{X}, d)$, there exists an encoder $f$ such that $d_z(f(x), f(y)) \approx d(x, y)$;
- $d_z$ is differentiable (for optimizing $f$ and possible integration with other gradient-based systems).

**Notation 5.1.** We use $x, y$ for elements of the data space $\mathcal{X}$, $u, v$ for elements of the latent space $\mathbb{R}^d$, upper-case letters for random variables, and $(\cdot)_z$ for indicating functions in latent space (e.g., $d_z$).

An existing line of machine learning research learns *quasipartitions*, or *partial orders*, via Order Embeddings (Vendrov et al., 2015). Quasipartitions are in fact special cases of quasimetrics whose distances are restricted to be binary, denoted as $\pi$. An Order Embedding is a representation of a quasipartition, where $\pi^{\mathsf{OE}}(x, y) = 0$ (i.e., $x$ is related to $y$) iff $f(x) \leq f(y)$ coordinate-wise:

$$\pi^{\mathsf{OE}}(x, y) \triangleq \pi_z^{\mathsf{OE}}(f(x), f(y)) \triangleq 1 - \prod_j \mathbf{1}_{f(x)_j - f(y)_j \leq 0}. \tag{3}$$

Order Embedding is *universal* and can model *any quasipartition* (see appendix and Hiraguchi (1951)).

Can we extend this discrete idea to general continuous quasimetrics? Quite naïvely, one may attempt a straightforward soft modification of Order Embedding:

$$\pi_z^{\mathsf{SoftOE}}(u, v) \triangleq 1 - \prod_j \exp\left(-(u_j - v_j)^+\right) = 1 - \exp\left(-\sum_j (u_j - v_j)^+\right), \tag{4}$$

which equals $0$ if $u \leq v$ coordinate-wise, and increases to $1$ as some coordinates violate this condition more. However, it is unclear whether this gives a quasimetric.

A more principled way is to parametrize a (scaled) *distribution of latent quasipartitions* $\Pi_z$, whose expectation naturally gives a continuous-valued quasimetric:

$$d_z(u, v; \Pi_z, \alpha) \triangleq \alpha \cdot \mathbb{E}_{\pi_z \sim \Pi_z}\left[\pi_z(u, v)\right], \qquad \alpha \geq 0. \tag{5}$$

Poisson Quasimetric Embedding (PQE) gives a general recipe for constructing such $\Pi_z$ distributions so that $d_z$ is *universal* and *differentiable*. Within this framework, we will see that $\pi_z^{\mathsf{SoftOE}}$ is actually a quasimetric based on such a distribution and is (almost) sufficient for our needs.

### 5.1 DISTRIBUTIONS OF LATENT QUASIPARTITIONS

A random latent quasipartition $\pi_z \colon \mathbb{R}^d \times \mathbb{R}^d \to \{0, 1\}$ is a difficult object to model, due to complicated quasipartition constraints. Fortunately, the Order Embedding representation (Eq. (3)) is without such constraints. If, instead of fixed latents $u, v$, we have *random latents* $R(u), R(v)$, we can compute:

$$\mathbb{E}_{\pi_z}[\pi_z(u, v)] = \mathbb{E}_{R(u), R(v)}\left[\pi_z^{\mathsf{OE}}(R(u), R(v))\right] = 1 - \mathbb{P}\left[R(u) \leq R(v) \text{ coordinate-wise}\right]. \tag{6}$$

In this view, we represent a random $\pi_z$ via a joint distribution of random vectors[1] $\{R(u)\}_{u \in \mathbb{R}^d}$, i.e., a *stochastic process*. To easily compute the probability of this coordinate-wise event, we assume that each dimension of random vectors is from an independent process, and obtain

$$\mathbb{E}_{\pi_z}\left[\pi_z(u, v)\right] = 1 - \prod_j \mathbb{P}\left[R_j(u) \leq R_j(v)\right]. \tag{7}$$

The choice of stochastic process is flexible. Using *Poisson processes* (with Lebesgue mean measure; Defn. 2.3) that count random points on half-lines[2] $(-\infty, a]$, we can have $R_j(u) = N_j((\infty, u_j])$, the (random) count of events in $(\infty, u_j]$ from $j$-th Poisson process:

$$\mathbb{E}_{\pi_z \sim \Pi_z}\left[\pi_z(u, v)\right] = 1 - \prod_j \mathbb{P}\left[N_j((-\infty, u_j]) \leq N_j((-\infty, v_j])\right] \tag{8}$$

$$= 1 - \prod_j \exp\left(-(u_j - v_j)^+\right) = \pi_z^{\mathsf{SoftOE}}(u, v), \tag{9}$$

---

[1] In general, these random vectors $R(u)$ do not have to be of the same dimension as $u \in \mathbb{R}^d$, although the dimensions do match in the PQE variants we experiment with.

[2] Half-lines has Lebesgue measure $\infty$. More rigorously, consider using a small value as the lower bounds of these intervals, which leads to same result.

where we used Fact 2.4 and the observation that one half-line is either subset or superset of another. Indeed, $\pi_z^{\mathsf{SoftOE}}$ is an expected quasipartition (and thus a quasimetric), and is *differentiable*.

Considering a mixture of such distributions for expressiveness, the full latent quasimetric formula is

$$d_z^{\mathsf{PQE\text{-}LH}}(u, v; \alpha) \triangleq \sum_i \alpha_i \cdot \Big( 1 - \exp\big( - \sum_j (u_{i,j} - v_{i,j})^+ \big) \Big), \tag{10}$$

where we slightly abuse notation and consider latents $u$ and $v$ as (reshaped to) 2-dimensional. We will see that this is a special PQE case with **L**ebesgue measure and **h**alf-lines, and thus denoted PQE-LH.

## 5.2 GENERAL PQE FORMULATION

We can easily generalize the above idea to independent Poisson processes of general mean measures $\mu_j$ and (sub)set parametrizations $u \to A_j(u)$, and obtain an expected quasipartition as:

$$\mathbb{E}_{\pi_z \sim \Pi_z^{\mathsf{PQE}}(\mu, A)}[\pi_z(u, v)] \triangleq 1 - \prod_j \mathbb{P}\left[ N_j(A_j(u)) \le N_j(A_j(v)) \right] \tag{11}$$

$$= 1 - \prod_j \mathbb{P}\Big[ \mathrm{Pois}(\underbrace{\mu_j(A_j(u) \setminus A_j(v))}_{\text{Poisson rate of points landing only in } A_j(u)}) \le \mathrm{Pois}(\mu_j(A_j(v) \setminus A_j(u))) \Big], \tag{12}$$

which is *differentiable* as long as the measures and set parametrizations are (after set differences). Similarly, considering a mixture gives us an expressive latent quasimetric.

*A general PQE latent quasimetric* is defined with $\{(\mu_{i,j}, A_{i,j})\}_{i,j}$ and weights $\alpha_i \ge 0$ as:

$$d_z^{\mathsf{PQE}}(u, v; \mu, A, \alpha) \triangleq \sum_i \alpha_i \cdot \mathbb{E}_{\pi_z \sim \Pi_z^{\mathsf{PQE}}(\mu_i, A_i)}[\pi_z(u, v)] \tag{13}$$

$$= \sum_i \alpha_i \Big( 1 - \prod_j \mathbb{P}\Big[ \mathrm{Pois}(\mu_{i,j}(A_{i,j}(u) \setminus A_{i,j}(v))) \le \mathrm{Pois}(\mu_{i,j}(A_{i,j}(v) \setminus A_{i,j}(u))) \Big] \Big),$$

whose optimizable parameters include $\{\alpha_i\}_i$, possible ones from $\{(\mu_{i,j}, A_{i,j})\}_{i,j}$ (and encoder $f$).

This general recipe can be instantiated in many ways. Setting $A_{i,j}(u) \to (-\infty, u_{i,j}]$ and Lebesgue $\mu_{i,j}$, recovers PQE-LH. In the appendix, we consider a form with **G**aussian-based measures and **G**aussian-shapes, denoted as PQE-GG. Unlike PQE-LH, PQE-GG always gives nonzero gradients.

The appendix also includes several implementation techniques that empirically improve stability, including learning $\alpha_i$'s with deep linear networks, a formulation that outputs discounted distance, etc.

## 5.3 CONTINUOUS-VALUED STOCHASTIC PROCESSES

But why Poisson processes over more common choices such as Gaussian processes? It turns out that common continuous-value processes fail to give a *differentiable* formula.

Consider a non-degenerate process $\{R(u)\}_u$, where $(R(u), R(v))$ has bounded density if $u \ne v$. Perturbing $u \to u + \delta$ leaves $\mathbb{P}[R(u) = R(u+\delta)] = 0$. Then one of $\mathbb{P}[R(u) \le R(u+\delta)]$ and $\mathbb{P}[R(u+\delta) \le R(u)]$ must be far away from 1 (as they sum to 1), breaking differentiability at $\mathbb{P}[R(u) \le R(u)] = 1$. (This argument is formalized in the appendix.) Discrete-valued processes, however, can leave most probability mass on $R(u) = R(u+\delta)$ and thus remain differentiable.

## 5.4 THEORETICAL GUARANTEES

Our PQEs bear similarity with the algorithmic quasimetric embedding construction in Mémoli et al. (2018). Extending their analysis to PQEs, we obtain the following distortion and violation guarantees.

**Theorem 5.2 (Distortion and violation of PQEs).** Under the assumptions of Sec. 4, *any* quasimetric space with size $n$ and treewidth $t$ admits a PQE-LH and a PQE-GG with distortion $\mathcal{O}(t \log^2 n)$ and violation 1, with an expressive encoder (e.g., a ReLU network with $\ge 3$ layers and polynomial width).

In fact, these guarantees apply to any PQE formulation that satisfies a mild condition. Informally, any PQE with $h \times k$ Poisson processes (i.e., $h$ mixtures) enjoys the above guarantees if it can approximate the discrete counterpart: mixtures of $h$ Order Embeddings, each specified with $k$ dimensions. In the appendix, we make this condition precise and provide a full proof of the above theorem.

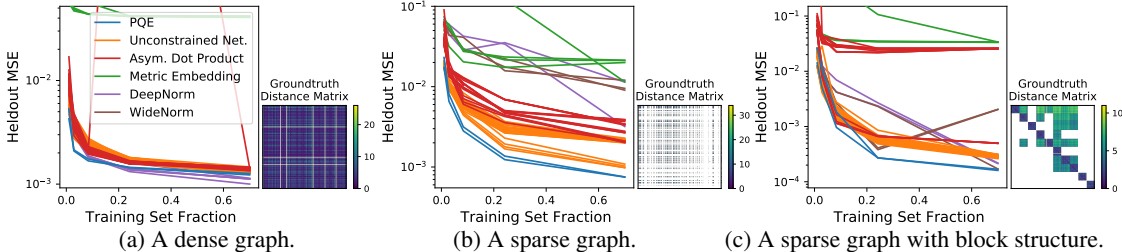

Figure 4: Comparison of PQE and baselines on quasimetric learning in random directed graphs.

## 6 EXPERIMENTS

Our experiments are designed to (1) confirm our theoretical findings and (2) compare PQEs against a wider range of baselines, across different types of tasks. In all experiments, we optimize $\gamma$-discounted distances (with $\gamma \in \{0.9, 0.95\}$), and compare the following five families of methods:

- **PQEs (2 formulations):** PQE-LH and PQE-GG with techniques mentioned in Sec. 5.2.
- **Unconstrained networks (20 formulations):** Predict raw distance (directly, with $\exp$ transform, and with $(\cdot)^2$ transform) or $\gamma$-discounted distance (directly, and with a sigmoid-transform). Each variant is run with a possible triangle inequality regularizer $\mathbb{E}_{x,y,z}\big[\max(0, \gamma^{\hat{d}(x,y)+\hat{d}(y,z)} - \gamma^{\hat{d}(x,z)})^2\big]$ for each of 4 weights $\in \{0, 0.3, 1, 3\}$.
- **Asymmetrical dot products (20 formulations):** On input pair $(x, y)$, encode each into a feature vector with a *different* network, and take the dot product. Identical to unconstrained networks, the output is used in the same 5 ways, with the same 4 triangle inequality regularizer options.
- **Metric encoders (4 formulations):** Embed into Euclidean space, $\ell_1$ space, hypersphere with (scaled) spherical distance, or a mixture of all three.
- **DeepNorm (2 formulations) and WideNorm (3 formulations):** Quasimetric embedding methods that often require significantly more parameters than PQEs (often on the order of $10^6 \sim 10^7$ more effective parameters; see the appendix for detailed comparisons) but can only approximate a subset of all possible quasimetrics (Pitis et al., 2020).

We show average results from 5 runs. The appendix provides experimental details, full results (including standard deviations), additional experiments, and ablation studies.

**Random directed graphs.** We start with randomly generated directed graphs of 300 nodes, with 64-dimensional node features given by randomly initialized neural networks. After training with MSE on discounted distances, we test the models' prediction error on the unseen pairs (i.e., generalization), measured also by MSE on discounted distances. On three graphs with distinct structures, PQEs significantly outperform baselines across almost all training set sizes (see Fig. 4). Notably, while DeepNorm and WideNorm do well on the dense graph quasimetric, they struggle on the other two, attaining both high test MSE (Fig. 4) and train MSE (not shown). This is consistent with the fact that they can only approximate a subset of all quasimetrics, while PQEs can approximate all quasimetrics.

**Large-scale social graph.** We choose the Berkeley-Stanford Web Graph (Leskovec & Krevl, 2014) as the real-wold social graph for evaluation. This graph consists of 685,230 pages as nodes, and 7,600,595 hyperlinks as directed edges. We use 128-dimensional node2vec features (Grover & Leskovec, 2016) and the landmark method (Rizi et al., 2018) to construct a training set of 2,500,000 pairs, and a test set of 150,000 pairs. PQEs generally perform better than other methods, accurately predicting finite distances while predicting high values for infinite distances (see Table 1). DeepNorms and WideNorms learn finite distances less accurately here, and also do much worse than PQEs on learning the (quasi)metric of an *undirected* social graph (shown in the appendix).

**Offline Q-learning.** Optimal goal-reaching plan costs in MDPs are quasimetrics (Bertsekas & Tsitsiklis, 1991; Tian et al., 2020) (see also the appendix). In practice, optimizing deep Q-functions often suffers from stability and sample efficiency issues (Henderson et al., 2018; Fujimoto et al., 2018). As a proof of concept, we use PQEs as goal-conditional Q-functions in offline Q-learning, on the grid-world environment with one-way doors built upon `gym-minigrid` (Chevalier-Boisvert et al., 2018) (see Fig. 1 right), following the algorithm and data sampling procedure described in Tian et al. (2020). Adding strong quasimetric structures greatly improves sample efficiency and greedy planning success rates over popular existing approaches such as unconstrained networks used in Tian et al. (2020) and asymmetrical dot products used in Schaul et al. (2015) (see Fig. 5). As an interesting observation, some metric embedding formulations work comparably well.

| | Triangle inequality regularizer | MSE w.r.t. $\gamma$-discounted distances ($\times 10^{-3}$) $\downarrow$ | L1 Error when true $d < \infty$ $\downarrow$ | Prediction $\hat{d}$ when true $d = \infty$ $\uparrow$ |
|---|---|---|---|---|
| PQE-LH | ✗ | 3.043 | 1.626 | 69.942 |
| PQE-GG | ✗ | 3.909 | 1.895 | 101.824 |
| Best Unconstrained Net. | ✗ | 3.086 | 2.115 | 59.524 |
| | ✓ | 2.813 | 2.211 | 61.371 |
| Best Asym. Dot Product | ✗ | 48.106 | $2.520 \times 10^{11}$ | $2.679 \times 10^{11}$ |
| | ✓ | 48.102 | $2.299 \times 10^{11}$ | $2.500 \times 10^{11}$ |
| Best Metric Embedding | ✗ | 17.595 | 7.540 | 53.850 |
| Best DeepNorm | ✗ | 5.071 | 2.085 | 120.045 |
| Best WideNorm | ✗ | 3.533 | 1.769 | 124.658 |

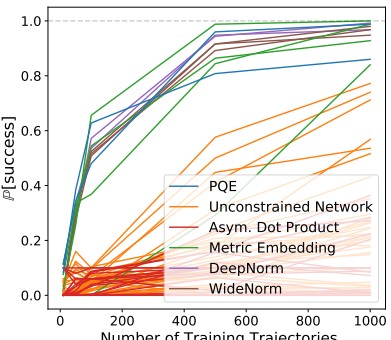

Table 1: Quasimetric learning on large-scale web graph. "Best" is selected by *test* MSE w.r.t. $\gamma$-discounted distances.

Figure 5: Offline Q-learning results.

## 7 RELATED WORK

**Metric learning.** Metric learning aims to approximate a target metric/similarity function, often via a learned embedding into a metric space. This idea has successful applications in dimensionality reduction (Tenenbaum et al., 2000), information retrieval (Wang et al., 2014), clustering (Xing et al., 2002), classification (Weinberger et al., 2006; Hoffer & Ailon, 2015), etc. While asymmetrical formulations have been explored, they either ignore quasimetric constraints (Oord et al., 2018; Logeswaran & Lee, 2018; Schaul et al., 2015), or are not general enough to approximate arbitrary quasimetric (Balashankar & Subramanian, 2021), which is the focus of the present paper.

**Isometric embeddings.** Isometric (distance-preserving) embeddings is a highly influential and well-studied topic in mathematics and statistics. Fundamental results, such as Bourgain's random embedding theorem (Bourgain, 1985), laid important ground work in understanding and constructing (approximately) isometric embeddings. While most such researches concern metric spaces, Mémoli et al. (2018) study an algorithmic construction of a quasimetric embedding via basic blocks called *quasipartitions*. Their approach requires knowledge of quasimetric distances between all pairs and thus is not suitable for learning. Our formulation takes inspiration from the form of their embedding, but is fully learnable with gradient-based optimization over a training subset.

**Quasimetrics and partial orders.** Partial orders (quasipartitions) are special cases of quasimetrics (see Sec. 5). A line of machine learning research studies embedding partial order structures into latent spaces for tasks such as relation discovery and information retrieval (Vendrov et al., 2015; Suzuki et al., 2019; Hata et al., 2020; Ganea et al., 2018). Unfortunately, unlike PQEs, such formulations do not straightforwardly generalize to arbitrary quasimetrics, which are more than binary relations. Similar to PQEs, DeepNorm and WideNorm are quasimetric embedding approaches learnable with gradient-based optimization (Pitis et al., 2020). Theoreically, they universally approximates a subset of quasimetrics (ones induced by asymmetrical norms). Despite often using many more parameters, they are restricted to this subset and unable to approximate general quasimetrics like PQEs do (Fig. 4).

## 8 IMPLICATIONS

In this work, we study quasimetric learning via both theoretical analysis and empirical evaluations.

Theoretically, we show strong negative results for a common family of learning algorithms, and positive guarantees for our proposed Poisson Quasimetric Embedding (PQE). Our results introduce the novel concept of equivariant learning algorithms, which may potentially be used for other learnability analyses with algorithms such as deep neural networks. Additionally, a thorough average-case or data-dependent analysis would nicely complement our results, and may shed light on conditions where algorithms like deep networks can learn decent approximations to quasimetrics in practice.

PQEs are the first quasimetric embedding formulation that can be learned via gradient-based optimization. Empirically, PQEs show promising performance in various tasks. Furthermore, PQEs are fully differentiable, and (implicitly) enforce a quasimetric structure in any latent space. They are particularly suited for integration in large deep learning systems, as we explore in the Q-learning experiments. This can potentially open the gate to many practical applications such as better embedding for planning with MDPs, efficient shortest path finding via learned quasimetric heuristics, representation learning with quasimetric similarities, causal relation learning, etc.

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

APPENDIX

## A  DISCUSSIONS FOR SEC. 2: PRELIMINARIES ON QUASIMETRICS AND POISSON PROCESSES

### A.1  QUASIMETRIC SPACES

**Definition 2.1 (Quasimetric Space).** A *quasimetric space* is a pair $(\mathcal{X}, d)$, where $\mathcal{X}$ is a set of points and $d\colon \mathcal{X} \times \mathcal{X} \to [0, \infty]$ is the quasimetric, satisfying the following conditions:

$$\forall x, y \in \mathcal{X}, \qquad x = y \iff d(x, y) = 0, \qquad \text{(Identity of Indiscernibles)}$$
$$\forall x, y, z \in \mathcal{X}, \qquad d(x, y) + d(y, z) \geq d(x, z). \qquad \text{(Triangle Inequality)}$$

**Definition A.1 (Quasipseudometric Space).** As a further generalization, we say $(\mathcal{X}, d)$ is a *quasipseudometric space* if the *Identity of Indiscernibles* requirement is only satisfied in one direction:

$$\forall x, y \in \mathcal{X}, \qquad x = y \implies d(x, y) = 0, \qquad \text{(Identity of Indiscernibles)}$$
$$\forall x, y, z \in \mathcal{X}, \qquad d(x, y) + d(y, z) \geq d(x, z). \qquad \text{(Triangle Inequality)}$$

#### A.1.1  EXAMPLES OF QUASIMETRIC SPACES

**Proposition A.2 (Expected Hitting Time of a Markov Chain).** Let random variables $(X_t)_t$ be a Markov Chain with support $\mathcal{X}$. Then $(\mathcal{X}, d_{\mathsf{hitting}})$ is a quasimetric space, where

$$d_{\mathsf{hitting}}(s, t) \triangleq \mathbb{E}\left[\text{time to hit } t \mid \text{start from } s\right], \tag{14}$$

where we define the hitting time of $s$ starting from $s$ to be $0$.

*Proof of Proposition A.2.* Obviously $d_{\mathsf{hitting}}$ is non-negative. We then verify the following quasimetric space properties:

- **Identity of Indiscernibles.** By definition, we have, $\forall x, y \in \mathcal{X}, x \neq y$,

$$d_{\mathsf{hitting}}(x, x) = 0 \tag{15}$$
$$d_{\mathsf{hitting}}(x, y) \geq 1. \tag{16}$$

- **Triangle Inequality.** For any $x, y, z \in \mathcal{X}$, we have

$$d_{\mathsf{hitting}}(x, y) + d_{\mathsf{hitting}}(y, z) = \mathbb{E}\left[\text{time to hit } y \text{ then hit } z \mid \text{start from } x\right] \tag{17}$$
$$\geq \mathbb{E}\left[\text{time to hit } z \mid \text{start from } x\right] \tag{18}$$
$$= d_{\mathsf{hitting}}(x, z). \tag{19}$$

Hence, $(\mathcal{X}, d_{\mathsf{hitting}})$ is a quasimetric space. $\qquad\square$

**Proposition A.3 (Conditional Shannon Entropy).** Let $\mathcal{X}$ be the set of random variables (of some probability space). Then $(\mathcal{X}, d_H)$ is a quasipseudometric space, where

$$d_H(X, Y) \triangleq H(Y \mid X). \tag{20}$$

If for all distinct $(X, Y) \in \mathcal{X} \times \mathcal{X}$, $X$ can not be written as (almost surely) a deterministic function of $Y$, then $(\mathcal{X}, d_H)$ is a quasimetric space.

*Proof of Proposition A.3.* Obviously $d_H$ is non-negative. We then verify the following quasipseudometric space properties:

- **Identity of Indiscernibles.** By definition, we have, $\forall X, Y \in \mathcal{X}$,

$$d_H(X, X) = H(X \mid X) = 0 \tag{21}$$
$$d_H(Y, X) = H(Y \mid X) \geq 0, \tag{22}$$

  where $\leq$ is $=$ iff $Y$ is a (almost surely) deterministic function of $X$.

- **Triangle Inequality.** For any $X, Y, Z \in \mathcal{X}$, we have

$$d_H(X, Y) + d_H(Y, Z) = H(Y \mid X) + H(Z \mid Y) \tag{23}$$
$$\geq H(Y \mid X) + H(Z \mid XY) \tag{24}$$
$$= H(YZ \mid X) \tag{25}$$
$$\geq H(Z \mid X) \tag{26}$$
$$= d_H(X, Z). \tag{27}$$

Hence, $(\mathcal{X}, d_H)$ is a quasipseudometric space, and a quasimetric space when the last condition is satisfied. $\quad\square$

**Conditional Kolmogorov Complexity** From algorithmic information theory, the conditional Kolmogorov complexity $K(y \mid x)$ also similarly measures "the bits needed to create $y$ given $x$ as input" (Kolmogorov, 1963). It is also almost a quasimetric, but the exact definition affects some constant/log terms that may make the quasimetric constraints non-exact. For instance, when defined with the prefix-free version, conditional Kolmogorov complexity is always strictly positive, even for $K(x \mid x) > 0$ (Li et al., 2008). One may remedy this with a definition using a universal Turing machine (UTM) that simply outputs the input on empty program. But to make triangle inequality work, one needs to reason about how the input and output parts work on the tape(s) of the UTM. Nonetheless, regardless of the definition details, conditional Kolmogorov complexity do satisfy a triangle inequality up to log terms (Grunwald & Vitányi, 2004). So intuitively, it behaves roughly like a quasimetric defined on the space of binary strings.

**Optimal Goal-Reaching Plan Costs in Markov Decision Processes (MDPs)** We define MDPs in the standard manner: $\mathcal{M} = (\mathcal{S}, \mathcal{A}, \mathcal{R}, \mathcal{P}, \gamma)$ (Puterman, 1994), where $\mathcal{S}$ is the state space, $\mathcal{A}$ is the action space, $\mathcal{R} \colon \mathcal{S} \times \mathcal{A} \to \mathbb{R}$ is the reward function, $\mathcal{P} \colon \mathcal{S} \times \mathcal{A} \to \Delta(\mathcal{S})$ is the transition function (where $\Delta(\mathcal{S})$ is the set of all distributions over $\mathcal{S}$), and $\gamma \in (0, 1)$ is the discount factor.

We define $\Pi$ as the collection of all stationary policies $\pi \colon \mathcal{S} \to \Delta(\mathcal{A})$ on $\mathcal{M}$. For a particular policy $\pi \in \Pi$, it induces random *trajectories*:

- *Trajectory* starting from state $s \in \mathcal{S}$ is the random variable
$$\xi_\pi(s) = (s_1, a_1, r_1, s_2, a_2, r_2, \dots), \tag{28}$$
distributed as
$$s_1 = s \tag{29}$$
$$a_i \sim \pi(s_i), \qquad \forall i \geq 1 \tag{30}$$
$$s_{i+1} \sim \mathcal{P}(s_i, a_i), \qquad \forall i \geq 1. \tag{31}$$

- *Trajectory* starting from state-action pair $(s, a) \in \mathcal{S} \times \mathcal{A}$ is the random variable
$$\xi_\pi(s, a) = (s_1, a_1, r_1, s_2, a_2, r_2, \dots), \tag{32}$$
distributed as
$$s_1 = s \tag{33}$$
$$a_1 = a \tag{34}$$
$$a_i \sim \pi(s_i), \qquad \forall i \geq 2 \tag{35}$$
$$s_{i+1} \sim \mathcal{P}(s_i, a_i), \qquad \forall i \geq 1. \tag{36}$$

**Proposition A.4 (Optimal Goal-Reaching Plan Costs in MDPs).** Consider an MDP $\mathcal{M} = (\mathcal{S}, \mathcal{A}, \mathcal{R}, \mathcal{P}, \gamma)$. WLOG, assume that $\mathcal{R} \colon \mathcal{S} \times \mathcal{A} \to (-\infty, 0]$ has only non-positive rewards (i.e., negated costs). Let $\mathcal{X} = \mathcal{S} \cup (\mathcal{S} \times \mathcal{A})$. Then $(\mathcal{X}, d_{\mathsf{sum}})$ and $(\mathcal{X}, d_\gamma)$ are quasipseudometric spaces, where

$$d_{\mathsf{sum}}(x, y) \triangleq \min_{\pi \in \Pi} \mathbb{E} \left[ \text{total costs from } x \text{ to } y \text{ under } \pi \right] \tag{37}$$

$$= \begin{cases} \min_{\pi \in \Pi} \mathbb{E}_{(s_1, a_1, r_1, \dots) = \xi_\pi(x)} \Big[ -\sum_t r_t \underbrace{\mathbf{1}_{s' \notin \{s_i\}_{i \in [t]}}}_{\text{not reached } s' \text{ yet}} \Big] & \text{if } \underbrace{y = s' \in \mathcal{S}}_{\text{goal is a state}}, \\[2em] \min_{\pi \in \Pi} \mathbb{E}_{(s_1, a_1, r_1, \dots) = \xi_\pi(x)} \Big[ -\sum_t r_t \underbrace{\mathbf{1}_{(s', a') \notin \{(s_i, a_i)\}_{i \in [t-1]}}}_{\text{not reached } s' \text{ \underline{and} performed } a' \text{ yet}} \Big] & \text{if } \underbrace{y = (s', a') \in \mathcal{S} \times \mathcal{A}}_{\text{goal is a state-action pair}}, \end{cases} \tag{38}$$

and

$$d_\gamma(x, y) \triangleq \log_\gamma \max_{\pi \in \Pi} \mathbb{E} \left[ \gamma^{\text{total costs from } x \text{ to } y \text{ under } \pi} \right] \tag{39}$$

is defined similarly.

If the reward function is always *negative*, $(\mathcal{X}, d_{\mathsf{sum}})$ and $(\mathcal{X}, d_\gamma)$ are *quasimetric* spaces.

*Proof of Proposition A.4.* Obviously both $d_{\mathsf{sum}}$ and $d_\gamma$ are non-negative, and satisfy *Identity of Indiscernibles* (for quasipseudometric spaces). For triangle inequality, note that for each $y$, we can instead consider alternative MDPs:

- If $y = s' \in \mathcal{S}$, modify the original MDP to make $s'$ a sink state, where performing any action yields 0 reward (i.e., 0 cost);

- If $y = (s', a') \in \mathcal{S} \times \mathcal{A}$, modify the original MDP such that performing action $a'$ in state $s'$ surely transitions to a new sink state, where performing any action yields 0 reward (i.e., 0 cost).

Obviously, both are Markovian. Furthermore, they are Stochastic Shortest Path problems with no negative costs (Guillot & Stauffer, 2020), implying that there are Markovian (i.e., stationary) optimal policies (respectively w.r.t. either minimizing expected total cost or maximizing expected $\gamma^{\text{total cost}}$). Thus optimizing over the set of stationary policies, $\Pi$, gives the optimal quantity over all possible policies, including concatenation of two stationary policies. Thus the triangle inequality is satisfied by both.

Hence, $(\mathcal{X}, d_{\mathsf{sum}})$ and $(\mathcal{X}, d_\gamma)$ are quasipseudometric spaces.

Finally, if the reward function is always *negative*, $x \neq y \implies d_{\mathsf{sum}}(x, y) > 0$ and $d_\gamma(x, y) > 0$, so $(\mathcal{X}, d_{\mathsf{sum}})$ and $(\mathcal{X}, d_\gamma)$ are quasimetric spaces. $\qquad\square$

**Remark A.5.** We make a couple remarks:

- Any MDP with a bounded reward function can be modified to have only non-positive rewards by subtracting the maximum reward (or larger);

- We have

$$d_{\mathsf{sum}}(s, (s, a)) = d_\gamma(s, (s, a)) = -\mathcal{R}(s, a). \tag{40}$$

- When the dynamics is deterministic, $d_{\mathsf{sum}} \equiv d_\gamma, \forall \gamma \in (0, 1)$.

- Unless $y$ is reachable from $x$ with probability 1 under some policy, $d_{\mathsf{sum}}(x, y) = \infty$.

- Unless $y$ is *unreachable* from $x$ with probability 1 under *all* policies, $d_{\mathsf{sum}}(x, y) < \infty$. Therefore, it is often favorable to consider $d_\gamma$ types.

- In certain MDP formulations, the reward is stochastic and/or dependent on the reached next state. The above definitions readily extend to those cases.

- $\gamma^{d_\gamma((s,a),y)}$ is very similar to Q-functions except that Q-function applies discount based on time, and $\gamma^{d_\gamma((s,a),y)}$ applies discount based on costs. We note that a Q-learning-like recurrence can also be found for $\gamma^{d_\gamma((s,a),y)}$.

  If the cost is constant in the sense for some fixed $c < 0$, $\mathcal{R}(s, a) = c, \forall (s, a) \in \mathcal{S} \times \mathcal{A}$, then time and cost are equivalent up to a scale. Therefore, $\gamma^{d_\gamma((s,a),y)}$ coincides with the optimal Q-functions for the MDPs described in proof, and $\gamma^{d_\gamma(s,y)}$ coincides with the optimal value functions for the respective MDPs.

### A.1.2    QUASIMETRIC TREEWIDTH AND GRAPH TREEWIDTH

**Definition 2.2 (Treewidth of Quasimetric Spaces (Mémoli et al., 2018)).** Consider representations of a quasimetric space $M$ as shortest-path distances on a positively-weighted directed graph. *Treewidth* of $M$ is the minimum over all such graphs' treewidths. (Recall that the treewidth of a graph (after replacing directed edges with undirected ones) is a measure of its complexity.)

Graph treewidth is a standard complexity measure of how "similar" a graph is to a tree (Robertson & Seymour, 1984). Informally speaking, if a graph has low treewidth, we can represent it as a tree, preserving all connected paths between vertices, except that in each tree node, we store a small number of vertices (from the original graph) rather than just 1.

Graph treewidth is widely used by the Theoretical Computer Science and Graph Theory communities, since many NP problems are solvable in polynomial time for graphs with bounded treewidth (Bertele & Brioschi, 1973).

## A.2 POISSON PROCESSES

**Definition 2.3 (Poisson Process).** For nonatomic measure $\mu$ on set $A$, a *Poisson process* on $A$ with *mean measure* $\mu$ is a random countable subset $P \subset A$ (i.e., the random events / points) such that

- for any disjoint measurable subsets $A_1, \ldots, A_n$ of $A$, the random variables $N(A_1), \ldots, N(A_n)$ are independent, where $N(B) \triangleq \#\{P \cap B\}$ is the number of points of $P$ in $B$, and
- $N(B)$ has the Poisson distribution with mean $\mu(B)$, denoted as $\mathrm{Pois}(\mu(B))$.

Poisson processes are usually used to model events that randomly happens "with no clear pattern", e.g., visible stars in a patch of the sky, arrival times of Internet packages to a data center. These events may randomly happen all over the sky / time. To an extent, we can say that their characteristic feature is a property of statistical independence (Kingman, 2005).

To understand this, imagine raindrops hitting the windshield of a car. Suppose that we already know that the rain is heavy, knowing the exact pattern of the raindrops hitting on the left side of the windshield tells you little about the hitting pattern on the right side. Then, we may assume that, as long as we look at regions that are disjoint on the windshield, the number of raindrops in each region are independent.

This is the fundamental motivation of Poisson processes. In a sense, from this characterization, Poisson processes are inevitable (see Sec. 1.4 of (Kingman, 2005)).

### A.2.1 POISSON RACE PROBABILITY $\mathbb{P}\left[\mathrm{Pois}(\mu_1) \leq \mathrm{Pois}(\mu_2)\right]$ AND ITS GRADIENT FORMULAS

In Fact 2.4 we made several remarks on the Poisson race probability, i.e., for *independent* $X \sim \mathrm{Pois}(\mu_1)$, $Y \sim \mathrm{Pois}(\mu_2)$, the quantity $\mathbb{P}\left[X \leq Y\right]$. In this section, we detailedly describe how we arrived at those conclusions, and provide the exact gradient formulas for differentiating $\mathbb{P}\left[X \leq Y\right]$ w.r.t. $\mu_1$ and $\mu_2$.

**From Skellam distribution CDF to Non-Central $\chi^2$ distribution CDF.** Distribution of the difference of two independent Poisson random variables is called the *Skellam* distribution (Skellam, 1946), with its parameter being the rate of the two Poissons. That is, $X - Y \sim \mathrm{Skellam}(\mu_1, \mu_2)$. Therefore, $\mathbb{P}\left[X \leq Y\right]$ is essentially the cumulative distribution function (CDF) of this Skellam at $0$. In Eq. (4) of (Johnson, 1959), a connection is made between the CDF of $\mathrm{Skellam}(\mu_1, \mu_2)$ distribution, and the CDF of a non-central $\chi^2$ distribution (which is a non-centered generalization of $\chi^2$ distribution) with two parameters $k > 0$ degree(s) of freedom and non-centrality parameter $\lambda \geq 0$): for integer $n > 0$,

$$\mathbb{P}\left[\mathrm{Skellam}(\mu_1, \mu_2) \geq n\right] = \mathbb{P}\left[\mathrm{NonCentral}\chi^2(\underbrace{2n}_{\text{degree(s) of freedom}}, \underbrace{2\mu_2}_{\text{non-centrality parameter}}) < 2\mu_1\right], \tag{41}$$

which can be evaluated using statistical computing packages such as `SciPy` (Virtanen et al., 2020) and `CDFLIB` (Burkardt, 2021; Brown et al., 1994).

**Marcum-Q-Function and gradient formulas.** To differentiate through Eq. (41), we consider representing the non-central $\chi^2$ CDF as a Marcum-Q-function (Marcum, 1950). One definition of the Marcum-Q-function $Q_M \colon \mathbb{R} \times \mathbb{R} \to \mathbb{R}$ in statistics is

$$Q_M(a, b) \triangleq \int_b^\infty x \left(\frac{x}{a}\right)^{M-1} \exp\left(-\frac{x^2 + a^2}{2}\right) I_{M-1}(ax)\, \mathrm{d}x, \tag{42}$$

where $I_{M-1}$ is the modified Bessel function of order $M - 1$. (When $M$ is non-integer, we refer readers to (Brychkov, 2012; Marcum, 1950) for definitions, which are not relevant to the discussion below.) When used in CDF of non-central $\chi^2$, we have

$$\mathbb{P}\left[\mathrm{NonCentral}\chi^2(k, \lambda) < x\right] = 1 - Q_{\frac{k}{2}}(\sqrt{\lambda}, \sqrt{x}). \tag{43}$$

Combining with Eq. (41), and using the symmetry $\text{Skellam}(\mu_1, \mu_2) \overset{d}{=} -\text{Skellam}(\mu_2, \mu_1)$, we have, for integer $n$,

$$\mathbb{P}\left[X \le Y + n\right] = \mathbb{P}\left[\text{Skellam}(\mu_1, \mu_2) \le n\right] \tag{44}$$

$$= \begin{cases} \mathbb{P}\left[\text{NonCentral}\chi^2(-2n, 2\mu_1) < 2\mu_2\right] & \text{if } n < 0 \\ 1 - \mathbb{P}\left[\text{NonCentral}\chi^2(2(n+1), 2\mu_2) < 2\mu_1\right] & \text{if } n \ge 0 \end{cases} \tag{45}$$

$$= \begin{cases} 1 - Q_{-n}(\sqrt{2\mu_1}, \sqrt{2\mu_2}) & \text{if } n < 0 \\ Q_{n+1}(\sqrt{2\mu_2}, \sqrt{2\mu_1}) & \text{if } n \ge 0. \end{cases} \tag{46}$$

Prior work (Brychkov, 2012) provides several derivative formula for the Marcum-Q-Function:

- For $n < 0$, we have

$$\frac{\partial}{\partial \mu_1} \mathbb{P}\left[X \le Y + n\right] = \frac{\partial}{\partial \mu_1} \left(1 - Q_{-n}(\sqrt{2\mu_1}, \sqrt{2\mu_2})\right) \tag{47}$$

$$= Q_{-n}(\sqrt{2\mu_1}, \sqrt{2\mu_2}) - Q_{-n+1}(\sqrt{2\mu_1}, \sqrt{2\mu_2})$$
$$\text{(Eq. (16) of (Brychkov, 2012))}$$

$$= -\left(\frac{\mu_2}{\mu_1}\right)^{-\frac{n}{2}} e^{-(\mu_1+\mu_2)} I_{-n}(2\sqrt{\mu_1\mu_2}) \quad \text{(Eq. (2) of (Brychkov, 2012))}$$

$$= -\left(\frac{\mu_2}{\mu_1}\right)^{-\frac{n}{2}} e^{-(\sqrt{\mu_1}-\sqrt{\mu_2})^2} I_{-n}^{(e)}(2\sqrt{\mu_1\mu_2}), \tag{48}$$

where $I_v^{(e)}(x) \triangleq e^{-|x|} I_v(x)$ is the exponentially-scaled version of $I_v$ that computing libraries often provide due to its superior numerical precision (e.g., SciPy (Virtanen et al., 2020)),

$$\frac{\partial}{\partial \mu_2} \mathbb{P}\left[X \le Y + n\right] = \frac{\partial}{\partial \mu_2} \left(1 - Q_{-n}(\sqrt{2\mu_1}, \sqrt{2\mu_2})\right) \tag{49}$$

$$= \left(\frac{\mu_2}{\mu_1}\right)^{-\frac{n+1}{2}} e^{-(\mu_1+\mu_2)} I_{-n-1}(2\sqrt{\mu_1\mu_2})$$
$$\text{(Eq. (19) of (Brychkov, 2012))}$$

$$= \left(\frac{\mu_2}{\mu_1}\right)^{-\frac{n+1}{2}} e^{-(\sqrt{\mu_1}-\sqrt{\mu_2})^2} I_{-n-1}^{(e)}(2\sqrt{\mu_1\mu_2}), \tag{50}$$

- For $n \ge 0$, we have

$$\frac{\partial}{\partial \mu_1} \mathbb{P}\left[X \le Y + n\right] = \frac{\partial}{\partial \mu_1} Q_{n+1}(\sqrt{2\mu_2}, \sqrt{2\mu_1}) \tag{51}$$

$$= -\left(\frac{\mu_1}{\mu_2}\right)^{n} e^{-(\mu_1+\mu_2)} I_n(2\sqrt{\mu_1\mu_2}) \quad \text{(Eq. (19) of (Brychkov, 2012))}$$

$$= -\left(\frac{\mu_1}{\mu_2}\right)^{n} e^{-(\sqrt{\mu_1}-\sqrt{\mu_2})^2} I_n^{(e)}(2\sqrt{\mu_1\mu_2}), \tag{52}$$

and,

$$\frac{\partial}{\partial \mu_2} \mathbb{P}\left[X \le Y + n\right] = \frac{\partial}{\partial \mu_2} Q_{n+1}(\sqrt{2\mu_2}, \sqrt{2\mu_1}) \tag{53}$$

$$= Q_{n+2}(\sqrt{2\mu_2}, \sqrt{2\mu_1}) - Q_{n+1}(\sqrt{2\mu_2}, \sqrt{2\mu_1})$$
$$\text{(Eq. (16) of (Brychkov, 2012))}$$

$$= \left(\frac{\mu_1}{\mu_2}\right)^{\frac{n+1}{2}} e^{-(\mu_1+\mu_2)} I_{n+1}(2\sqrt{\mu_1\mu_2}) \quad \text{(Eq. (2) of (Brychkov, 2012))}$$

$$= \left(\frac{\mu_1}{\mu_2}\right)^{\frac{n+1}{2}} e^{-(\sqrt{\mu_1}-\sqrt{\mu_2})^2} I_{n+1}^{(e)}(2\sqrt{\mu_1\mu_2}). \tag{54}$$

Setting $n = 0$ gives the proper forward and backward formulas for $\mathbb{P}\left[X \le Y\right]$.

## B  PROOFS, DISCUSSIONS AND ADDITIONAL RESULTS FOR SEC. 4: THEORETICAL ANALYSIS OF VARIOUS LEARNING ALGORITHMS

**Assumptions.**  Recall that we assumed a quasimetric space, which is stronger than a quasipseudometric space (Defn. A.1), with finite distances. These are rather mild assumptions, since any quasipseudometric with infinities can always be modified to obey these assumptions by (1) adding a small metric (e.g., $d_\epsilon(x, y) \triangleq \epsilon \mathbf{1}_{x \neq y}$ with small $\epsilon > 0$) and (2) capping the infinite distances to a large value higher than any finite distance.

**Worst-case analysis.**  In this work we focus on the *worst-case* scenario, as is common in standard (quasi)metric embedding analyses (Bourgain, 1985; Johnson & Lindenstrauss, 1984; Indyk, 2001; Mémoli et al., 2018). Such results are important because embeddings are often used as heuristics in downstream tasks (e.g., planning) which are sensitive to any error. While our negative result readily extends to the average-case scenario (since the error (distortion or violation) is arbitrary), we leave a thorough average-case analysis as future work.

**Data-independent bounds.**  We analyze possible *data-independent* bounds for various algorithms. In this sense, the positive result for PQEs (Thm. C.4) is really strong, showing good guarantees *regardless data quasimetric*. The negative result (Thm. 4.6) is also revealing, indicating that a family of algorithms should probably not be used, unless we know something more about data. *Data-independent* bounds are often of great interest in machine learning (e.g., concepts of VC-dimension (Vapnik & Chervonenkis, 2015) and PAC learning (Valiant, 1984)). An important future work is to explore data-dependent results, possibly via defining a quasimetric complexity metric that is both friendly for machine learning analysis, and connects well with combinatorics measures such as quasimetric treewidth.

**Violation and distortion metrics.**  The optimal violation has value 1. Specifically, it is 1 iff $\hat{d}$ is a quasimetric on $\mathcal{X}$ (assuming *non-negativity*). Distortion (over training set) and violation together quantify how well $\hat{d}$ learns a quasimetric consistent with the training data. A predictor can fit training data well (low distortion), but ignores basic quasimetric constraints on heldout data (high violation). Conversely, a predictor can perfectly obey the training data constraints (low violation), but doesn't actually fit training data well (high distortion). Indeed, (assuming *non-negativity* and *Identity of Indiscernibles*), perfect distortion (value 1) and violation (value 1) imply that $\hat{d}$ is a quasimetric consistent with training data.

**Relation with classical in-distribution generalization studies.**  Classical generalization studies the prediction error over the underlying data distribution, and often involves complexity of the hypothesis class and/or training data (Vapnik & Chervonenkis, 2015; McAllester, 1999). Our focus on quasimetric constraints violation is, in fact, not an orthogonal problem, but potentially a core part of in-distribution generalization for this setting. Here, the underlying distribution is supported on all pairs of $\mathcal{X} \times \mathcal{X}$. Indeed, if a learning algorithm has large distortion, it must attain large prediction error on $S \subset \mathcal{X} \times \mathcal{X}$; if it has large violation, it must violates the quasimetric constraints and necessarily admits bad prediction error on some pairs (whose true distances obey the quasimetric constraints). Thm. 4.3 (proved below) formalizes this idea, where we characterize generalization with the distortion *over all possible pairs in $\mathcal{X} \times \mathcal{X}$*.

### B.1  THM. 4.3: DISTORTION AND VIOLATION LOWER-BOUND GENERALIZATION ERROR

**Theorem 4.3 (Distortion and Violation Lower-Bound Generalization Error).**  For non-negative $\hat{d}$, $\mathsf{dis}(\hat{d}) \geq \max(\mathsf{dis}_S(\hat{d}), \sqrt{\mathsf{vio}(\hat{d})})$, where $\mathsf{dis}(\hat{d})$ captures generalization over the entire $\mathcal{X}$ space.

#### B.1.1  PROOF

*Proof of Thm. 4.3.*  It is obvious that

$$\mathsf{dis}(\hat{d}) \geq \mathsf{dis}_S(\hat{d}). \tag{55}$$

Therefore, it remains to show that $\mathsf{dis}(\hat{d}) \geq \sqrt{\mathsf{vio}(\hat{d})}$.

WLOG, say $\mathsf{vio}(\hat{d}) > 1$. Otherwise, the statement is trivially true.

By the definition of violation (see Defn. 4.2), we have, for some $x, y, z \in \mathcal{X}$, with $\hat{d}(x, z) > 0$,

$$\frac{\hat{d}(x, z)}{\hat{d}(x, y) + \hat{d}(y, z)} = \mathsf{vio}(\hat{d}). \tag{56}$$

If $\hat{d}(x, y) + \hat{d}(y, z) = 0$, then we must have one of the following two cases:

- If $d(x, y) > 0$ or $d(y, z) > 0$, the statement is true because $\mathsf{dis}(\hat{d}) = \infty$.

- If $d(x, y) = d(y, z) = 0$, then $d(x, z) = 0$ and the statement is true since $\mathsf{dis}(\hat{d}) \geq \frac{\hat{d}(x,z)}{d(x,z)} = \infty$.

It is sufficient to prove the case that $\hat{d}(x, y) + \hat{d}(y, z) > 0$. We can derive

$$\hat{d}(x, z) = \mathsf{vio}(\hat{d}) \left( \hat{d}(x, y) + \hat{d}(y, z) \right) \tag{57}$$

$$\geq \frac{\mathsf{vio}(\hat{d})}{\mathsf{dis}(\hat{d})} \left( d(x, y) + d(y, z) \right) \tag{58}$$

$$\geq \frac{\mathsf{vio}(\hat{d})}{\mathsf{dis}(\hat{d})} d(x, z). \tag{59}$$

If $d(x, z) = 0$, then $\mathsf{dis}(\hat{d}) = \infty$ and the statement is trivially true.

If $d(x, z) > 0$, above Eq. (59) implies

$$\mathsf{dis}(\hat{d}) \geq \frac{\hat{d}(x, z)}{d(x, z)} \geq \frac{\mathsf{vio}(\hat{d})}{\mathsf{dis}(\hat{d})} \implies \mathsf{dis}(\hat{d}) \geq \sqrt{\mathsf{vio}(\hat{d})}. \tag{60}$$

Combining Eqs. (55) and (60) gives the desired statement.

$\square$

## B.2 LEMMA 4.5: EXAMPLES OF OREQ ALGORITHMS

**Lemma 4.5 (Examples of OrEq Algorithms).** $k$-nearest-neighbor with Euclidean distance, MLP trained with squared loss in NTK regime, and min-norm least-squares linear regression are OrEq.

Recall the definition of Equivariant Learning Transforms.

**Definition 4.4 (Equivariant Learning Algorithms).** Given training set $\mathcal{D} = \{(z_i, y_i)\}_i \subset \mathcal{Z} \times \mathcal{Y}$, where $z_i$ are inputs and $y_i$ are targets, a learning algorithm Alg produces a function $\mathsf{Alg}(\mathcal{D}) : \mathcal{Z} \to Y$ such that $\mathsf{Alg}(\mathcal{D})(z')$ is the function's prediction on sample $z'$. Consider $\mathcal{T}$ a set of transformations $\mathcal{Z} \to \mathcal{Z}$. Alg is equivariant to $\mathcal{T}$ iff for all transform $T \in \mathcal{T}$, training set $\mathcal{D}$, $\mathsf{Alg}(\mathcal{D}) = \mathsf{Alg}(T\mathcal{D}) \circ T$, where $T\mathcal{D} = \{(Tz, y) : (z, y) \in \mathcal{D}\}$ is the training set with transformed inputs.

### B.2.1 PROOF

*Proof of Lemma 4.5.* We consider the three algorithms individually:

- $k$-**nearest neighbor with Euclidean distance.**

  It is evident that if a learning algorithm only depend on pairwise dot products (or distances), it is equivariant to orthogonal transforms, which preserve dot products (and distances). $k$-nearest-neighbor with Euclidean distance only depends on pairwise distances, which can be written in terms of dot products:

$$\|x - y\|_2^2 = x^\mathsf{T} x + y^\mathsf{T} y - 2 x^\mathsf{T} y. \tag{61}$$

  Therefore, it is equivariant to orthogonal transforms.

- **Min-norm least-squares linear regression.**

  Recall that the solution to min-norm least-squares linear regression $Ax = b$ is given by Moore–Penrose pseudo-inverse $x = A^+ b$. For any matrix $A \in \mathbb{R}^{m \times n}$ with SVD $U \Sigma V^* = A$,

and $T \in O(n)$ (where $O(n)$ is the orthogonal group in dimension $n$), we have

$$(AT^{\mathsf{T}})^+ = (U\Sigma V^*T^{\mathsf{T}})^+ = TV\Sigma^+U^* = TA^+, \tag{62}$$

where we used $T^* = T^{\mathsf{T}}$ for $T \in O(n)$. The solution for the transformed data $AT^{\mathsf{T}}$ and $b$ is thus

$$(AT^{\mathsf{T}})^+b = TA^+b. \tag{63}$$

Thus, for any new data point $\tilde{x} \in \mathbb{R}^n$ and its transformed version $T\tilde{x} \in \mathbb{R}^n$,

$$\underbrace{(T\tilde{x})^{\mathsf{T}}(AT^{\mathsf{T}})^+b}_{\text{transformed problem prediction}} = \tilde{x}^{\mathsf{T}}T^{\mathsf{T}}TA^+ = \underbrace{\tilde{x}A^+}_{\text{original problem prediction}}. \tag{64}$$

Hence, min-norm least-squares linear regression is equivariant to orthogonal transforms.

- **MLP trained with squared loss in NTK regime.**

We first recall the NTK recursive formula from (Jacot et al., 2018).

Denote the NTK for a MLP with $L$ layers with the scalar kernel $\Theta^{(L)} \colon \mathbb{R}^d \times \mathbb{R}^d \to \mathbb{R}$. Let $\beta > 0$ be the (fixed) parameter for the bias strength in the network model, and $\sigma$ be the activation function. Given $x, z \in \mathbb{R}^d$, it can be recursively defined as following. For $h \in [L]$,

$$\Theta^{(h)}(x,z) \triangleq \Theta^{(h-1)}(x,z)\dot{\Sigma}^{(h)}(x,z) + \Sigma^{(h)}(x,z), \tag{65}$$

where

$$\Sigma^{(0)}(x,z) = \frac{1}{d}x^{\mathsf{T}}z + \beta^2, \tag{66}$$

$$\Lambda^{(h-1)}(x,z) = \begin{pmatrix} \Sigma^{(h-1)}(x,x) & \Sigma^{(h-1)}(x,z) \\ \Sigma^{(h-1)}(z,x) & \Sigma^{(h-1)}(z,z) \end{pmatrix}, \tag{67}$$

$$\Sigma^{(h)}(x,z) = c \cdot \mathbb{E}_{(u,v)\sim\mathcal{N}(0,\Lambda^{(h-1)})}\left[\sigma(u)\sigma(v)\right] + \beta^2, \tag{68}$$

$$\dot{\Sigma}^{(h)}(x,z) = c \cdot \mathbb{E}_{(u,v)\sim\mathcal{N}(0,\Lambda^{(h-1)})}\left[\dot{\sigma}(u)\dot{\sigma}(v)\right], \tag{69}$$

for some constant $c$.

It is evident from the recursive formula, that $\Theta^{(h)}(x,z)$ only depends on $x^{\mathsf{T}}x$, $z^{\mathsf{T}}z$ and $x^{\mathsf{T}}z$. Therefore, the NTK is *invariant* to orthogonal transforms.

Furthermore, training an MLP in NTK regime is the same as kernel regression with the NTK (Jacot et al., 2018), which has a unique solution only depending on the kernel matrix on training set, denoted as $K_{\text{train}} \in \mathbb{R}^{n \times n}$, where $n$ is the training set size. Specifically, for training data $\{(x_i, y_i)\}_{i\in[n]}$, the solution $f^*_{\text{NTK}} \colon \mathbb{R} \to \mathbb{R}$ can be written as

$$f^*_{\text{NTK}}(x) = \begin{pmatrix} \Theta^{(L)}(x,x_1) & \Theta^{(L)}(x,x_2) & \cdots & \Theta^{(L)}(x,x_n) \end{pmatrix} K_{\text{train}}^{-1}y, \tag{70}$$

where $y = \begin{pmatrix} y_1 & y_2 & \cdots & y_n \end{pmatrix}$ is the vector of training labels.

Consider any orthogonal transform $T \in O(d)$, and the NTK regression trained on the transformed data $\{(Tx_i, y_i)\}_{i\in[n]}$. Denote the solution as $f^*_{\text{NTK},T} \colon \mathbb{R} \to \mathbb{R}$. As we have shown, $K_{\text{train}}^{-1}$ is invariant to such transforms, and remains the same. Therefore,

$$f^*_{\text{NTK},T}(Tx) = \begin{pmatrix} \Theta^{(L)}(Tx,Tx_1) & \Theta^{(L)}(Tx,Tx_2) & \cdots & \Theta^{(L)}(Tx,Tx_n) \end{pmatrix} K_{\text{train}}^{-1}y \tag{71}$$

$$= \begin{pmatrix} \Theta^{(L)}(x,x_1) & \Theta^{(L)}(x,x_2) & \cdots & \Theta^{(L)}(x,x_n) \end{pmatrix} K_{\text{train}}^{-1}y \tag{72}$$

$$= f^*_{\text{NTK}}(x). \tag{73}$$

Hence, MLPs trained (with squared loss) in NTK regime is equivariant to orthogonal transforms.

Furthermore, we note that there are many variants of MLP NTK formulas depending on details such as the particular initialization scheme and bias settings. However, they usually only lead to slight changes that do not affect our results. For example, while the above recursive NTK formula are derived assuming that the bias terms are initialized with a normal distribution (Jacot et al., 2018), the formulas for initializing bias as zeros (Geifman et al., 2020) does not affect the dependency only on dot product, and thus our results still hold true.

These cases conclude the proof. $\qquad\qquad\square$

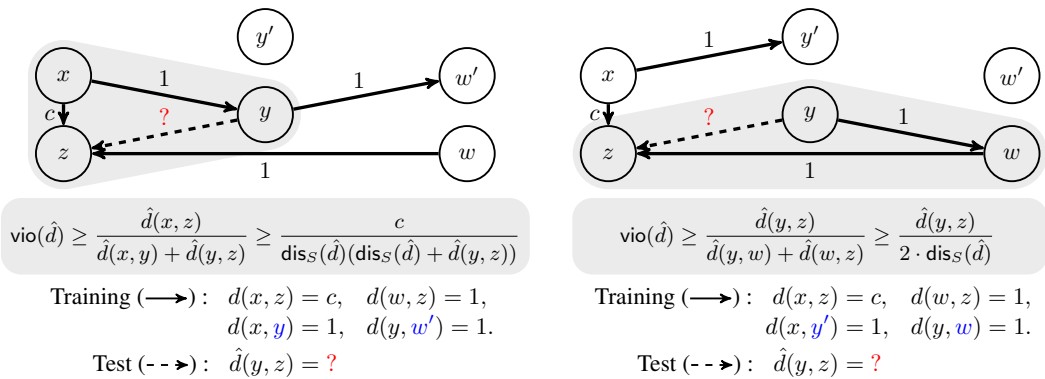

Figure 6: Two training sets pose incompatible constraints ( ⬤ ) for the test pair distance $d(y,z)$. With one-hot features, an orthogonal transform can exchange $(*, y) \leftrightarrow (*, y')$ and $(*, w) \leftrightarrow (*, w')$, leaving the test pair $(y, z)$ unchanged, but transforming the training pairs from one scenario to the other. Given either training set, an OrEq algorithm must attain same training distortion and predict identically on $(y, z)$. For appropriate $c$, this implies large distortion or violation in one of these cases.

### B.3 THM. 4.6: FAILURE OF OREQ ALGORITHMS

We start with a more precise statement of Thm. 4.6 that takes into consideration divergent $m/n^2$:

**Theorem 4.6 (Failure of OrEq Algorithms).** Let $(f_n)_n$ be an *arbitrary* sequence of large values. There is an infinite sequence of quasimetric spaces $((\mathcal{X}_n, d_n))_n$ with $|\mathcal{X}_n| = n$, $\mathcal{X}_n \subset \mathbb{R}^n$ such that, over the random training set $S$ of size $m$, any OrEq algorithm must output a predictor $\hat{d}$ that satisfies

- $\hat{d}$ fails *non-negativity*, or
- $\max(\mathsf{dis}_S(\hat{d}), \mathsf{vio}(\hat{d})) \geq f_n$  (i.e., $\hat{d}$ approximates training $S$ badly or is far from a quasimetric),

with probability $1/2 - o(1)$, as long as $S$ does not contain almost all pairs $1 - m/n^2 = \omega(n^{-1/3})$, and does not only include few pairs $m/n^2 = \omega(n^{-1/2})$.

Recall that the little-Omega notation means $f = \omega(g) \iff g = o(f)$.

#### B.3.1 PROOF

**Proof strategy.** In our proof below, we will extend the construction discussed in Sec. 4.2 to large quasimetric spaces (reproduced here as Fig. 6). To do so, we

1. Construct large quasimetric spaces containing many copies of the (potentially failing) structure in Fig. 6, where we can consider training sets of certain properties such that
   - we can pair up such training sets,
   - an algorithm equivariant to orthogonal transforms must fail on one of them,
   - for each pair, the two training sets has equal probability of being sampled;

   Then, it remains to show that with probability $1 - o(1)$ we end up with a training set of such properties.

2. Consider sampling training set as individually collecting each pair with a certain probability $p$, and carefully analyze the conditions to sample a training set with the special properties with high probability $1 - o(1)$.

3. Extend to fixed-size training sets and show that, under similar conditions, we sample a training set with the special properties with high probability $1 - o(1)$.

In the discussion below and the proof, we will freely speak of infinite distances between two elements of $\mathcal{X}$, but really mean a very large value (possibly finite). This allows us to make the argument clearer and less verbose. Therefore, we are not restricting the applicable settings of Thm. 4.6 to quasimetrics with (or without) infinite distances.

In Sec. 4.2, we showed how orthogonal-transform-equivariant algorithms can not predict $\hat{d}(y,z)$ differently for the two particular quasimetric spaces and their training sets shown in Fig. 6.

But are these the only bad training sets? Before the proof, let us consider what kinds of training sets are bad for these two quasimetric spaces. Consider the quasimetrics $d_{\mathsf{left}}$ and $d_{\mathsf{right}}$ over $\mathcal{X} \triangleq$

$\{x, y, y', z, w, w'\}$, with distances as shown in the left and right parts of Fig. 6, where we assume that the unlabeled pairs have infinite distances except in the left pattern $d(x, w') \leq 2$, and in the both patterns $d(y, z)$ has some appropriate value consistent with the respective triangle inequality.

Specifically, we ask:

- For what training sets $S_{\mathsf{left}} \subset \mathcal{X} \times \mathcal{X}$ can we interchange $y \leftrightarrow y'$ and $w \leftrightarrow w'$ on 2nd input to obtain a valid training set for $d_{\mathsf{right}}$, regardless of $c$?

- For what training sets $S_{\mathsf{right}} \subset \mathcal{X} \times \mathcal{X}$ can we interchange $y \leftrightarrow y'$ and $w \leftrightarrow w'$ on 2nd input to obtain a valid training set for $d_{\mathsf{left}}$, regardless of $c$?

Note that if $S_{\mathsf{left}}$ (or $S_{\mathsf{right}}$) satisfies its condition, the predictor $\hat{d}$ from an algorithm equivariant to orthogonal transforms must (1) predict $\hat{d}(y, z)$ identically and (2) attain the same training set distortion on it and its transformed training set. As we will see in the proof for Thm. 4.6, this implies large distortion or violation for appropriate $c$.

Intuitively, all we need is that the transformed data do not break quasimetric constraints. However, its conditions are actually nontrivial as we want to set $c$ to arbitrary:

- We can't have $(x, w) \in S_{\mathsf{right}}$ because it would be transformed into $(x, w)$ which has $d_{\mathsf{left}}(x, w) \leq 2$. Then $d_{\mathsf{right}}(x, w) \leq 2$ and then restricts the possible values of $c$ due to triangle inequality with $d_{\mathsf{right}}(w, z) = 1$. For similar reasons, we can't have $(x, w') \in S_{\mathsf{left}}$. In fact, we can't have a path of finite total distance from $x$ to $w$ (or $w'$) in $S_{\mathsf{right}}$ (or $S_{\mathsf{left}}$).

- We can not have $(y', y') \in S_{(\cdot)}$ (which has distance 0), which would get transformed into $(y', y)$ with distance 0, which (on the right pattern) would restrict the possible values of $c$ due to triangle inequality, and break our assumption of $d_{(\cdot)}$ not being a quasipseudometric. For similar reasons $(w', w')$, and cycles containing $y'$ or $w'$ with finite total distance, should be avoided.

  We note that having $(y, y)$ or $(w, w)$ would also break the non-quasipseudometric assumptions, and thus should avoid them as well (although cycles are okay here since they do not restrict values of $c$). In fact, with metrics more friendly to zero distances (than distortion and violation, which are based on distance ratios), it might be possible to allow them and obtain better bounds in the second-moment argument below in the proof for Thm. 4.6.

- Similarly, we can't have $(y, y')$, $(y', y)$, $(w, w')$, or $(w', w)$ in $S_{(\cdot)}$, as they will be mapped to $(y, y), (y', y'), (w, w)$, or $(w', w')$.

With these understandings of the pattern shown in Fig. 6, we are ready to discuss the constructed quasimetric space and training sets.

*Proof of Thm. 4.6.* Our proof follows the outline listed above.

1. **Construct large quasimetric spaces containing many copies of the (potentially failing) structure in Fig. 6.**

   For any $n > 0$, consider the following quasimetric space $(\mathcal{X}_n, d_n)$ of size $n$, with one-hot features. WLOG, assume $n = 12k$ is a multiple of 12. If it is not, set at most 11 elements to have infinite distance with every other node. This won't affect the asymptotics. Let the $n = 12k$ elements of the space be

$$
\begin{aligned}
\mathcal{X}_n = \{ & x_1^{\mathsf{left}}, \ldots, x_k^{\mathsf{left}}, \ x_1^{\mathsf{right}}, \ldots, x_k^{\mathsf{right}}, w_1^{\mathsf{left}}, \ldots, w_k^{\mathsf{left}}, \ w_1^{\mathsf{right}}, \ldots, w_k^{\mathsf{right}}, \\
& y_1^{\mathsf{left}}, \ldots, y_k^{\mathsf{left}}, \ y_1^{\mathsf{right}}, \ldots, y_k^{\mathsf{right}}, w_1'^{\mathsf{left}}, \ldots, w_k'^{\mathsf{left}}, w_{k+1}'^{\mathsf{right}}, \ldots, w_{2k}'^{\mathsf{right}}, \\
& y_1'^{\mathsf{left}}, \ldots, y_k'^{\mathsf{left}}, y_{k+1}'^{\mathsf{right}}, \ldots y_{2k}'^{\mathsf{right}}, z_1, \ldots, z_k, \quad z_{k+1}, \ldots, z_{2k} \},
\end{aligned} \tag{74}
$$

with quasimetric distances, $\forall i, j$,

$$d_n(x_i^{\text{left}}, z_j) = d_n(x_i^{\text{right}}, z_j) \quad = c \tag{75}$$

$$d_n(w_i^{\text{left}}, z_j) = d_n(w_i^{\text{right}}, z_j) \quad = 1 \tag{76}$$

$$d_n(x_i^{\text{left}}, y_i^{\text{left}}) = d_n(x_i^{\text{right}}, y_i'^{\text{right}}) = 1 \tag{77}$$

$$d_n(y_i^{\text{left}}, w_i'^{\text{left}}) = d_n(y_i^{\text{right}}, w_i^{\text{right}}) = 1 \tag{78}$$

$$d_n(x_i^{\text{left}}, w_i'^{\text{left}}) = 2 \tag{79}$$

$$d_n(y_i^{\text{left}}, z_j) = c \tag{80}$$

$$d_n(y_i^{\text{right}}, z_j) = 2, \tag{81}$$

where subscripts are colored to better show when they are the same (or different), unlisted distances are infinite (except that $d_n(u, u) = 0, \forall u \in \mathcal{X}$). Essentially, we equally divide the $12k$ nodes into 6 "types", $\{x, y, w, z, w', y'\}$, corresponding to the 6 nodes from Fig. 6, where each type has half of its nodes corresponding to the left pattern (of Fig. 6), and the other half corresponding to the right pattern, except for the $z$ types.

Furthermore,

- Among the left-pattern nodes, each set with the same subscript are bundled together in the sense that $x_i^{\text{left}}$ only has finite distance to $y_i^{\text{left}}$ which only has finite distance to $w_i'^{\text{left}}$ (instead of other $y_j^{\text{left}}$'s or $w_k'^{\text{left}}$'s). However, since distance to/from $y_i^{\text{left}}$ and $w_i^{\text{left}}$ are infinite anyways, we can pair

$$(x_i^{\text{left}}, y_i^{\text{left}}, w_i'^{\text{left}}, y_j'^{\text{left}}, w_l^{\text{left}}, z_h) \tag{82}$$

for any $i, j, l, h$, to obtain a left pattern.

- Among the right-pattern nodes, each set with the same subscript are bundled together in the sense that $x_i^{\text{right}}$ only has finite distance to $y_i'^{\text{right}}$, and $y_j^{\text{right}}$ which only has finite distance to $w_j^{\text{right}}$ (instead of other $y_j'^{\text{right}}$'s or $w_k^{\text{right}}$'s). However, since are distances are infinite anyways, we can pair

$$(x_i^{\text{right}}, y_i'^{\text{right}}, y_j^{\text{right}}, w_j^{\text{right}}, w_l'^{\text{right}}, z_h) \tag{83}$$

for any $i, j, l, h$, to obtain a right pattern.

We can see that $(\mathcal{X}, d)$ indeed satisfies all quasimetric space requirements (Defn. 2.1), including triangle inequalities (e.g., by, for each $(a, b)$ with finite distance $d_n(a, b) < \infty$, enumerating finite-length paths from $a$ to $b$).

Now consider the sampled training set $S$.

- We say $S$ is *bad* on a left pattern specified by $i_{\text{left}}, j_{\text{left}}, l_{\text{left}}, h_{\text{left}}$, if

$$S \supset \{(x_{i_{\text{left}}}^{\text{left}}, z_{h_{\text{left}}}), (x_{i_{\text{left}}}^{\text{left}}, y_{i_{\text{left}}}^{\text{left}}), (y_{i_{\text{left}}}^{\text{left}}, w_{i_{\text{left}}}'^{\text{left}}), (w_{l_{\text{left}}}^{\text{left}}, z_{h_{\text{left}}})\} \tag{84}$$

$$\emptyset = S \cap \{(y_{i_{\text{left}}}^{\text{left}}, z_{h_{\text{left}}}), (y_{i_{\text{left}}}^{\text{left}}, y_{i_{\text{left}}}^{\text{left}}), (w_{l_{\text{left}}}^{\text{left}}, w_{l_{\text{left}}}^{\text{left}}), (y_{j_{\text{left}}}'^{\text{left}}, y_{j_{\text{left}}}'^{\text{left}}), (w_{i_{\text{left}}}'^{\text{left}}, w_{i_{\text{left}}}'^{\text{left}}),$$

$$(x_{i_{\text{left}}}^{\text{left}}, w_{i_{\text{left}}}'^{\text{left}}), (y_{i_{\text{left}}}^{\text{left}}, y_{j_{\text{left}}}'^{\text{left}}), (w_{l_{\text{left}}}^{\text{left}}, w_{i_{\text{left}}}'^{\text{left}}), (y_{j_{\text{left}}}'^{\text{left}}, y_{i_{\text{left}}}^{\text{left}}), (w_{i_{\text{left}}}'^{\text{left}}, w_{l_{\text{left}}}^{\text{left}})\} \tag{85}$$

- We say $S$ is *bad* on a right pattern specified by $i_{\text{right}}, j_{\text{right}}, l_{\text{right}}, h_{\text{right}}$, if

$$S \supset \{(x_{i_{\text{right}}}^{\text{right}}, z_{h_{\text{right}}}), (x_{i_{\text{right}}}^{\text{right}}, y_{i_{\text{right}}}'^{\text{right}}), (y_{j_{\text{right}}}'^{\text{right}}, w_{j_{\text{right}}}^{\text{right}}), (w_{j_{\text{right}}}^{\text{right}}, z_{h_{\text{right}}})\} \tag{86}$$

$$\emptyset = S \cap \{(y_{j_{\text{right}}}^{\text{right}}, z_{h_{\text{right}}}), (y_{j_{\text{right}}}^{\text{right}}, y_{j_{\text{right}}}^{\text{right}}), (w_{j_{\text{right}}}^{\text{right}}, w_{j_{\text{right}}}^{\text{right}}), (y_{i_{\text{right}}}'^{\text{right}}, y_{i_{\text{right}}}'^{\text{right}}), (w_{l_{\text{right}}}'^{\text{right}}, w_{l_{\text{right}}}'^{\text{right}}),$$

$$(x_{i_{\text{right}}}^{\text{right}}, w_{j_{\text{right}}}'^{\text{right}}), (y_{j_{\text{right}}}^{\text{right}}, y_{i_{\text{right}}}'^{\text{right}}), (w_{j_{\text{right}}}^{\text{right}}, w_{l_{\text{right}}}'^{\text{right}}), (y_{i_{\text{right}}}'^{\text{right}}, y_{j_{\text{right}}}^{\text{right}}), (w_{l_{\text{right}}}'^{\text{right}}, w_{j_{\text{right}}}^{\text{right}})\} \tag{87}$$

Most importantly,

- If $S$ is bad on a left pattern specified by $i_{\text{left}}, j_{\text{left}}, l_{\text{left}}, h_{\text{left}}$, consider the orthogonal transform that interchanges $y_{i_{\text{left}}}^{\text{left}} \leftrightarrow y_{j_{\text{left}}}'^{\text{left}}$ and $w_{l_{\text{left}}}^{\text{left}} \leftrightarrow w_{i_{\text{left}}}'^{\text{left}}$ on 2nd input. In $S$, the

possible transformed pairs are

$$d(x^{\mathsf{left}}_{i_{\mathsf{left}}}, y^{\mathsf{left}}_{i_{\mathsf{left}}}) = 1 \quad \longrightarrow \quad d(x^{\mathsf{left}}_{i_{\mathsf{left}}}, y'^{\mathsf{left}}_{j_{\mathsf{left}}}) = 1, \qquad\qquad\qquad\qquad (\text{known in } S)$$

$$d(y^{\mathsf{left}}_{i_{\mathsf{left}}}, w'^{\mathsf{left}}_{i_{\mathsf{left}}}) = 1 \quad \longrightarrow \quad d(y^{\mathsf{left}}_{i_{\mathsf{left}}}, w^{\mathsf{left}}_{l_{\mathsf{left}}}) = 1, \qquad\qquad\qquad\quad (\text{known in } S)$$

$$d(u, y^{\mathsf{left}}_{i_{\mathsf{left}}}) = \infty \quad \longrightarrow \quad d(u, y'^{\mathsf{left}}_{j_{\mathsf{left}}}) = \infty, \quad (\text{poissble in } S \text{ for some } u \neq x^{\mathsf{left}}_{i_{\mathsf{left}}})$$

$$d(u, y'^{\mathsf{left}}_{j_{\mathsf{left}}}) = \infty \quad \longrightarrow \quad d(u, y^{\mathsf{left}}_{i_{\mathsf{left}}}) = \infty, \qquad\qquad (\text{poissble in } S \text{ for some } u)$$

$$d(u, w'^{\mathsf{left}}_{i_{\mathsf{left}}}) = \infty \quad \longrightarrow \quad d(u, w^{\mathsf{left}}_{l_{\mathsf{left}}}) = \infty,$$
$$(\text{poissble in } S \text{ for some } u \notin \{x^{\mathsf{left}}_{i_{\mathsf{left}}}, y^{\mathsf{left}}_{i_{\mathsf{left}}}\})$$

$$d(u, w^{\mathsf{left}}_{l_{\mathsf{left}}}) = \infty \quad \longrightarrow \quad d(u, w'^{\mathsf{left}}_{i_{\mathsf{left}}}) = \infty. \qquad (\text{poissble in } S \text{ for some } u)$$

The crucial observation is that the transformed training set just look like one sampled from a quasimetric space where

- the quasimetric space has one less set of left-pattern elements,
- the quasimetric space has one more set of right-pattern elements, and
- transformed training set is *bad* on that extra right pattern (given by the extra set of right-pattern elements),

which can be easily verified by comparing the transformed training set with the requirements in Eqs. (86) and (87).

- Similarly, if $S$ is bad on a right pattern specified by $i_{\mathsf{right}}, j_{\mathsf{right}}, l_{\mathsf{right}}, h_{\mathsf{right}}$, consider the orthogonal transform that interchanges $y^{\mathsf{right}}_{j_{\mathsf{right}}} \leftrightarrow y'^{\mathsf{right}}_{i_{\mathsf{right}}}$ and $w^{\mathsf{right}}_{j_{\mathsf{right}}} \leftrightarrow w'^{\mathsf{right}}_{l_{\mathsf{right}}}$ on 2nd input. In $S$ the possible transformed pairs are

$$d(x^{\mathsf{right}}_{i_{\mathsf{right}}}, y'^{\mathsf{right}}_{i_{\mathsf{right}}}) = 1 \quad \longrightarrow \quad d(x^{\mathsf{right}}_{i_{\mathsf{right}}}, y^{\mathsf{right}}_{j_{\mathsf{right}}}) = 1, \qquad\qquad\qquad (\text{known in } S)$$

$$d(y^{\mathsf{right}}_{j_{\mathsf{right}}}, w^{\mathsf{right}}_{j_{\mathsf{right}}}) = 1 \quad \longrightarrow \quad d(y^{\mathsf{right}}_{j_{\mathsf{right}}}, w'^{\mathsf{right}}_{l_{\mathsf{right}}}) = 1, \qquad\qquad\qquad (\text{known in } S)$$

$$d(u, y^{\mathsf{right}}_{j_{\mathsf{right}}}) = \infty \quad \longrightarrow \quad d(u, y'^{\mathsf{right}}_{i_{\mathsf{right}}}) = \infty, \quad (\text{poissble in } S \text{ for some } u)$$

$$d(u, y'^{\mathsf{right}}_{i_{\mathsf{right}}}) = \infty \quad \longrightarrow \quad d(u, y^{\mathsf{right}}_{j_{\mathsf{right}}}) = \infty,$$
$$(\text{poissble in } S \text{ for some } u \neq x^{\mathsf{right}}_{i_{\mathsf{right}}})$$

$$d(u, w'^{\mathsf{right}}_{l_{\mathsf{right}}}) = \infty \quad \longrightarrow \quad d(u, w^{\mathsf{right}}_{j_{\mathsf{right}}}) = \infty, \quad (\text{poissble in } S \text{ for some } u)$$

$$d(u, w^{\mathsf{right}}_{j_{\mathsf{right}}}) = \infty \quad \longrightarrow \quad d(u, w'^{\mathsf{right}}_{l_{\mathsf{right}}}) = \infty.$$
$$(\text{poissble in } S \text{ for some } u \notin \{x^{\mathsf{right}}_{i_{\mathsf{right}}}, y^{\mathsf{right}}_{j_{\mathsf{right}}}\})$$

Again, the crucial observation is that the transformed training set just look like one sampled from a quasimetric space where

- the quasimetric space has one less set of right-pattern elements,
- the quasimetric space has one more set of left-pattern elements, and
- transformed training set is *bad* on that extra left pattern (given by the extra set of left-pattern elements),

which can be easily verified by comparing the transformed training set with the requirements in Eqs. (84) and (85).

Therefore, when $S$ is bad on *both a left pattern and a right pattern* (necessarily on disjoint sets of pairs), we consider the following orthogonal transform composed of:

(a) both transforms specified above (which only transforms 2nd inputs),

   (so that after this we obtain *another possible training set of same size from the quasimetric space that is only different up to some permutation of $\mathcal{X}$*)

(b) a permutation of $\mathcal{X}$ (on both inputs) so that the bad left-pattern nodes and the bad right-pattern nodes exchange features,

This transforms gives *another possible training set of same size from the same quasimetric space, also is bad on a left pattern and a right pattern*. Moreover, with a particular way of select bad patterns (e.g., by the order of the subscripts), this process is *reversible*. Therefore, we have defined a way to pair up all such bad training sets.

Consider the predictors $\hat{d}_{\text{before}}$ and $\hat{d}_{\text{after}}$ trained on these two training sets (before and after transform) with an learning algorithm equivariant to orthogonal transforms. Assuming that they satisfy non-negativity and Identity of Indiscernibles, we have,

- The predictors have the same distortion over respective training sets.

  Therefore we denote this distortion as $\text{dis}_S(\hat{d})$ without specifying the predictor $\hat{d}$ or training set $S$.
- the predictors must predict the same on heldout pairs in the sense that

$$\hat{d}_{\text{before}}(y^{\text{left}}_{i_{\text{left}}}, z_{h_{\text{left}}}) = \hat{d}_{\text{after}}(y^{\text{right}}_{j_{\text{right}}}, z_{h_{\text{right}}}) \tag{88}$$

$$\hat{d}_{\text{before}}(y^{\text{right}}_{j_{\text{right}}}, z_{h_{\text{right}}}) = \hat{d}_{\text{after}}(y^{\text{left}}_{i_{\text{left}}}, z_{h_{\text{left}}}). \tag{89}$$

Focusing on the first, we denote

$$\hat{d}(y, z) \triangleq \hat{d}_{\text{before}}(y^{\text{left}}_{i_{\text{left}}}, z_{h_{\text{left}}}) = \hat{d}_{\text{after}}(y^{\text{right}}_{j_{\text{right}}}, z_{h_{\text{right}}}) \tag{90}$$

without specifying the predictor $\hat{d}$ or the specific $y$ and $z$.

However, the quasimetric constraints on heldout pairs $(y^{\text{left}}_{i_{\text{left}}}, z_{h_{\text{left}}})$ and $(y^{\text{right}}_{j_{\text{right}}}, z_{h_{\text{right}}})$ are completely different (see the left vs. right part of Fig. 6). Therefore, as shown in Fig. 6, assuming *non-negativity*, *one of the two predictors* must have total violation at least

$$\text{vio}(\hat{d}) \geq \max\left(\frac{c}{\text{dis}_S(\hat{d})(\text{dis}_S(\hat{d}) + \hat{d}(y, z))}, \frac{\hat{d}(y, z)}{2 \cdot \text{dis}_S(\hat{d})}\right). \tag{91}$$

Fixing a large enough $c$, two terms in the $\max$ of Eq. (91) can equal for some $\hat{d}(y, z)$, and are respectively decreasing and increasing in $\hat{d}(y, z)$. In that case, we have

$$\text{vio}(\hat{d}) \geq \frac{\delta}{2 \cdot \text{dis}_S(\hat{d})}, \tag{92}$$

for $\delta > 0$ such that

$$\frac{c}{\text{dis}_S(\hat{d})(\text{dis}_S(\hat{d}) + \delta)} = \frac{\delta}{2 \cdot \text{dis}_S(\hat{d})}. \tag{93}$$

Solving the above quadratic equation gives

$$\delta = \frac{-\text{dis}_S(\hat{d}) + \sqrt{\text{dis}_S(\hat{d})^2 + 8c}}{2}, \tag{94}$$

leading to

$$\text{vio}(\hat{d}) \geq \frac{-1 + \sqrt{1 + 8c/\text{dis}_S(\hat{d})^2}}{4}. \tag{95}$$

Therefore, choosing $c \geq f_n^2(4f_n + 1)^2$ gives

$$\text{dis}_S(\hat{d}) \leq f_n \tag{96}$$

$$\implies \text{vio}(\hat{d}) \geq \frac{-1 + \sqrt{1 + 8c/\text{dis}_S(\hat{d})^2}}{4} \tag{97}$$

$$\geq \frac{-1 + \sqrt{1 + 8f_n^2(4f_n + 1)^2/f_n^2}}{4} \tag{98}$$

$$= \frac{-1 + \sqrt{1 + 8(4f_n + 1)^2}}{4} \tag{99}$$

$$\geq \frac{-1 + 4f_n + 1}{4} \tag{100}$$

$$= f_n. \tag{101}$$

Hence, for training sets that are *bad* on *both a _left pattern_ and a _right pattern_*, we have shown a way to pair them up such that

- each pair of training sets have the same size, and
- the algorithm fail on one of each pair by producing a distance predictor that

- has either distortion over training set $\geq f_n$, or violation $\geq f_n$, and
- has test MSE $\geq f_n$.

**Remark B.1.** Note that all training sets of size $m$ has equal probability of being sampled. Therefore, to prove the theorem, it suffices to show that with probability $1 - o(1)$, we can sample a training set of size $m$ that is *bad* on *both a left pattern and a right pattern*.

2. **Consider sampling training set as individually collecting each pair with a certain probability** $p$**, and carefully analyze the conditions to sample a training set with the special properties with high probability** $1 - o(1)$**.**

In probabilistic methods, it is often much easier to work with independent random variables. Therefore, instead of considering uniform sampling a training set $S$ of fixed size $m$, we consider including each pair in $S$ with probability $p$, chosen independently. We will first show result based on this sampling procedure via a second moment argument, and later extend to the case with a fixed-size training set.

First, let's define some notations that ignore constants:
$$f \sim g \iff f = (1 + o(1))g \tag{102}$$
$$f \ll g \iff f = o(g). \tag{103}$$

We start with stating a standard result from the second moment method (Alon & Spencer, 2004).

**Corollary B.2 (Corollary 4.3.5 of (Alon & Spencer, 2004)).** Consider random variable $X = X_1 + X_2 + \cdots + X_n$, where $X_i$ is the indicator random variable for event $A_i$. Write $i \sim j$ if $i \neq j$ and the pair of events $(A_i, A_j)$ are not independent. Suppose the following quantity does not depend on $i$:
$$\Delta^* \triangleq \sum_{j \sim i} \mathbb{P}\left[A_j \mid A_i\right]. \tag{104}$$
If $\mathbb{E}\left[X\right] \to \infty$ and $\Delta^* \ll \mathbb{E}\left[X\right]$, then $X \sim \mathbb{E}\left[X\right]$ with probability $1 - o(1)$.

We will apply this corollary to obtain conditions on $p$ such that $S$ with probability $1 - o(1)$ is *bad* on some left pattern, and conditions such that $S$ with probability $1 - o(1)$ is *bad* on some right pattern. A union bound would then give the desired result.

- $S$ **is** *bad* **on some left pattern.**

  Recall that a left pattern is specified by $i_{\text{left}}, j_{\text{left}}, l_{\text{left}}, h_{\text{left}}$ all $\in [k]$:
  $$(x^{\text{left}}_{i_{\text{left}}}, y^{\text{left}}_{i_{\text{left}}}, w'^{\text{left}}_{i_{\text{left}}}, y'^{\text{left}}_{j_{\text{left}}}, w^{\text{left}}_{l_{\text{left}}}, z_{h_{\text{left}}}) \tag{105}$$

  Therefore, we consider $k^4 = \left(\frac{n}{12}\right)^4$ events of the form
  $$A_{i_{\text{left}}, j_{\text{left}}, l_{\text{left}}, h_{\text{left}}} \triangleq \{S \text{ is bad on the } \underline{\text{left pattern}} \text{ at } i_{\text{left}}, j_{\text{left}}, l_{\text{left}}, h_{\text{left}}\}. \tag{106}$$

  Obviously, these events are symmetrical, and the $\Delta^*$ in Eq. (104) does not depend on $i$.

  By the quasimetric space construction and the requirement for $S$ to be bad on a left pattern in Eqs. (84) and (85), we can see that $(i_{\text{left}}, j_{\text{left}}, l_{\text{left}}, h_{\text{left}}) \sim (i'_{\text{left}}, j'_{\text{left}}, l'_{\text{left}}, h'_{\text{left}})$ only if $i_{\text{left}} = i'_{\text{left}}$ or $j_{\text{left}} = j'_{\text{left}}$ or $l_{\text{left}} = l'_{\text{left}}$ or $h_{\text{left}} = h'_{\text{left}}$.

Therefore, we have

$$\mathbb{E}\left[X\right] \sim n^4 p^4 (1-p)^{10} \qquad \text{(include 4 pairs \& exclude 10 pairs)}$$

$$\Delta^* \ll n^3 p^4 (1-p)^9 \qquad \text{(share } j_{\text{left}})$$

$$+ n^3 p^2 (1-p)^7 \qquad \text{(share } i_{\text{left}})$$

$$+ n^3 p^4 (1-p)^9 \qquad \text{(share } l_{\text{left}})$$

$$+ n^3 p^4 (1-p)^{10} \qquad \text{(share } h_{\text{left}})$$

$$+ n^2 p^2 (1-p)^4 \qquad \text{(share } j_{\text{left}}, i_{\text{left}})$$

$$+ n^2 p^4 (1-p)^8 \qquad \text{(share } j_{\text{left}}, l_{\text{left}})$$

$$+ n^2 p^4 (1-p)^9 \qquad \text{(share } j_{\text{left}}, h_{\text{left}})$$

$$+ n^2 p^2 (1-p)^4 \qquad \text{(share } i_{\text{left}}, l_{\text{left}})$$

$$+ n^2 p (1-p)^6 \qquad \text{(share } i_{\text{left}}, h_{\text{left}})$$

$$+ n^2 p^3 (1-p)^9 \qquad \text{(share } l_{\text{left}}, h_{\text{left}})$$

$$+ n(1-p)^3 \qquad \text{(share } i_{\text{left}}, l_{\text{left}}, h_{\text{left}})$$

$$+ n p^3 (1-p)^8 \qquad \text{(share } j_{\text{left}}, l_{\text{left}}, h_{\text{left}})$$

$$+ n p (1-p)^3 \qquad \text{(share } j_{\text{left}}, i_{\text{left}}, h_{\text{left}})$$

$$+ n p^2 (1-p) \qquad \text{(share } j_{\text{left}}, i_{\text{left}}, l_{\text{left}})$$

$$\sim n^3 p^2 (1-p)^7 + n^2 (p^2 (1-p)^4 + p(1-p)^6) \tag{107}$$

$$+ n((1-p)^3 + p^2(1-p)). \tag{108}$$

Therefore, to apply Corollary B.2, we need to have

$$n^4 p^4 (1-p)^{10} \to \infty \tag{109}$$

$$n^3 p^2 (1-p)^7 \ll n^4 p^4 (1-p)^{10} \tag{110}$$

$$n^2 (p^2 (1-p)^4 + p(1-p)^6) \ll n^4 p^4 (1-p)^{10} \tag{111}$$

$$n((1-p)^3 + p^2(1-p)) \ll n^4 p^4 (1-p)^{10}, \tag{112}$$

which gives

$$p \gg n^{-1/2} \tag{113}$$

$$1 - p \gg n^{-1/3} \tag{114}$$

as a sufficient condition to for $S$ to be bad on some left pattern with probability $1 - o(1)$.

- $S$ **is** *bad* **on some** right pattern.

Recall that a right pattern is specified by $i_{\text{right}}, j_{\text{right}}, l_{\text{right}}, h_{\text{right}}$ all $\in [k]$:

$$(x_{i_{\text{right}}}^{\text{right}}, y_{i_{\text{right}}}^{\prime\text{right}}, y_{j_{\text{right}}}^{\text{right}}, w_{j_{\text{right}}}^{\text{right}}, w_{l_{\text{right}}}^{\prime\text{right}}, z_{h_{\text{right}}}) \tag{115}$$

Similarly, we consider $k^4 = (\frac{n}{12})^4$ events of the form

$$A_{i_{\text{right}}, j_{\text{right}}, l_{\text{right}}, h_{\text{right}}} \triangleq \{S \text{ is bad on the } \underline{\text{left pattern}} \text{ at } i_{\text{right}}, j_{\text{right}}, l_{\text{right}}, h_{\text{right}}\}. \tag{116}$$

Again, these events are symmetrical, and $\Delta^*$ in Eq. (104) does not depend on $i$.

Similarly, we have

$$\mathbb{E}\left[X\right] \sim n^4 p^4 (1-p)^{10} \qquad \text{(include 4 pairs \& exclude 10 pairs)}$$

$$\Delta^* \ll n^3 p^3 (1-p)^9 \qquad \text{(share } i_{\text{right}})$$
$$+ n^3 p^3 (1-p)^8 \qquad \text{(share } j_{\text{right}})$$
$$+ n^3 p^4 (1-p)^{10} \qquad \text{(share } h_{\text{right}})$$
$$+ n^3 p^4 (1-p)^9 \qquad \text{(share } l_{\text{right}})$$
$$+ n^2 p^2 (1-p)^4 \qquad \text{(share } i_{\text{right}}, j_{\text{right}})$$
$$+ n^2 p^2 (1-p)^9 \qquad \text{(share } i_{\text{right}}, h_{\text{right}})$$
$$+ n^2 p^3 (1-p)^8 \qquad \text{(share } i_{\text{right}}, l_{\text{right}})$$
$$+ n^2 p^2 (1-p)^7 \qquad \text{(share } j_{\text{right}}, h_{\text{right}})$$
$$+ n^2 p^3 (1-p)^5 \qquad \text{(share } j_{\text{right}}, l_{\text{right}})$$
$$+ n^2 p^4 (1-p)^9 \qquad \text{(share } h_{\text{right}}, l_{\text{right}})$$
$$+ n p^2 (1-p)^4 \qquad \text{(share } j_{\text{right}}, h_{\text{right}}, l_{\text{right}})$$
$$+ n p^2 (1-p)^8 \qquad \text{(share } i_{\text{right}}, h_{\text{right}}, l_{\text{right}})$$
$$+ n p^2 (1-p) \qquad \text{(share } i_{\text{right}}, j_{\text{right}}, l_{\text{right}})$$
$$+ n (1-p) \qquad \text{(share } i_{\text{right}}, j_{\text{right}}, h_{\text{right}})$$

$$\sim n^3 p^3 (1-p)^8 + n^2 p^2 (1-p)^4 \tag{117}$$
$$+ n(1-p). \tag{118}$$

Therefore, to apply Corollary B.2, we need to have

$$n^4 p^4 (1-p)^{10} \to \infty \tag{119}$$
$$n^3 p^3 (1-p)^8 \ll n^4 p^4 (1-p)^{10} \tag{120}$$
$$n^2 p^2 (1-p)^4 \ll n^4 p^4 (1-p)^{10} \tag{121}$$
$$n(1-p) \ll n^4 p^4 (1-p)^{10}, \tag{122}$$

which gives

$$p \gg n^{-3/4} \tag{123}$$
$$1-p \gg n^{-1/3} \tag{124}$$

as a sufficient condition to for $S$ to be bad on some right pattern with probability $1 - o(1)$.

So, by union bound, as long as

$$p \gg n^{-1/2} \tag{125}$$
$$1-p \gg n^{-1/3}, \tag{126}$$

$S$ is bad on some left pattern *and* some right pattern with probability $1 - o(1)$.

3. **Extend to fixed-size training sets and show that, under similar conditions, we sample a training set with the special properties with high probability** $1 - o(1)$**.**

   To extend to fixed-size training sets, we consider the following alteration procedure:

   (a) Sample training set $S$ by independently include each pair with probability $p \triangleq \frac{m+\delta}{n^2}$, for some $\delta > 0$.

   (b) Show that with high probability $1 - o(1)$, we end up with $[m, m + 2\delta]$ pairs in $S$.

   (c) Make sure that $p$ satisfy Eq. (125) and Eq. (126) so that $S$ is bad on some left pattern *and* some right pattern with high probability $1 - o(1)$.

   (d) Randomly discard the additional pairs, and show that with high probability $1 - o(1)$ this won't affect that $S$ is bad on some left pattern *and* some right pattern.

   We now consider each step in details:

(a) **Sample training set $S$ by independently include each pair with probability $p \triangleq \frac{m+\delta}{n^2}$, for some $\delta > 0$.**

For $p \triangleq \frac{m+\delta}{n^2}$, the number of pairs in the training set is distributed as

$$\text{Binomial}(n^2, \frac{m+\delta}{n^2}). \tag{127}$$

(b) **Show that with high probability $1 - o(1)$, we end up with $[m, m + 2\delta]$ pairs in $S$.**

Standard Binomial concentration tells us that,

$$\delta \gg n\sqrt{p(1-p)} \implies \mathbb{P}\left[\text{Binomial}(n^2, \frac{m+\delta}{n^2}) \notin [m, m+2\delta]\right] \to 0, \tag{128}$$

which can be satisfied if

$$\delta \gg n. \tag{129}$$

(c) **Make sure that $p$ satisfy Eq. (125) and Eq. (126) so that $S$ is bad on some left pattern *and* some right pattern with high probability $1 - o(1)$.**

Therefore, we want

$$\frac{m+\delta}{n^2} \gg n^{-1/2} \tag{130}$$

$$1 - \frac{m+\delta}{n^2} \gg n^{-1/3}. \tag{131}$$

(d) **Randomly discard the additional pairs, and show that with high probability $1 - o(1)$ this won't affect that $S$ is bad on some left pattern *and* some right pattern.**

Consider any specific bad left pattern *and a* right pattern in $S$. It is sufficient that we don't break these two patterns during discarding.

Since we only discard pairs, it suffices to only consider the pairs we want to preserve, which are a total of 8 pairs across two patterns.

Each such pair is discarded the probability $\leq \frac{2\delta}{m}$, since we remove at most $2\delta$ pairs. By union bound,

$$\mathbb{P}\left[\text{all 8 pairs are preserved}\right] \geq 1 - \frac{16\delta}{m}. \tag{132}$$

Hence, it suffices to make sure that

$$\delta \ll m. \tag{133}$$

Collecting all requirements, we have

$$\delta \gg n \tag{134}$$

$$\frac{m+\delta}{n^2} \gg n^{-1/2} \tag{135}$$

$$1 - \frac{m+\delta}{n^2} \gg n^{-1/3} \tag{136}$$

$$\delta \ll m. \tag{137}$$

Assume that

$$\frac{m}{n^2} \gg n^{-1/2} \tag{138}$$

$$1 - \frac{m}{n^2} \gg n^{-1/3}. \tag{139}$$

It can be easily verified that using $\delta \triangleq n^{1.1}$ satisfies all conditions.

Hence, for a uniformly randomly sampled training set $S$ with size $m$, $S$ is bad on some left pattern *and* some right pattern with high probability $1 - o(1)$, as long as

$$\frac{m}{n^2} \gg n^{-1/2} \tag{140}$$

$$1 - \frac{m}{n^2} \gg n^{-1/3}. \tag{141}$$

This is exactly the condition we need to prove the theorem (see Remark B.1).

This concludes the proof. □

### B.3.2 DISCUSSIONS

**Training set size dependency.** Intuitively, when the training set has almost all pairs, violation can be lowered by simply fitting training set well; when it is small and sparse, the learning algorithm may have an easier job finding some consistent quasimetric. Thm. 4.6 shows that, outside these two cases, algorithms equivariant to orthogonal transforms can fail. Note that for the latter case, Thm. 4.6 requires the training fraction to decrease slower than $n^{-1/2}$, which rules out training sizes that is linear in $n$. We leave improving this result as future work. Nonetheless, Thm. 4.6 still covers common scenarios such as a fixed fraction of all pairs, and highlights that a training-data-agnostic result (such as the ones for PQEs) is not possible for these algorithms.

**Proof techniques.** In embedding theory, it is quite standard to analyze quasimetrics as directed graphs due to their lack of nice metric structure. In the proof for Thm. 4.6, we used abundant techniques from the probabilistic method, which are commonly used for analyzing graph properties in the asymptotic case, including Corollary B.2 from the second moment technique, and the alteration technique to extend to fixed-size training sets. While such techniques may be new in learning theory, they are standard for characterizing asymptotic probabilities on graphs, which quasimetrics are often analyzed as (Charikar et al., 2006; Mémoli et al., 2018).

To provide more intuition on why these techniques are useful here, we note that the construction of a training set of pairs is essentially like constructing an Erdős-Rényi random graph on $n^2$ vertices. Erdős-Rényi (undirected) random graphs come in two kinds:

- Uniformly sampling a fixed number of $m$ edges;
- Adding an edge between each pair with probability $p$, decided independently.

The latter, due to its independent decisions, is often much easy to analyze and preferred by many. The alteration technique (that we used in the proof) is also a standard way to transfer a result on a random graph of the latter type, to a random graph of the former type (Bollobás & Béla, 2001). Readers can refer to (Alon & Spencer, 2004; Bollobás & Béla, 2001; Erdős & Rényi, 1959) for more in-depth treatment of these topics.

**Generalization to other transforms.** The core of this construction only relies on the ability to swap (concatenated) inputs between $(x, y) \leftrightarrow (x, y')$ and between $(y, w) \leftrightarrow (y, w')$ via a transform. For instance, here the orthogonal transforms satisfy this requirement on one-hot features. Therefore, the result can also be generalized to other transforms and features with the same property. Our stated theorem focuses on orthogonal transforms because they correspond to several common learning algorithms (see Lemma 4.5). If a learning algorithm is equivariant to some other transform family, it would be meaningful to generalize this result to that transform family, and obtain a similar negative result. We leave such extensions as future work.

### B.3.3 COROLLARY OF DISTORTION AND VIOLATION FOR UNCONSTRAINED MLPS

**Corollary B.3 (Distortion and Violation of Unconstrained MLPs).** Let $(f_n)_n$ be an arbitrary sequence of desired violation values. There is an infinite collection of quasimetric spaces $((\mathcal{X}_n, d_n))_{n=1,2,\dots}$ with $|\mathcal{X}_n| = n$, $\mathcal{X}_n \subset \mathbb{R}^n$ such that MLP trained with squared loss in NTK regime converges to a function $\hat{d}$ that either

- fails non-negativity, or
- $\mathsf{vio}(\hat{d}) \geq f_n$,

with probability $1/2 - o(1)$ over the random training set $S$ of size $m$, as long as $S$ does not contain almost all pairs $1 - m/n^2 = \omega(n^{-1/3})$, and does not only include few pairs $m/n^2 = \omega(n^{-1/2})$.

*Proof of Corollary B.3.* This follows directly from Thm. 4.6 and standard NTK convergence results obtained from the kernel regression optimality and the positive-definiteness of the NTK. In particular, Proposition 2 of (Jacot et al., 2018) claims that the NTK is positive-definite when restricted to a hypersphere. Since the construction in proof of Thm. 4.6 uses one-hot features, the input (concatenation of two features) lie on the hypersphere with radius $\sqrt{2}$. Hence, the NTK is guaranteed positive definite. $\square$

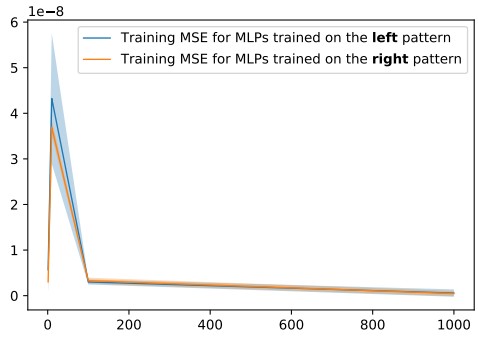
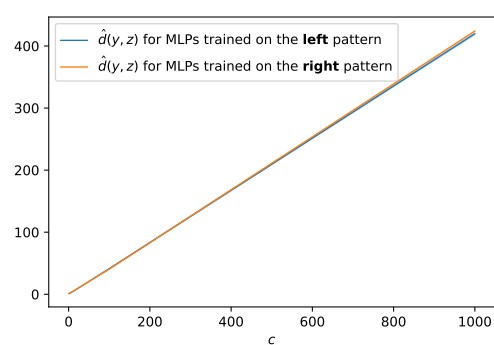

(a) Training losses for varying $c$. Note the scale of the vertical axis.

(b) Prediction on heldout pair $\hat{d}(y, z)$ for varying $c$.

Figure 7: Training unconstrained MLPs on the toy failure construction discussed in Sec. 4.2 (reproduced as Fig. 6). Two patterns in the construction have different constraints on distance of the heldout pair $(y, z)$. Plots show mean and standard deviations over 5 runs. **Left:** All training conclude with small training error. **Right:** Trained MLPs predict identically for both patterns. Here standard deviation is small compared to mean and thus not very visible.

### B.3.4   EMPIRICAL VERIFICATION OF THE FAILURE CONSTRUCTION

We train unconstrained MLPs on the toy failure construction discussed in Sec. 4.2 (reproduced as Fig. 6). The MLP uses 12-1024-1 architecture with ReLU activations, takes in the concatenated one-hot features, and directly outputs predicted distances. Varying $c \in \{1, 10, 100, 1000\}$, we train the above MLP 5 times on each of the two patterns in Fig. 6, by regressing towards the training distances via MSE loss.

In Fig. 7, we can see that all training runs conclude with small training error, and indeed the trained MLPs predict very similarly on the heldout pair, regardless whether it is trained on the left or right pattern of Fig. 6, which restricts the heldout pair distance differently.

This verifies our theory (Thm. 4.6 and Corollary B.3) that algorithms equivariant to orthogonal transforms (including MLPs in NTK regime) cannot distinguish these two cases and thus must fail on one of them.

## C   PROOFS AND DISCUSSIONS FOR SEC. 5: POISSON QUASIMETRIC EMBEDDINGS (PQES)

### C.1   NON-DIFFERENTIABILITY OF CONTINUOUS-VALUED STOCHASTIC PROCESSES

In this section we formalize the argument presented in Sec. 5.3 to show why continuous-valued stochastic processes lead to non-differentiability. Fig. 8 also provides a graphical illustration of the general idea.

**Proposition C.1 (Quasimetric Embeddings with Continuous-Valued Stochastic Processes are not Differentiable).** Consider any $\mathbb{R}^k$-valued stochastic process $\{R(u)\}_{u \in \mathbb{R}^d}$ such that $u \neq u' \implies \mathbb{P}[R(u) = R(u')] < c$ for some universal constant $c < 1$. Then $\mathbb{P}[R(u) \leq R(u')]$ is not differentiable at any $u = u'$.

*Proof of Proposition C.1.* Assume that the quantity is differentiable. Then it must be continuous in $u$ and $v$.

We will use the $(\epsilon, \delta)$-definition of continuity.

At any $u \in \mathbb{R}^d$, consider small $\epsilon \in (0, \frac{1-c}{3})$. By continuity, since
$$\mathbb{P}[R(u) \leq R(u)] = \mathbb{P}[R(u+\delta) \leq R(u+\delta)] = 1 \tag{142}$$

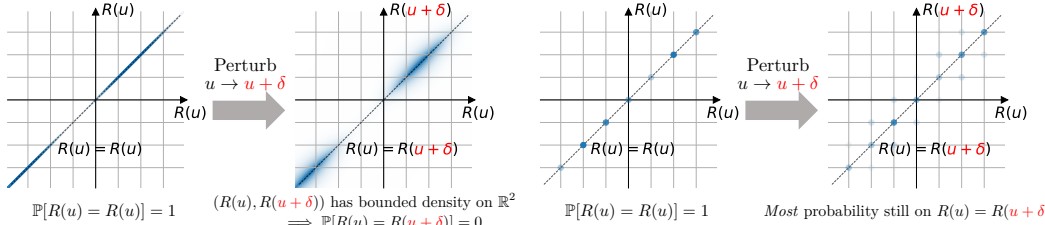

Figure 8: Bivariate distributions from different stochastic processes. **Left:** In a continuous-valued process (where $(N_\theta, N_{\theta'})$ has bounded density if $\theta \neq \theta'$), perturbing one $\theta \to \theta + \epsilon$ leaves $\mathbb{P}[N_\theta = N_{\theta+\epsilon}] = 0$. Then one of $\mathbb{P}[N_\theta \leq N_{\theta+\epsilon}]$ and $\mathbb{P}[N_{\theta+\epsilon} \leq N_\theta]$ must be far away from 1 (as they sum to 1), breaking differentiability at either $\mathbb{P}[N_\theta \leq N_\theta] = 1$ or $\mathbb{P}[N_{\theta+\epsilon} \leq N_{\theta+\epsilon}] = 1$. **Right:** For discrete-valued processes, most probability can still be left on $N_\theta = N_{\theta+\epsilon}$ and thus do not break differentiability.

we can find $\epsilon \in \mathbb{R}^d$ such that

$$\mathbb{P}[R(u) \leq R(u+\delta)] \geq 1 - \epsilon \tag{143}$$

$$\mathbb{P}[R(u+\delta) \leq R(u)] \geq 1 - \epsilon. \tag{144}$$

However, by assumption, $\mathbb{P}[R(u) = R(u+\delta)] < c$. Therefore,

$$\mathbb{P}[R(u) \leq R(u+\delta)] \geq 1 - \epsilon \tag{145}$$

$$\mathbb{P}[R(u+\delta) < R(u)] \geq 1 - \epsilon - c, \tag{146}$$

which implies

$$1 = \mathbb{P}[R(u) \leq R(u+\delta)] + \mathbb{P}[R(u+\delta) < R(u)] \geq 2 - 2\epsilon - c \geq \frac{5}{3} - \frac{2}{3}c > 1. \tag{147}$$

By contradiction, the quantity must not be differentiable at any $u = u'$. $\qquad\square$

### C.2 PQE-GG: GAUSSIAN-BASED MEASURE AND GAUSSIAN SHAPES

In Sec. 5.1, we presented the following PQE-LH formulation for Lebesgue measures and half-lines:

$$d_z^{\mathsf{PQE\text{-}LH}}(u, v) \triangleq \sum_i \alpha_i \cdot \left(1 - \exp\left(-\sum_j (u_{i,j} - v_{i,j})^+\right)\right). \tag{10}$$

Here, $u_{i,j}$ and $v_{i,j}$ receive zero gradient when $u_{i,j} \leq v_{i,j}$.

**Gaussian shapes parametrization.** We therefore consider a set parametrization where no one set is entirely contained in a different set— the regions regions $\subset \mathbb{R}^2$ between an axis and a 1D Gaussian density function of fixed variance $\sigma^2_{\mathsf{shape}} = 1$. That is, for each given $u \in R$, we consider sets

$$A_{\mathcal{N}}(\mu) \triangleq \{(a, b) : b \in [0, f_{\mathcal{N}}(a; \mu, 1)]\}, \tag{148}$$

where $f_{\mathcal{N}}(b; \mu, \sigma^2)$ denotes the density of 1D Gaussian $\mathcal{N}(\mu, \sigma^2)$ with mean $\mu$ and variance $\sigma^2$ evaluated at $b$. Since the Gaussian density function have unbounded support, these sets, which are translated versions of each other, never have one set fully contained in another. For latent $u \in \mathbb{R}^{h \times k}$ reshaped as 2D, our set parametrizations are,

$$u \to A_{i,j}(u) \triangleq A_{\mathcal{N}}(u_{i,j}), \qquad i \in [h], j \in [k]. \tag{149}$$

**A Gaussian-based measure.** These subsets of $\mathbb{R}^2$ always have Lebesgue measure 1, which would make PQE symmetrical (if used with a (scaled) Lebesgue measure). Thus, we use an alternative $\mathbb{R}^2$ measure given by the product of a $\mathbb{R}$ Lebesgue measure on the $b$-dimension (i.e., dimension of the function value of the Gaussian density) and a $\mathbb{R}$ Gaussian measure on the $a$-dimension (i.e., dimension on the input of the Gaussian density) centered at 0 with *learnable* variances $(\sigma^2_{\mathsf{measure}})_{i,j}$. To avoid being constrained by the bounded total measure of 1, we also optimize learnable positive scales $c_{i,j} > 0$. Hence, the each Poisson process has a mean measure as the product of a $\mathbb{R}$ Lebesgue measure and a $\mathbb{R}$ Gaussian with learnable standard deviation, then scaled with a learnable scale.

Note that the Gaussian measure should not be confused with the Gaussian shape. Their parameters also are fully independent with one another.

**Computing measures of Gaussian shapes and their intersections.** The intersection of two such Gaussian shapes is formed by two Gaussian tail shapes, reflected around the middle point of the two Gaussian means (since they have the same standard deviation $\sigma^{\text{shape}} = 1$). Hence, it is sufficient to describe how to integrate a Gaussian density on a Gaussian measure over an interval. Applying this with different intervals would give the measure of the intersection, and the measures of the two Gaussian shapes. Omit indices $i, j$ for clarity. Formally, we integrate the Gaussian density $f_{\mathcal{N}}(a; u, \sigma^2_{\text{shape}})$ over the centered Gaussian measure with variance $\sigma^2_{\text{measure}}$, which has density $f_{\mathcal{N}}(a; 0, \sigma^2_{\text{measure}})$:

$$\int c \cdot f_{\mathcal{N}}(a; u, \sigma^2_{\text{shape}}) f_{\mathcal{N}}(a; 0, \sigma^2_{\text{measure}}) \, \mathrm{d}a, \tag{150}$$

which is also another Gaussian integral (e.g., considered as integrating the product measure along the a line of the form $y = x + u$). After standard algebraic manipulations (omitted here), we obtain

$$\int c \cdot f_{\mathcal{N}}(a; u, \sigma^2_{\text{shape}}) f_{\mathcal{N}}(a; 0, \sigma^2_{\text{measure}}) \, \mathrm{d}a \tag{151}$$

$$= \frac{c \cdot \exp\left(-u^2/\sigma^2_{\text{total}}\right)}{\sqrt{2\pi\sigma^2_{\text{total}}}} \int f_{\mathcal{N}}\left(a; u\frac{\sigma^2_{\text{measure}}}{\sigma^2_{\text{total}}}, \frac{\sigma^2_{\text{shape}}\sigma^2_{\text{measure}}}{\sigma^2_{\text{total}}}\right) \, \mathrm{d}a, \tag{152}$$

for

$$\sigma^2_{\text{total}} \triangleq \sigma^2_{\text{shape}} + \sigma^2_{\text{measure}}. \tag{153}$$

This can be easily evaluated using statistical computing packages that supports computing the error function and/or Gaussian CDF. Moreover, this final form is also readily differentiable with standard gradient formulas. To summarize,

- each set $A(u)$ has total measure

$$\frac{c}{\sqrt{2\pi\sigma^2_{\text{total}}}} \exp\left(-u^2/\sigma^2_{\text{total}}\right); \tag{154}$$

- the intersection of $A(v)$ and $A(u_2)$, for $v \leq u_2$ has measure

$$\frac{c \cdot \exp\left(-u_2^2/\sigma^2_{\text{total}}\right)}{\sqrt{2\pi\sigma^2_{\text{total}}}} \int_{-\infty}^{\frac{v+u_2}{2}} f_{\mathcal{N}}\left(a; u_2\frac{\sigma^2_{\text{measure}}}{\sigma^2_{\text{total}}}, \frac{\sigma^2_{\text{shape}}\sigma^2_{\text{measure}}}{\sigma^2_{\text{total}}}\right) \, \mathrm{d}a \tag{155}$$

$$+ \frac{c \cdot \exp\left(-v^2/\sigma^2_{\text{total}}\right)}{\sqrt{2\pi\sigma^2_{\text{total}}}} \int_{\frac{v+u_2}{2}}^{+\infty} f_{\mathcal{N}}\left(a; v\frac{\sigma^2_{\text{measure}}}{\sigma^2_{\text{total}}}, \frac{\sigma^2_{\text{shape}}\sigma^2_{\text{measure}}}{\sigma^2_{\text{total}}}\right) \, \mathrm{d}a. \tag{156}$$

**Interpretation and representing any total order.** Consider two Gaussian shapes $A(v)$ and $A(u_2)$. Note that the Gaussian-based measure $\mu_{\text{Gaussian}}$ is symmetric around and centered at 0. Therefore,

$$|v| < |u_2| \implies \mu_{\text{Gaussian}}(A(v)) > \mu_{\text{Gaussian}}(A(u_2)) \tag{157}$$

$$\implies \mu_{\text{Gaussian}}(A(v) \setminus A(u_2)) > \mu_{\text{Gaussian}}(A(u_2) \setminus A(v)). \tag{158}$$

Moreover, scaling the rates of a Poisson makes it more concentrated (as a Poisson's mean grows as the square of its standard deviation) so that $\lim_{c \to \infty} \mathbb{P}\left[\text{Pois}(c\mu_1) \leq \text{Pois}(c\mu_2)\right] = \mathbf{1}_{\mu_1 < \mu_2}$ for $\mu_1 \neq \mu_2$. Then any total order can be represented as the limit of a Poisson process with Gaussian shapes, with the shapes' having their means arranged according to the total order, as the scale on the Gaussian-based measure grows to infinity.

## C.3 Theoretical Guarantees for PQEs

**Theorem 5.2 (Distortion and violation of PQEs).** Under the assumptions of Sec. 4, *any* quasimetric space with size $n$ and treewidth $t$ admits a PQE-LH and a PQE-GG with distortion $\mathcal{O}(t \log^2 n)$ and violation 1, with an expressive encoder (e.g., a ReLU network with $\geq 3$ layers and polynomial width).

In Sec. 5.4, we presented the above theoretical distortion and violation guarantees for PQE-LH and PQE-GG. Furthermore, we commented that the same guarantees apply to more generally to PQEs satisfying a mild condition. Here, we first precisely describe this condition, show that PQE-LH and

PQE-GG do satisfy it, state and prove the general result, and then show the above as a straightforward corollary.

### C.3.1 THE CONCENTRATION PROPERTY

Recall that PQEs are generally defined with measures $\mu$ and set parametrizations $A$ as

$$d_z^{\text{PQE}}(u, v; \mu, A, \alpha) \triangleq \sum_i \alpha_i \cdot \mathbb{E}_{\pi_z \sim \Pi_z^{\text{PQE}}(\mu_i, A_i)} [\pi_z(u, v)], \tag{13}$$

where

$$\mathbb{E}_{\pi_z \sim \Pi_z^{\text{PQE}}(\mu, A)}[\pi_z(u, v)] \triangleq 1 - \prod_j \mathbb{P}\left[N_j(A_j(u)) \leq N_j(A_j(v))\right]. \tag{12}$$

Because the measures $\mu$ and set parametrizations $A$ themselves may have parameters (e.g., as in PQE-GG), we consider them as classes of PQEs. E.g., PQE-GG is a class of PQEs such that the $\mu$ is the specific Gaussian-based form, and $A$ is the specific Guassian-shape.

**Definition C.2 (Concetration Property of PQEs).** Consider a PQE class with $h$ mixtures of quasi-partition distributions, each from $k$ Poisson processes. We say that it has concentration property if it satisfies the following. Consider any finite subset of $\mathcal{X}' \subset \mathcal{X}$, and arbitrary function $g \colon \mathcal{X} \to \mathbb{R}^{h \times k}$. There exists a sequence of $((f^{(n)}, \mu^{(n)}, A^{(n)})_n$ such that

- $f^{(n)} \colon \mathcal{X}' \to \mathbb{R}^d$,
- $\mu^{(n)}, A^{(n)}$ are valid members of this PQE,
- $\mathbb{E}_{\pi_z \sim \Pi_z^{\text{PQE}}(\mu_i, A_i)} \left[\pi_z(f^{(n)}(x'), f^{(n)}(y'))\right]$ uniformly converges to $1 - \prod_j \mathbf{1}_{g(x)_{i,j} \leq g(y)_{i,j}}$, over all mixtures $i$ and pairs $x, y \in \mathcal{X}'$.

**A sufficient condition.** It suffices to make the probabilities

$$(x, y, i, j) \to \mathbb{P}\left[N_j(A_j(u)) \leq N_j(A_j(v))\right], \tag{159}$$

along some PQE sequence uniformly converge to the indicators

$$(x, y, i, j) \to \mathbf{1}_{g(x')_{i,j} \leq g(y')_{i,j}}. \tag{160}$$

This is sufficient since product of bounded functions is uniformly convergent, if each function is. Both statements below together form **a sufficient condition** for Eq. (159) to uniformly converge to Eq. (160):

1. For any $g$, there exists a specific PQE of this class satisfying
   - Measures (of set differences) are consistent with $g$ with some margin $\epsilon > 0$: $\forall i \in [h], j \in [k], x \in \mathcal{X}', y \in \mathcal{X}'$,
     $g(x)_{i,j} < g(y)_{i,j} \iff \mu_{i,j}(A_{i,j}(f(x)) \setminus A_{i,j}(f(y))) + \epsilon < \mu_{i,j}(A_{i,j}(f(y)) \setminus A_{i,j}(f(x)))$
     $g(x)_{i,j} = g(y)_{i,j} \iff \mu_{i,j}(A_{i,j}(f(x)) \setminus A_{i,j}(f(y))) = \mu_{i,j}(A_{i,j}(f(y)) \setminus A_{i,j}(f(x))) = 0$.
   - Either of the following:
     - One side must be zero: $\forall i \in [h], j \in [k], x \in \mathcal{X}, y \in \mathcal{X}$,
       $$(\mu_{i,j}(A_{i,j}(f(x)) \setminus A_{i,j}(f(y)))) (\mu_{i,j}(A_{i,j}(f(y)) \setminus A_{i,j}(f(x)))) = 0, \tag{161}$$
     - Max measure is bounded by some constant $c > 0$:
       $$\max_{x,y,i,j} \mu_{i,j}(A_{i,j}(f(x)) \setminus A_{i,j}(f(y))) \leq c. \tag{162}$$

2. For any given specific PQE of this class, for any positive scale $d > 0$, there is another PQE (with same formulation) whose measures (of set differences) equal exactly those of the given PQE scaled by $d$.

We now show that this is a sufficient condition. Note that a Poisson distribution has standard deviation equal to square root of its mean. This means that as we scale the rate of a Poisson, it becomes more concentrated. Applying to Poisson race probability, we have, for $0 \leq \mu_1 + \epsilon < \mu_2$,

- one direction of Poisson race probability:

$$\mathbb{P}\left[\operatorname{Pois}(d \cdot \mu_1) \le \operatorname{Pois}(d \cdot \mu_2)\right] \tag{163}$$

$$\ge \mathbb{P}\left[|\operatorname{Pois}(d \cdot \mu_2) - \operatorname{Pois}(d \cdot \mu_1) - d(\mu_2 - \mu_1)| \le d(\mu_2 - \mu_1)\right] \tag{164}$$

$$\ge 1 - \frac{\mu_1 + \mu_2}{d(\mu_2 - \mu_1)^2} \tag{165}$$

$$\ge \begin{cases} 1 - \frac{2}{d\epsilon} & \text{if } \mu_1 = 0 \\ 1 - \frac{2c}{d\epsilon^2} & \text{if } \mu_2 < c; \end{cases} \tag{166}$$

- the other direction of Poisson race probability:

$$\mathbb{P}\left[\operatorname{Pois}(d \cdot \mu_2) \le \operatorname{Pois}(d \cdot \mu_1)\right] \tag{167}$$

$$\le \mathbb{P}\left[|\operatorname{Pois}(d \cdot \mu_2) - \operatorname{Pois}(d \cdot \mu_1) - d(\mu_2 - \mu_1)| \ge d(\mu_2 - \mu_1)\right] \tag{168}$$

$$\le \frac{\mu_1 + \mu_2}{d(\mu_2 - \mu_1)^2} \tag{169}$$

$$\le \begin{cases} \frac{2}{d\epsilon} & \text{if } \mu_1 = 0 \\ \frac{2c}{d\epsilon^2} & \text{if } \mu_2 < c. \end{cases} \tag{170}$$

Therefore, applying to scaled versions of the PQE from Item 1 above, we have thus obtained the desired sequence, where Eq. (159) uniformly converges to Eq. (160) with rate $\mathcal{O}(1/d)$.

**Lemma C.3.** PQE-LH and PQE-GG both have the concentration property.

*Proof of Lemma C.3.* We show that both classes satisfy the above sufficient condition.

- PQE-LH: Lebesgue measure $\lambda$ and half-lines.

  WLOG, since $\mathcal{X}$ is countable, we assume that $g$ satisfies

  $$g(x)_{i,j} \ne g(y)_{i,j} \implies |g(x)_{i,j} - g(y)_{i,j}| > 1, \qquad \forall i \in [h], j \in [k], x \in \mathcal{X}', y \in \mathcal{X}'. \tag{171}$$

  The encoder in Item 1 above $f \colon \mathcal{X} \to \mathbb{R}^{h \times k}$ can simply be $g$. We then have

  $$\mu_{i,j}(A_{i,j}(f(y)) \backslash A_{i,j}(f(x))) = \operatorname{Leb}((-\infty, g(y)] \backslash (-\infty, g(x)]) = (g(y)_{i,j} - g(x)_{i,j})^+. \tag{172}$$

  This ensures that one side is always zero. Furthermore, scaling can be done by simply scaling the encoder $f$. Hence, PQE-LH satisfies this constraint.

- PQE-GG: Gaussian-based measure and Gaussian shapes (see Appendix C.2).

  Because $\mathcal{X}'$ is finit, we can have positive constant margin for the PQE requirements in Item 1. (Infinite $\mathcal{X}'$ does not work because the total measure is finite (for a specific PQE-GG with specific values of the scaling).) Concretely, we satisfy both requirements via

  - in descending order of $g(\cdot)_{i,j}$ we assign Gaussian shapes increasingly further from the origin;
  - scaling comes from that we allow scaling the Gaussian-based measure.

  Hence, PQE-GG satisfies this constraint for finite $\mathcal{X}$.

  $\square$

### C.3.2  A GENERAL STATEMENT

We now state the general theorem for PQEs with the above concentration property.

**Theorem C.4 (Distortion and violation of PQEs (General)).** Consider any PQE class with the concentration property. Under the assumptions of Sec. 4, *any* quasimetric space with size $n$ and treewidth $t$ admits such a PQE with distortion $\mathcal{O}(t \log^2 n)$ and violation 1, with an expressive encoder (e.g., a ReLU network with $\ge 3$ hidden layers, $\mathcal{O}(n)$ hidden width, and $\mathcal{O}(n^2)$ quasipartition distributions, each with $\mathcal{O}(n)$ Poisson processes.).

Before proving this more general theorem, let us extend a result from Mémoli et al. (2018).

**Lemma C.5 (Quasimetric Embeddings with Low Distortion; Adapted from Corollary 2 in Mémoli et al. (2018)).** Let $M = (X, d)$ be a quasipseudometric space with treewidth $t$, and $n = |X|$. Then $M$ admits an embedding into a convex combination (i.e., scaled mixture) of $\mathcal{O}(n^2)$ quasipartitions with distortion $\mathcal{O}(t \log^2 n)$.

*Proof of Lemma C.5.* The distortion bound is proved in Corollary 2 in (Mémoli et al., 2018), which states that any quasipseudometric space with $n$ elements and $t$ treewidth admits an embedding into a convex combination of quasipartitions with distortion $\mathcal{O}(t \log^2 n)$.

To see that $n^2$ quasipartitions suffice, we scrutinize their construction of quasipartitions in Algorithm 2 of (Mémoli et al., 2018), reproduced below as Algorithm 1.

---

**Algorithm 1** Random quasipartition of a bounded treewidth graph. Algorithm 2 of (Mémoli et al., 2018).

---

**Input:** A digraph $G$ of treewidth $t$, a hierarchical tree of separators of $G$ $(H, f)$ with width $t$, and $r > 0$.
**Output:** A random $r$-bounded quasipartition $R$.

   **Initialization:** Set $G^* = G$, $H^* = H$ and $R = E(G)$. Perform the following recursive algorithm on $G^*$ and $H^*$.
**Step 1.** Pick $z \in [0, r/2]$ uniformly at random.
**Step 2.** If $|V(G^*)| \leq 1$, terminate the current recursive call. Otherwise pick the set of vertices $K = G^*$. Let $H_1, \ldots, H_m$ be the sub-trees of $H^*$ below $\mathsf{root}(H^*)$ that are hierarchical trees of separators of $C_1, \ldots, C_m$ respectively.
**Step 3.** For all $(u, v) \in E(G^*)$ remove $(u, v)$ from $R$ if one of the following holds:
    (a) $d_G(u, x) > z$ and $d_G(v, x) \leq z$ for some vertex $x \in K$.
    (b) $d_G(x, v) > z$ and $d_G(x, u) \leq z$ for some vertex $x \in K$.
**Step 4.** For all $i \in \{1, \ldots, m\}$ perform a recursive call of Steps 2-4 setting $G^* = G^*[C_i]$ and $H^* = H_i$.
**Step 5.** Once all branches of the recursive terminate, enforce transitivity on $R$: For all $u, v, w \in V(G)$ if $(u, v) \in R$ and $(v, w) \in R$, add $(u, w)$ to $R$.

---

Many concepts used in Algorithm 1 are not relevant for our purpose (e.g., $r$-bounded quasipartition). Importantly, we observe that for a given quasimetric space, the produced quasipartition is entirely determined by the random choice of $z$ in Step 1, which is only used to compare with distance values between node pairs. Note that there are $n^2$ node pairs, whose minimum distance is exactly 0 (i.e., distance from a node to itself). Since $z \geq 0$, there are at most $n^2$ choices of $z$ that lead to at most $n^2$ different quasipartitions, for all possible values of $r$.

The construction used to prove Corollary 2 of (Mémoli et al., 2018) uses exactly quasipartitions given by this algorithm. Therefore, the lemma is proved. □

Lemma C.5 essentially proves the first half of Thm. C.4. Before proving the full Thm. C.4, we restate the following result from (Hiraguchi, 1951), which gives us a bound on how many total orders are needed to represent a general partial order (i.e., quasipartition).

**Theorem C.6 (Hiraguchi's Theorem (Hiraguchi, 1951; Bogart, 1973)).** Let $(X, P)$ be a partially ordered set such that $|X| \geq 4$. Then there exists a mapping $f \colon X \to \mathbb{R}^{\lfloor |X|/2 \rfloor}$ such that

$$\forall x, y \in X, \qquad xPy \iff f(x) \leq f(y) \text{ coordinate-wise} . \tag{173}$$

*Proof of Thm. C.4.* It immediately follows from Lemma C.5 and Thm. C.6 that any quasimetric space with $n$ elements and treewidth $t$ admits an embedding with distortion $\mathcal{O}(t \log^2 n)$ into a convex combination of $n^2$ quasipartitions, each represented with an intersection of $\mathcal{O}(n)$ total orders.

Because the PQE class has concentration property, for any finite quasimetric space, we can simply select a PQE that is close enough to the desired convex combination of $n^2$ quasipartitions, to obtain distortion $\mathcal{O}(t \log^2 n)$. Since each Poisson process in PQE takes a constant number of latent dimensions, we can have such a PQE with $\mathcal{O}(n^3)$-dimensional latents and $n^2$ quasipartition distributions.

It remains only to prove that we can compute such required latents using the described architecture.

Consider any $x \in \mathcal{X} \subset \mathbb{R}^d$. Since $\mathcal{X}$ is finite, we can always find direction $u_x \in \mathbb{R}^d$ such that $\forall y \in \mathcal{X} \setminus \{x\}, y^\mathsf{T} u_x \neq x^\mathsf{T} u_x$. That is, $x$ has a unique projection onto $u_x$. Therefore, we can have $c, b_+, b_- \in \mathbb{R}$ such that

$$c \cdot u_x^\mathsf{T} x + b_+ = 1 \tag{174}$$

$$-c \cdot u_x^\mathsf{T} x + b_- = 1, \tag{175}$$

but for $y \in \mathcal{X} \setminus \{x\}$, we have, for some $a > 0$, either

$$c \cdot u_x^\mathsf{T} y + b_+ = -a \tag{176}$$

$$-c \cdot u_x^\mathsf{T} y + b_- = a + 2, \tag{177}$$

or

$$c \cdot u_x^\mathsf{T} y + b_+ = a + 2 \tag{178}$$

$$-c \cdot u_x^\mathsf{T} y + b_- = -a. \tag{179}$$

Then, consider computing two of the first layer features as, on input $z$,

$$[\text{ReLU}(c \cdot u_x^\mathsf{T} z + b_+) \quad \text{ReLU}(-c \cdot u_x^\mathsf{T} z + b_-)], \tag{180}$$

which, if $z = x$, is $[1, 1]$; if $z \neq x$, is either $[0, 2 + a]$ or $[2 + a, 0]$, for some $a > 0$.

Then, one of the second layer features may sum these two features and threshold it properly would single out $x$, i.e., activate only when input is $x$.

After doing this for all $x \in \mathcal{X}$, we obtain an $n$-dimensional second layer feature space that is just one-hot features.

The third layer can then just be a simple embedding look up, able to represent any embedding, including the one allowing a PQE to have distortion $\mathcal{O}(t \log n)$, as described above.

Because quasimetric embeddings naturally have violation 1, this concludes the proof.  □

### C.3.3   PROOF OF THM. 5.2: DISTORTION AND VIOLATION OF PQES

*Proof of Thm. 5.2.* Lemma C.3 and Thm. C.4 imply the result. To see that polynomial width is sufficient, note that the hidden width are polynomial by Thm. C.4, and that the embedding dimensions needed to represent each of the $\mathcal{O}(n^3)$ Poisson processes is constant 1 in both PQE-LH and PQE-GG. Hence the latent space is also polynomial. This concludes the result.  □

### C.3.4   DISCUSSIONS

**Dependency on** $\log n$**.**   $\log n$ dependency frequently occurs in distortion results. Perhaps the most well-known ones are Bourgain's Embedding Theorem (Bourgain, 1985) and the Johnson-Lindenstrauss Lemma (Johnson & Lindenstrauss, 1984), which concern *metric* embeddings into Euclidean spaces.

**Dependency on treewidth** $t$**.**   Treewidth $t$ here works as a complexity measure of the quasimetric. We will use a simple example to illustrate why low-treewidth is easy. Consider the extreme case where the quasimetric is the shortest-path distance on a tree, whose each edge is converted into two opposing directed ones and assigned arbitrary non-negative weights. Such a quasimetric space has treewidth 1 (see Defn. 2.2). On a tree,

1. the shortest path between two points is fixed, regardless of the weights assigned,
2. for each internal node $u$ and one of its child $c$, the followings are quasipartitions:

$$d'_{01}(x, y) \triangleq \mathbf{1}_{\text{shortest path from } x \text{ to } y \text{ passes } (u, c)}$$

$$d''_{01}(x, y) \triangleq \mathbf{1}_{\text{shortest path from } x \text{ to } y \text{ passes } (c, u)}.$$

Hence it can be *exactly* represented as a convex combination of quasipartitions. However, both of observations becomes false when the graph structure becomes more complex (higher treewidth) and the shortest paths can are less well represented as tree paths of the tree composition.

**Comparison with unconstrained MLPs.**   Thm. C.4 requires a poly-width encoder to achieve low distortion. This is comparable with deep unconstrained MLPs trained in NTK regime, which can reach 0 training error (distortion 1 on training set) in the limit but also requires polynomial width (Arora et al., 2019).

**Quasipseudometrics and infinite distances.**   Thm. C.4 relies on our assumptions that $(\mathcal{X}, d)$ is not a quasipseudometric space and has all finite distances. In fact, if we allow a PQE to have infinite convex combination weights, it can readily represent quasipseudometric spaces with infinite distances.

Additionally, PQE can still well approximate the quasimetric space with infinities replaced with any sufficiently large finite value (e.g., larger than the maximum finite distance). Thus, this limit is generally not important in practice (e.g., learning $\gamma$-discounted distances), where a large value and infinity are usually not treated much differently.

**Optimizing quasimetric embeddings.**   From Thm. C.4, we know that optimizing PQEs over the training set $S$ w.r.t. distortion achieves low distortion (and optimal violation by definition). While directly optimizing distortion (or error on log distance or distance ratios, equivalently) seems a valid choice, such objectives do not always train stably in practice, with possible infinities and zeros. Often more stable losses are used, such as MSE over raw distances or $\gamma$-discounted distances $\gamma^d$, for $\gamma \in (0, 1)$. These objectives do not directly relate to distortion, except for some elementary loose bounds. To better theoretically characterize their behavior, an alternative approach with an average-case analysis might be necessary.

### C.4   IMPLEMENTATION OF POISSON QUASIMETRIC EMBEDDINGS (PQES)

Sec. 5.2 mentioned a couple implementation techniques for PQEs. In this section, we present them in full details.

#### C.4.1   NORMALIZED MEASURES

Consider a PQE whose each of $j$ expected quasipartitions is defined via $k$ Poisson processes, with set parametrizations $u \to A_{i,j}(u), i \in [h], j \in [k]$. To be robust to the choice of $k$, we instead use the normalized set parametrizations $A'_{i,j}$:

$$A'_{i,j}(u) \triangleq A_{i,j}(u)/k, \qquad i \in [h], j \in [k]. \tag{181}$$

This does not change the PQE's concentration property (Defn. C.2) or its theoretical guarantees (e.g., Thms. 5.2 and C.4).

#### C.4.2   OUTPUTTING $\gamma$-DISCOUNTED DISTANCES

Recall the PQE quasimetric formulation in Eq. (13), for $\alpha_i \geq 0$, and encoder $f \colon \mathcal{X} \to \mathbb{R}^d$:

$$\hat{d}(x, y) \triangleq \sum_i \alpha_i \left( 1 - \prod_j \mathbb{P}\left[ \mathrm{Pois}(\mu_{i,j}(A_{i,j}(f(x)) \setminus A_{i,j}(f(y)))) \leq \mathrm{Pois}(\mu_{i,j}(A_{i,j}(f(y)) \setminus A_{i,j}(f(x)))) \right] \right).$$
$$\tag{13}$$

With discount factor $\gamma \in (0, 1)$, we can write the $\gamma$-discounted PQE distance as

$$\gamma^{\hat{d}(x,y)} = \prod_i (\underbrace{\gamma^{\alpha_i}}_{\text{a scalar that can take value in any } (0,1)})^{1 - \prod_j \mathbb{P}[\mathrm{Pois}(\mu_{i,j}(A_{i,j}(f(x)) \setminus A_{i,j}(f(y)))) \leq \mathrm{Pois}(\mu_{i,j}(A_{i,j}(f(y)) \setminus A_{i,j}(f(x))))]}. \tag{182}$$

Therefore, instead of learning $\alpha_i \in [0, \infty)$, we can learn bases $\beta_i \in (0, 1)$ such and define the $\gamma$-discounted PQE distance as

$$\gamma^{\hat{d}(x,y)} \triangleq \prod_i \beta_i^{1 - \prod_j \mathbb{P}[\mathrm{Pois}(\mu_{i,j}(A_{i,j}(f(x)) \setminus A_{i,j}(f(y)))) \leq \mathrm{Pois}(\mu_{i,j}(A_{i,j}(f(y)) \setminus A_{i,j}(f(x))))]}. \tag{183}$$

These bases $\beta_i \in (0, 1)$ can be parametrized via a sigmoid transform. Consider quasimetric learning w.r.t. errors on $\gamma$-discounted distances (e.g., MSE). Unlike the parametrization with directly learning the convex combination weights $\alpha_i$'s, such a parametrization (that learns the bases $\beta_i$'s) does not explicitly include $\gamma$ and thus can potentially be more stable for a wider range of $\gamma$ choices.

**Initialization.**   Consider learning bases $\beta_i$'s via a sigmoid transform: learning $b_i$ and defining $\beta_i \triangleq \sigma(b_i)$. We must take care in initializing these $b_i$'s so that $\sigma(b_i)$'s are not too close to 0 or 1, since we take a product of powers with these bases. To be robust to different $h$ numbers of quasipartition distributions, we initialize the each $b_i$ to be from the uniform distribution

$$\mathcal{U}[\sigma^{-1}(0.5^{2/h}), \sigma^{-1}(0.75^{2/h})], \tag{184}$$

which means that, at initialization,

$$\prod_{i\in[h]} \beta_i^{0.5} = \prod_{i\in[h]} \sigma(b_i)^{0.5} \in [0.5, 0.75], \tag{185}$$

providing a good range of initial outputs, assuming that the exponents (expected outputs of quasipartition distributions) are close to $0.5$. Alternatively, $b_i$'s maybe parametrized by a deep linear network, a similar initialization is employed. See Appendix C.4.3 below for details.

### C.4.3 LEARNING LINEAR/CONVEX COMBINATIONS WITH DEEP LINEAR NETWORKS

Deep linear networks have the same expressive power as regular linear models, but enjoy many empirical and theoretical benefits in optimization (Saxe et al., 2013; Pennington et al., 2018; Huh et al., 2021). Specifically, instead of directly learning a matrix $\in \mathbb{R}^{m\times n}$, a deep linear network (with bias) of $l$ layers learns a sequence of matrices

$$M_1 \in \mathbb{R}^{m_1 \times n} \tag{186}$$

$$M_2 \in \mathbb{R}^{m_2 \times m_1} \tag{187}$$

$$\vdots \quad \vdots \tag{188}$$

$$M_{l-1} \in \mathbb{R}^{m_{l-1} \times m_{l-2}} \tag{189}$$

$$M_l \in \mathbb{R}^{m \times m_{l-1}} \tag{190}$$

$$B \in \mathbb{R}^{m \times n}, \tag{191}$$

where the linear matrix can be obtained with

$$M_l \, M_{l-1} \dots M_2 \, M_1 + B, \tag{192}$$

and we require

$$\min(m_1, m_2, \dots, m_{l-1}) \geq \min(m, n). \tag{193}$$

In our case, the convex combination weights for the quasipartition distributions often need to be large, in order to represent large quasimetric distances; in Poisson process mean measures with learnable scales (e.g., the Gaussian-based measure described in Appendix C.2), the scales may also need to be large to approximate particular quasipartitions (see Appendix C.3.1).

Therefore, we choose to use deep linear networks to optimize these parameters. In particular,

- **For the convex combination weights for $h$ quasipartition distributions,**
  - When learning the convex combination weights $\{\alpha_i\}_{i\in[h]}$, we use a deep linear network to parametrize a matrix $\in \mathbb{R}^{1\times h}$ (i.e., a linear map from $\mathbb{R}^h$ to $\mathbb{R}$), which is then viewed as a vector $\in \mathbb{R}^h$ and applied an element-wise square transform $a \to a^2$ to obtain non-negative weights $\alpha \in [0, \infty)^h$;
  - When learning the bases for discounted quasimetric distances $\beta_i$'s (see Appendix C.4.2), we use a deep linear network to parametrize a matrix $\in \mathbb{R}^{h\times 1}$, which is then viewed as a vector $\in \mathbb{R}^h$ and applied an element-wise sigmoid transform $a \to \sigma(a)$ to obtain bases $\beta \in (0, 1)^h$.

    Note that here we parametrize a matrix $\in \mathbb{R}^{h\times 1}$ rather than $\mathbb{R}^{1\times h}$ as above for $\alpha_i$'s. The reason for this choice is entirely specific to the initialization scheme we use (i.e., (fully-connected layer weight matrix initialization, as discussed below). Here the interpretation of a linear map is no longer true. If we use $\mathbb{R}^{1\times h}$, the initialization method would lead to the entries distributed with variance roughly $1/n$, which only makes sense if they are then added together. Therefore, we use $\mathbb{R}^{h\times 1}$, which would lead to constant variance.

- **For scales of the Poisson process mean measure, such as PQE-GG,** we consider a slightly different strategy.

  Consider a PQE formulation with $h \times k$ independent Poisson processes, from which we form $h$ quasipartition distributions, each from $k$ total orders parametrized by $k$ Poisson processes. The Poisson processes are defined on sets

$$\{A_{i,j}\}_{i\in[h],j\in[k]}, \tag{194}$$

  use mean measures

$$\{\mu_{i,j}\}_{i\in[h],j\in[k]}, \tag{195}$$

and set parametrizations

$$\{u \to A_{i,j}(u)\}_{i \in [h], j \in [k]}, \tag{196}$$

to compute quantities

$$\mu_{i,j}(A_{i,j}(u) \setminus A_{i,j}(v)) \qquad \text{for } u \in \mathbb{R}^d, v \in \mathbb{R}^d, i \in [h], j \in [k]. \tag{197}$$

**Scaling each mean measure independently.** Essentially, adding learnable scales (of mean measures) $w \in [0, \infty)^{h \times k}$ (or, equivalently, $\{w_{i,j} \in [0, \infty)\}_{i,j}$) gives a scaled set of measures

$$\{w_{i,j} \cdot \mu_{i,j}\}_{i \in [h], j \in [k]}. \tag{198}$$

This means that the quantities in Eq. (197) becomes respectively scaled as

$$w_{i,j} \cdot \mu_{i,j}(A_{i,j}(u) \setminus A_{i,j}(v)) \qquad \text{for } u \in \mathbb{R}^d, v \in \mathbb{R}^d, i \in [h], j \in [k]. \tag{199}$$

**Convex combinations of *all* measures.** However, we can be more flexible here, and allow not just scaling each measure independently, but also *convex combinations of all measures*. Instead of having $w$ as a collection of $h \times k$ scalar numbers $\in [0, \infty)$, we have a collection of $(h \times k)$ vectors each having length $(h \times k)$ (or $h \times k$-shape tensors)

$$\{w_{i,j} \in [0, \infty)^{h \times k}\}_{i \in [h], j \in [k]}, \tag{200}$$

and have the quantities in Eq. (197) respectively scaled and combined as

$$\sum_{i',j'} w_{i,j,i',j'} \cdot \mu_{i',j'}(Ai', j'(u) \setminus A_{i',j'}(v)) \qquad \text{for } u \in \mathbb{R}^d, v \in \mathbb{R}^d, i \in [h], j \in [k]. \tag{201}$$

Note that these still are valid Poisson processes for a PQE. Specifically, the new Poisson processes now all use the same set parametrization (as the collection of original ones), with different measures (as different weighted combinations of the original measures). This generalizes the case where each mean measure is scaled independently (as $w$ can be diagonal).

Therefore, we will apply this more general strategy using **convex combinations of *all* measures**.

Similarly to learning the convex combination weights of quasipartition distributions, we collapse a deep linear network into a tensor $\in \mathbb{R}^{h \times k \times h \times k}$, and apply an element-wise square $a \to a^2$, result of which is used as the convex combination weights $w$ to ensure non-negativity.

**Initialization.** For initializing the matrices $(M_1, M_2, \ldots, M_l)$ of a deep linear network (Eq. (190)), we use the standard weight matrix initialization of fully-connected layers in PyTorch (Paszke et al., 2019). The bias matrix $B$ (Eq. (191)) is initialized to all zeros.

When used for learning the bases for discounted quasimetric distances $\beta_i$'s (as described in Appendix C.4.2), we have a deep linear network parametrizing a matrix $\in \mathbb{R}^{h \times 1}$, initialized in the same way as above (including initializing $B$ as all zeros). Consider the matrix up to before the last one:

$$M^* \triangleq M_{l-1} \cdot M_2 \, M_1 \in \mathbb{R}^{m_{l-1} \times 1}. \tag{202}$$

$M^*$ is essentially a projection to be applied on each row of the last matrix $M_l \in \mathbb{R}^{h \times m_{l-1}}$, to obtain $b_i$ (which is then used to obtain bases $\beta_i \triangleq \sigma(b_i)$). Therefore, we simply rescale the $M^*$ subspace for each row of $M_l$ and keep the orthogonal space intact, such that the projections would be distributed according to the distribution specified in Eq. (184):

$$\mathcal{U}[\sigma^{-1}(0.5^{2/h}), \sigma^{-1}(0.75^{2/h})], , \tag{184}$$

which has good initial value properties, as shown in Appendix C.4.2.

### C.4.4 CHOOSING $h$ THE NUMBER OF QUASIPARTITION DISTRIBUTIONS AND $k$ THE NUMBER OF POISSON PROCESSES FOR EACH QUASIPARTITION DISTRIBUTION

A PQE (class) is defined with $h \times k$ independent Poisson processes with means $\{\mu_{i,j}\}_{i \in [h], j \in [k]}$ along with $h \times k$ set parametrizations $\{A_{i,j}\}_{i \in [h], j \in [k]}$. For $k$ pairs of means and set parametrizations, we obtain a random quasipartition. A mixture (convex combination) of the resulting $h$ random quasipartitions gives the quasimetric. The choices of $\mu$ and $A$ are flexible. In this work we explore PQE-LH and PQE-GG as two options, both using essentially the same measure and parametrization across all $i, j$ (up to individual learnable scales). These two instantiations both perform well empirically. In this section we aim to provide some intuition on choosing these two hyperparameters $h$ and $k$.

$h$ **the Number of Quasipartition Distributions** Theoretical result Thm. C.4 suggest thats, for a quasimetric space with $n$ elements, $n^2$ quasipartition distributions suffice to learn a low distortion

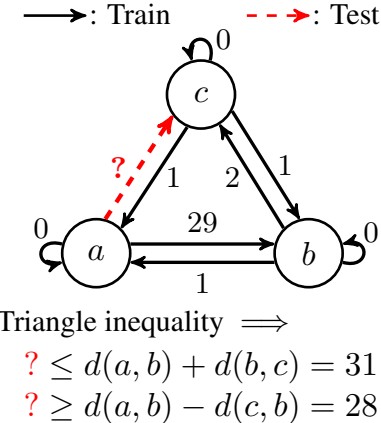

$$? \le d(a,b) + d(b,c) = 31$$
$$? \ge d(a,b) - d(c,b) = 28$$

Figure 9: The 3-element quasimetric space, and the training pairs.Training set contains all pairs except for $(a,c)$. Arrows show quasimetric distances (rather than edge weights of some graph).

embedding. Since this is a worst-case result, the practical scenario may require much fewer quasipartitions. For instance, Appendix C.3.4 shows that $\mathcal{O}(n)$ quasipartitions is sufficient for any quasimetric space with a tree structure. In our experiments, $h \in [8, 128]$ quasipartition distributions are used.

$k$ **the Number of Poisson Processes for Each Quasipartition Distribution (Random Partial Order)** It is well-known that such intersection of sufficiently many total orders can represent *any* partial order (Trotter, 1995; Hiraguchi, 1951). This idea is equivalent with the dominance drawing dimension of directed graphs (Ortali & Tollis, 2019), which concerns an order embedding of the vertices to preserve the poset specified by the reachability relation. In this graph theoretical view, several results are known. (Felsner et al., 2010) prove that planar graphs have at most 8 dimension. (Ortali & Tollis, 2019) show that the dimension of any graph with $n$ vertices is at most $\min(w_P, \frac{n}{2})$, where $w_P$ the maximum size of a set of incomparable vertices. A simpler and more fundamental result can be traced to Hiraguchi from 1951:

**Theorem C.6 (Hiraguchi's Theorem (Hiraguchi, 1951; Bogart, 1973)).** Let $(X, P)$ be a partially ordered set such that $|X| \ge 4$. Then there exists a mapping $f \colon X \to \mathbb{R}^{\lfloor |X|/2 \rfloor}$ such that
$$\forall x, y \in X, \qquad xPy \iff f(x) \le f(y) \text{ coordinate-wise}. \tag{173}$$

Thm. C.6 states that $\frac{n}{2}$ dimensions generally suffice for any poset of size $n \ge 4$.

In our formulation, this means that using $k = \frac{n}{2}$ Poisson processes (giving $\frac{n}{2}$ random total orders) will be maximally expressive. In practice, this is likely unnecessary and sometimes impractical. In our experiments, we choose a small fixed number $k = 4$.

## D EXPERIMENT SETTINGS AND ADDITIONAL RESULTS

**Computation power.** All our experiments run on a single GPU and finish within 3 hours. GPUs we used include NVIDIA 1080, NVIDIA 2080 Ti, NVIDIA 3080 Ti, NVIDIA Titan Xp, NVIDIA Titan RTX, and NVIDIA Titan V.

### D.1 EXPERIMENTS FROM SEC. 3.2: A TOY EXAMPLE

In Sec. 3.2 and Fig. 2, we show experiment results on a simple 3-element quasimetric space.

**Quasimetric space.** The quasimetric space has 3 elements with one-hot features $\in \mathbb{R}^3$. Thequasimetric and training pairs are shown in Fig. 9.

**Unconstrained network.** The unconstrained network has architecture 6-128-128-32-1, with ReLU activations.

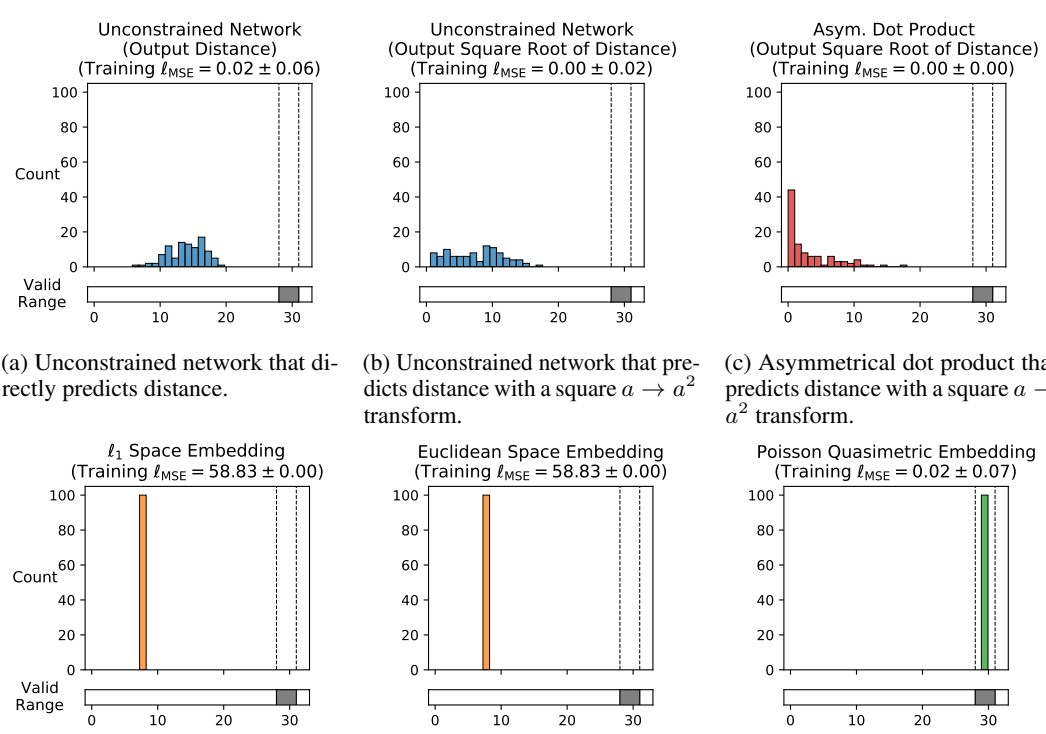

(a) Unconstrained network that directly predicts distance.

(b) Unconstrained network that predicts distance with a square $a \rightarrow a^2$ transform.

(c) Asymmetrical dot product that predicts distance with a square $a \rightarrow a^2$ transform.

(d) Metric embedding into an $\ell_1$ space.

(e) Metric embedding into an Euclidean space.

(f) Poisson Quasimetric Embedding specified in Appendix D.1.

Figure 10: Training different formulations to fit training pairs distances via MSE, and using them to predict on the test pair. Plots show distribution of the prediction over 100 runs. Standard deviations of the training error are shown.

**Metric embedding.** The embedding space is 32-dimensional, upon which corresponding metric is applied. The encoder network has architecture 6-128-128-32, with ReLU activations.

**Asymmetric dot products.** The embedding space is 32-dimensional. The two inputs are encoded with a *different* encoder of architecture 6-128-128-32, with ReLU activations. Then the dot product of the two 32-dimensional vector is taken, which parametrizes a distance estimate

**Poisson Quasimetric Embeddings.** The embedding space is 32-dimensional, which parametrizes 8 quasimetric distributions, each from 4 independent Poisson processes using (scaled) Lebesgue measure and half-lines. We use deep linear networks, as described in Appendix C.4.3. A deep linear network (without bias) of architecture 8-32-32-1 parametrizes the convex combination weights $\{\alpha_i\}_{i \in [8]}$. Another deep linear network (without bias) of architecture 32-64-64-32 parametrizes convex combination weights of the mean measures $d \in [0, \infty)^{32 \times 32}$. Note that these *do not* give many more effective parameters to PQEs as they are equivalent with simple linear transforms.

**Optimization.** All models are trained w.r.t. MSE on distances with the Adam optimizer (Kingma & Ba, 2014) with learning rate 0.0003 for 1000 iterations (without mini-batching since the training set has size 8).

**Additional results.** Results with additional formulations (together with the ones presented in Fig. 2) are shown in Fig. 10.

## D.2 EXPERIMENTS FROM SEC. 6: EXPERIMENTS

**Triangle inequality regularizer.** For methods that do not inherently respect triangle inequalities (e.g., unconstrained networks and asymmetrical dot products), we explore training with a regularizer that encourages following these inequalities. By sampling random triplets uniformly over the training set, the regularizer is formulated as,

$$\mathbb{E}_{x,y,z}\big[\max(0, \gamma^{\hat{d}(x,y)+\hat{d}(y,z)} - \gamma^{\hat{d}(x,z)})^2\big], \tag{203}$$

where the $\gamma$-discounted terms and the squared form allows easier balancing with the training loss, which, across all experiments, are MSEs on some $\gamma$-discounted distances.

**PQE settings.** Across all experiments of this section, when given an encoder architecture mapping input to an $\mathbb{R}^d$ latent space, we construct PQEs according to the following general recipe, to obtain the two PQEs settings used across all experiments: PQE-LH (PQE with Lebesgue measure and half-lines) and PQE-GG (PQE with Gaussian-based measure and Gaussian shapes, see see Appendix C.2):

- (Assuming $d$ is a multiple of 4,) We use $h \triangleq= d/4$ quasipartition distributions, each given by $k \triangleq 4$ Poisson processes;

- A deep linear network (see Appendix C.4.3), is used for parametrizing the convex combination weights $\alpha \in \mathbb{R}^{d/4}$ or the bases $\beta \in \mathbb{R}^{d/4}$ (see Appendix C.4.2), we follow the initialization and parametrization described in Appendix C.4.3, with hidden sizes $[n_{\text{hidden}}, n_{\text{hidden}}, n_{\text{hidden}}]$ (i.e., 4 matrices/layers), where $n_{\text{hidden}} \triangleq \max(64, 2^{1+\lceil \log_2(d/4) \rceil})$.

- For PQE-GG,

  - The learnable $\sigma^2_{\text{measure}} \in (0, \infty)^d$ (one for each Poisson Process) is achieved by optimizing the log variance, which is initialized as all zeros.
  - The Gaussian-based measures need learnable scales. We use a deep linear network to parametrize the $[0, \infty)^{d \times d}$ weights for the convex combinations of measures, as described in Appendix C.4.3. Similarly, it has hidden sizes $[n_{\text{hidden}}, n_{\text{hidden}}, n_{\text{hidden}}]$ (i.e., 4 matrices/layers), where $n_{\text{hidden}} \triangleq \max(64, 2^{1+\lceil \log_2 d \rceil})$.

Note that the PQEs add only a few extra effective parameters on top of the encoder ($d$ for PQE-LH, and $d + d^2$ for PQE-GG), as the deep linear networks do not add extra effective parameters.

**Mixed space metric embedding settings.** Across all experiments of this section, when given an encoder architecture mapping input to an $\mathbb{R}^d$ latent space, we construct the metric embedding into mixed space as follows:

- (Assuming $d$ is a multiple of 4,) We use (1) a $(d/2)$-dimensional Euclidean space (2) a $(d/4)$-dimensional $\ell_1$ space, and a $(d/4)$-dimensional spherical distance space (without scale).

- Additionally, we optimize three scalar values representing the log weights of the convex combination to mix these spaces.

**DeepNorm and WideNorm method overview and parameter count comparison with PQEs.** Both DeepNorm and WideNorm parametrize asymmetrical norms. When used to approximate quasimetrics, they are applied as $\hat{d}(x, y) \triangleq f_{\text{AsymNorm}}(f_{\text{Enc}}(x) - f_{\text{Enc}}(y))$, where $f_{\text{Enc}}$ is the encoder mapping from data space to an $\mathbb{R}^d$ latent space and $f_{\text{AsymNorm}}$ is either the DeepNorm or the WideNorm predictor on that latent space (Pitis et al., 2020).

- **DeepNorm** is a modification from Input Convex Neural Network (ICNN; Amos et al. (2017)), with restricted weight matrices and activation functions for positive homogeneity (a requirement of asymmetrical norms), and additional concave function for expressivity.

  For an input latent space of $\mathbb{R}^d$, consider an $n$-layer DeepNorm with width $w$ (i.e., ICNN output size) and the suggested intermediate MaxReLU activation and MaxMean final aggregation (see (Pitis et al., 2020) for details of these functions). This DeepNorm predictor $f_{\text{DeepNorm}}$ (on latent

space) has

$$\text{\#parmaters of } f_{\text{DeepNorm}} = \underbrace{n \times (d \times w)}_{U \text{ matrices from input to each layer}}$$

$$+ \underbrace{(n-1) \times w^2}_{W \text{ matrices between neighboring layer activations}}$$

$$+ \underbrace{n \times w}_{\text{intermediate MaxReLU activations}}$$

$$+ \underbrace{w \times (4+5)}_{\text{concave function (with 5 components) parameters}}$$

$$+ \underbrace{1}_{\text{final MaxMean aggregation}},$$

which is on the order of $\mathcal{O}(nw \max(d, w))$. In the common case where the hidden size $w$ is chosen to be on the same magnitude as $d$, this becomes $\mathcal{O}(nd^2)$.

- **WideNorm** is based on the observation that

$$x \to \|W \, \text{ReLU}(x :: -x)\|_2 \tag{204}$$

is an asymmetric norm when $W$ is non-negative, where :: denotes vector concatenation. Wide-Norm then learns many such norms each with a different $W$ matrix parameter, before (again) feeding the norm values into a concave function and aggregating them together with MaxMean. For an input latent space of $\mathbb{R}^d$, consider a WideNorm with $c$ such learned norms with $W$ matrices of shape $\mathbb{R}^{(2d) \times w}_{\geq 0}$. This WideNorm predictor $f_{\text{WideNorm}}$ (on latent space), has

$$\text{\#parmaters of } f_{\text{WideNorm}} = \underbrace{c \times (2d \times w)}_{W \text{ matrices}}$$

$$+ \underbrace{c \times (4+5)}_{\text{concave function (with 5 components) parameters}}$$

$$+ \underbrace{1}_{\text{final MaxMean aggregation}},$$

which is on the order of $\mathcal{O}(cdw)$. In the common case where both the number of components $c$ and the output size of each component (before applying the $l2$-norm) are chosen to be on the same magnitude as $d$, this becomes $\mathcal{O}(d^3)$.

For both DeepNorm and WideNorm, their parameter counts are much larger than the number of effective parameters of PQEs ($d$ for PQE-LH and $d + d^2$ for PQE-GG). For a $256$-dimensional latent space, this difference can be on the order of $10^6 \sim 10^7$.

**DeepNorm and WideNorm settings.** Across all experiments of this section, we evaluate 2 Deep-Norm settings and 3 WideNorm settings, all derived from the experiment setting of the original paper (Pitis et al., 2020). For both DeepNorm and WideNorm, we use MaxReLU activations, MaxMean aggregation, and concave function of 5 components. For DeepNorm, we use 3-layer networks with 2 different hidden sizes: 48 and 128 for the $48$-dimensional latent space in random directed graphs experiments, 512 and 128 for the $512$-dimensional latent space in the large-scale social graph experiments, 128 and 64 for the $128$-dimensional latent space in offline Q-learning experiments. For WideNorm, we components of size 32 and experiment with 3 different numbers of components: 32, 48, and 128.

**Error range.** Results are gathered across 5 random seeds, showing both averages and population standard deviations.

### D.2.1 RANDOM DIRECTED GRAPHS QUASIMETRIC LEARNING

**Graph generation.** The random graph generation is controlled by three parameters $d$, $\rho_{\text{un}}$ and $\rho_{\text{di}}$. $d$ is the dimension of the vertex features. $\rho_{\text{un}}$ specifies the fraction of pairs that should have at least one (directed) edge between them. $\rho_{\text{di}}$ specifies the fraction of *such* pairs that should *only* have one (directed) edge between them. Therefore, if $\rho_{\text{un}} = 1, \rho_{\text{di}} = 0$, we have a fully connected graph; if

$\rho_{\mathsf{un}} = 0.5, \rho_{\mathsf{di}} = 1$, we have a graph where half of the vertex pairs have exactly one (directed) edge between them, and the other half are not connected. For completeness, the exact generation procedure for a graph of $n$ vertices is the following:

1. randomly add $\rho_{\mathsf{un}} \cdot n^2$ undirected edges, each represented as two opposite directed edges;
2. optimize $\mathbb{R}^{n \times d}$ vertex feature matrix using Adam (Kingma & Ba, 2014) w.r.t. $\mathcal{L}_{\mathsf{align}}(\alpha = 2) + 0.3 \cdot \mathcal{L}_{\mathsf{uniform}}(t = 3)$ from (Wang & Isola, 2020), where each two node is considered a positive pair if they are connected;
3. randomly initialize a network $f$ of architecture $d$-4096-4096-4096-4096-1 with $\tanh$ activations;
4. for each connected vertex pair $(u, v)$, obtain $d_{u \to v} \triangleq f(\mathsf{feature}(u)) - f(\mathsf{feature}(v))$ and $d_{v \to u} = -d_{u \to v}$;
5. for each $(u, v)$ such that $d_{u \to v}$ is among the top $1 - \rho_{\mathsf{di}}/2$ of such values (which is guaranteed to not include both directions of the same pair due to symmetry of $d_{u \to v}$), make $v \to u$ the only directed edge between $u$ and $v$.

We experiment with three graphs of 300 vertices and 64-dimensional vertex features:

- **Fig. 11:** A graph generated with $\rho_{\mathsf{un}} = 0.15, \rho_{\mathsf{di}} = 0.85$;
- **Fig. 12:** A sparser graph generated with $\rho_{\mathsf{un}} = 0.05, \rho_{\mathsf{di}} = 0.85$;
- **Fig. 13:** A sparse graph with block structure by

  1. generating 10 small dense graphs of 30 vertices and 32-dimensional vertex features, using $\rho_{\mathsf{un}} = 0.18, \rho_{\mathsf{di}} = 0.15$,
  2. generating a sparse 10-vertex "supergraph" with 32-dimensional vertex features, using $\rho_{\mathsf{un}} = 0.22, \rho_{\mathsf{di}} = 0.925$,
  3. for each supergraph vertex
     (a) associating it with a different small graph,
     (b) for all vertices of the small graph, concatenate the supergraph vertex's feature to the existing feature, forming 64-dimensional vertex features for the small graph vertices,
     (c) picking a random representative vertex from the small graph,
  4. connecting all 10 representative vertices in the same way as their respective supergraph vertices are connected in the supergraph.

**Architecture.** All encoder based methods (PQEs, metric embeddings, dot products) use 64-128-128-128-48 network with ReLU activations, mapping 64-dimensional inputs to a 48-dimensional latent space. Unconstrained networks use a similar 128-128-128-128-48-1 network, mapping concatenated the 128-dimensional input to a scalar output.

**Data.** For each graph, we solve the groundtruth distance matrix and obtain $300^2$ pairs, from which we randomly sample the training set, and use the rest as the test set. We run on 5 training fractions evenly spaced on the logarithm scale, from 0.01 to 0.7.

**Training.** We use 2048 batch size with the Adam optimizer (Kingma & Ba, 2014), with learning rate decaying according to the cosine schedule without restarting (Loshchilov & Hutter, 2016) starting from $10^{-4}$ to 0 over 3000 epochs. All models are optimized w.r.t. MSE on the $\gamma$-discounted distances, with $\gamma = 0.9$. When running with the triangle inequality regularizer, $683 \approx 2048/3$ triplets are uniformly sampled at each iteration.

**Full results and ablation studies.** Figs. 11 to 13 show full results of all methods running on all three graphs. In Fig. 14, we perform ablation studies on the implementation techniques for PQEs mentioned in Appendix C.4: outputting discounted distance and deep linear networks. On the simple directed graphs such as the dense graph, the basic PQE-LH without theses techniques works really well, even surpassing the results with both techniques. However, on graphs with more complex structures (e.g., the sparse graph and the sparse graph with block structure), basic versions of PQE-LH and PQE-GG starts to perform badly and show large variance, while the versions with both techniques stably trains to the best results. Therefore, for robustness, we use both techniques in other experiments.

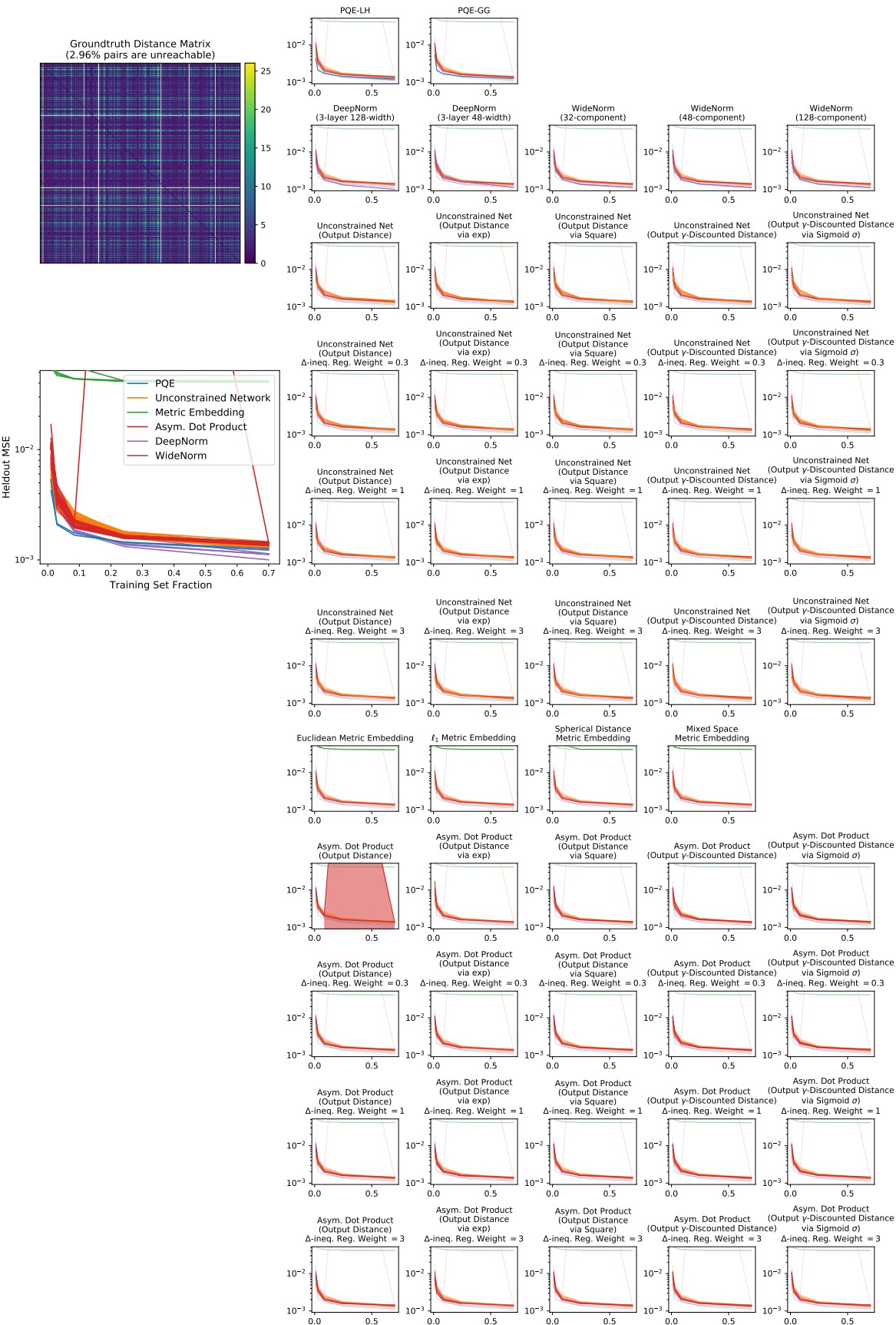

Figure 11: A dense graph. Individual plots on the right show standard deviations.

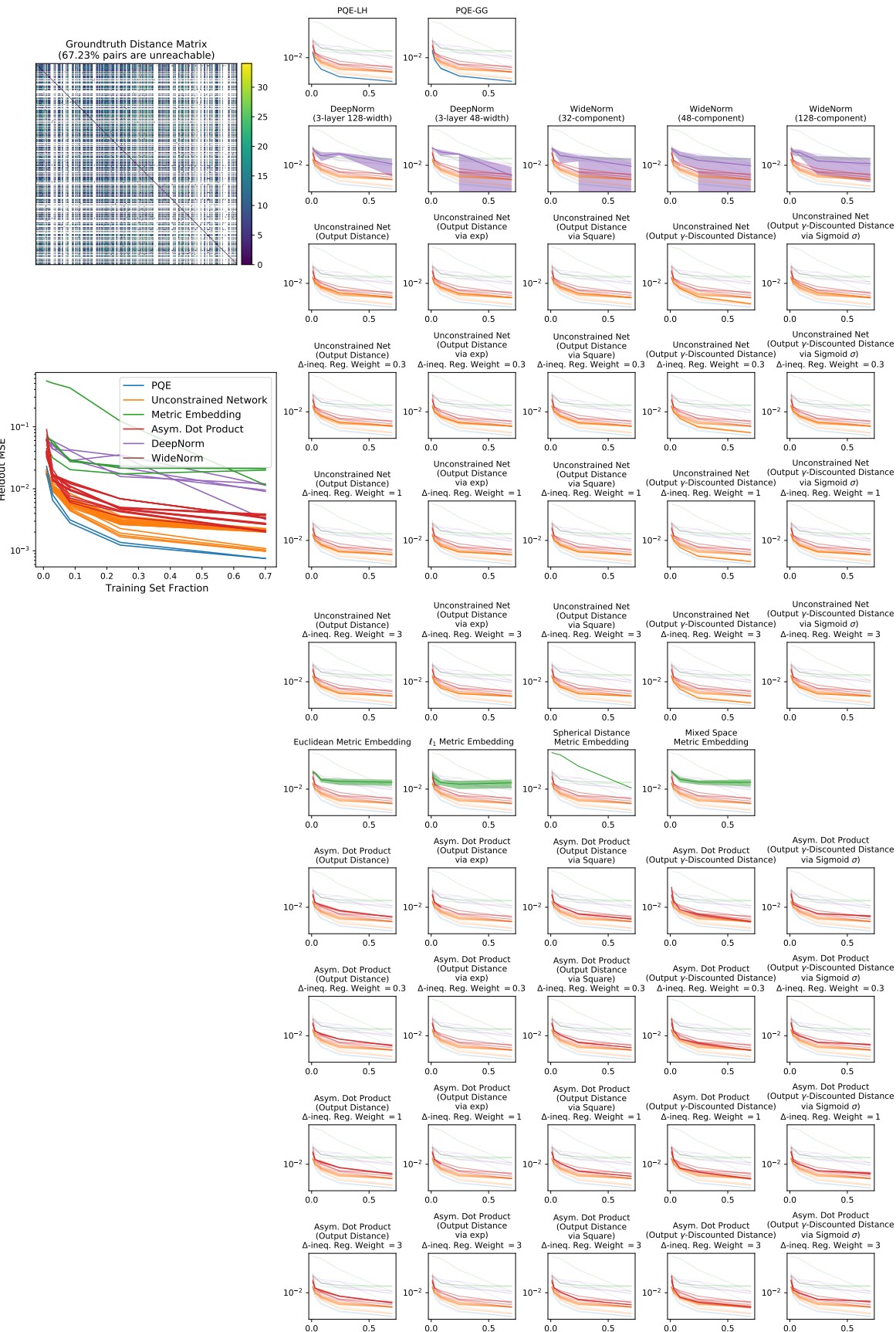

Figure 12: A sparse graph. Individual plots on the right show standard deviations.

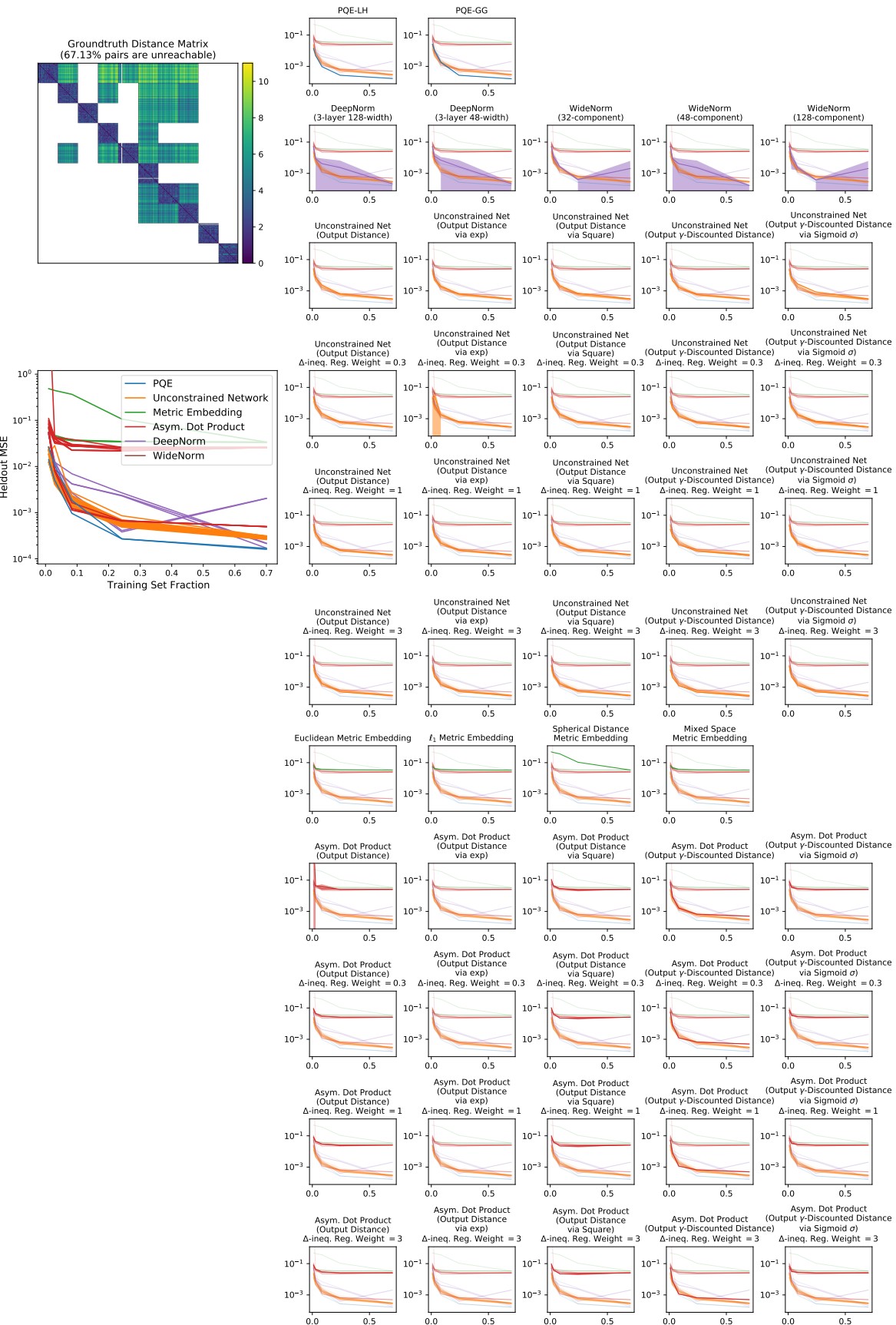

Figure 13: A sparse graph with block structure. Individual plots on the right show standard deviations.

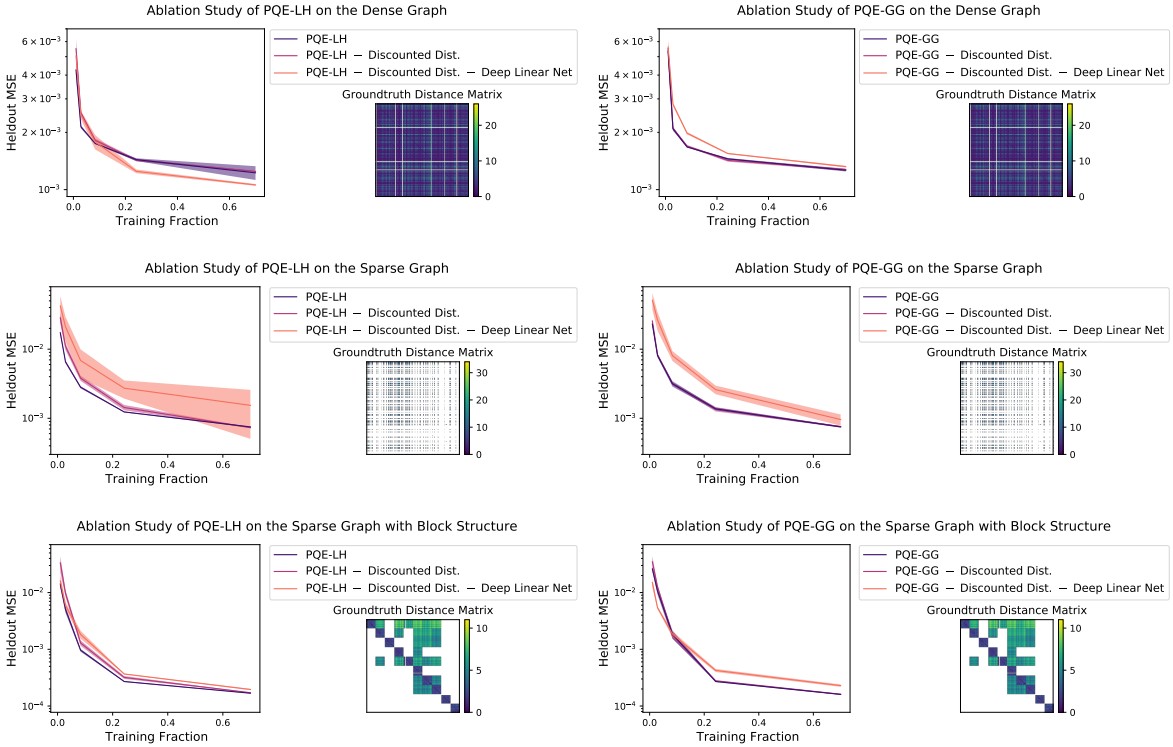

Figure 14: Ablation studies of PQE-LH and PQE-GG on three random graphs.

### D.2.2 LARGE-SCALE SOCIAL GRAPHS QUASIMETRIC LEARNING

**Data source.** We choose the Berkeley-StanfordWebGraph (Leskovec & Krevl, 2014) as the large-scale *directed* social graph, which consists of 685,230 pages as nodes, and 7,600,595 hyperlinks as directed edges. Additionally, we also use the Youtube social network (Leskovec & Krevl, 2014; Mislove et al., 2007) as a *undirected* social graph, which consists of 1,134,890 users as nodes, and 2,987,624 friendship relations as undirected edges. Both datasets are available from the SNAP website (Leskovec & Krevl, 2014) under the BSD license.

**Data processing.** For each graph, we use node2vec to obtain 128-dimensional node features (Grover & Leskovec, 2016). Since the graph is large, we use the landmark method (Rizi et al., 2018) to construct training and test sets. Specifically, we randomly choose 150 nodes, called landmarks, and compute the distances between these landmarks and all nodes. For directed graph, this means computing distances of both directions. From the obtained pairs and distances, we randomly sample 2,500,000 pairs to form the training set. Similarly, we form a test set of 150,000 from a disjoint set of 50 landmarks. For the undirected graph, we double the size of each set by reversing the pairs, since the distance is symmetrical.

**Architecture.** All encoder based methods (PQEs, metric embeddings, dot products) use 128-2048-2048-2048-512 network with ReLU activations and Batch Normalization (Ioffe & Szegedy, 2015) after each activation, mapping 128-dimensional inputs to a 512-dimensional latent space. Unconstrained networks use a similar 256-2048-2048-2048-512-1 network, mapping concatenated the 256-dimensional input to a scalar output.

**Training.** We use 1024 batch size with the Adam optimizer (Kingma & Ba, 2014), with learning rate decaying according to the cosine schedule without restarting (Loshchilov & Hutter, 2016) starting from $10^{-4}$ to 0 over 80 epochs. All models are optimized w.r.t. MSE on the $\gamma$-discounted distances, with $\gamma = 0.9$. When running with the triangle inequality regularizer, $342 \approx 1024/3$ triplets are uniformly sampled at each iteration.

| Method Family | Formulation | | MSE w.r.t. $\gamma$-discounted distances $(\times 10^{-3}) \downarrow$ | L1 Error when true $d < \infty \downarrow$ | Prediction $\hat{d}$ when true $d = \infty \uparrow$ |
|---|---|---|---|---|---|
| PQEs | PQE-LH | | $3.0427 \pm 0.1527$ | $1.6263 \pm 0.0550$ | $69.9424 \pm 0.4930$ |
| | PQE-GG | | $3.9085 \pm 0.1258$ | $1.8951 \pm 0.0336$ | $101.8240 \pm 10.3970$ |
| Unconstrained Nets (without Triangle Inequality Regularizer) | Output distance | $\hat{d}(x,y) \triangleq f(x,y)$ | $3.0862 \pm 0.0392$ | $2.1151 \pm 0.0241$ | $59.5243 \pm 0.3700$ |
| | Output distance via $\exp(\cdot)$ | $\hat{d}(x,y) \triangleq \exp(f(x,y))$ | $3.3541 \pm 0.1759$ | $1.0090 \times 10^{23} \pm 2.0179 \times 10^{23}$ | $5.3583 \times 10^{5} \pm 1.0582 \times 10^{6}$ |
| | Output distance via squaring $a \to a^2$ | $\hat{d}(x,y) \triangleq (f(x,y))^2$ | $4.5663 \pm 0.2294$ | $3.3459 \pm 0.2494$ | $68.2613 \pm 11.6061$ |
| | Output $\gamma$-discounted distance | $\gamma^{\hat{d}(x,y)} \triangleq f(x,y)$ | NaN $\pm$ NaN | NaN $\pm$ NaN | NaN $\pm$ NaN |
| | Output $\gamma$-discounted distance via sigmoid $\sigma(\cdot)$ | $\gamma^{\hat{d}(x,y)} \triangleq \sigma(f(x,y))$ | $3.1823 \pm 0.1133$ | $\infty \pm$ NaN | $65.8630 \pm 0.4287$ |
| Unconstrained Nets (Triangle Inequality Regularizer Weight $= 0.3$) | Output distance | $\hat{d}(x,y) \triangleq f(x,y)$ | $2.8128 \pm 0.0625$ | $2.2109 \pm 0.0341$ | $61.3709 \pm 0.3936$ |
| | Output distance via $\exp(\cdot)$ | $\hat{d}(x,y) \triangleq \exp(f(x,y))$ | $2.9344 \pm 0.0455$ | $\infty \pm$ NaN | $\infty \pm$ NaN |
| | Output distance via squaring $a \to a^2$ | $\hat{d}(x,y) \triangleq (f(x,y))^2$ | $4.9947 \pm 0.4198$ | $16.5445 \pm 29.3175$ | $58.9205 \pm 6.4216$ |
| | Output $\gamma$-discounted distance | $\gamma^{\hat{d}(x,y)} \triangleq f(x,y)$ | NaN $\pm$ NaN | NaN $\pm$ NaN | NaN $\pm$ NaN |
| | Output $\gamma$-discounted distance via sigmoid $\sigma(\cdot)$ | $\gamma^{\hat{d}(x,y)} \triangleq \sigma(f(x,y))$ | $2.9178 \pm 0.1351$ | $\infty \pm$ NaN | $\infty \pm$ NaN |
| Unconstrained Nets (Triangle Inequality Regularizer Weight $= 1$) | Output distance | $\hat{d}(x,y) \triangleq f(x,y)$ | $3.0481 \pm 0.1272$ | $2.3729 \pm 0.1378$ | $60.4040 \pm 0.1890$ |
| | Output distance via $\exp(\cdot)$ | $\hat{d}(x,y) \triangleq \exp(f(x,y))$ | $3.0161 \pm 0.0718$ | $\infty \pm$ NaN | $3.1289 \times 10^{16} \pm 6.2579 \times 10^{16}$ |
| | Output distance via squaring $a \to a^2$ | $\hat{d}(x,y) \triangleq (f(x,y))^2$ | $4.4921 \pm 0.3534$ | $3.6930 \pm 0.4896$ | $90.6206 \pm 66.5704$ |
| | Output $\gamma$-discounted distance | $\gamma^{\hat{d}(x,y)} \triangleq f(x,y)$ | $4.4046 \pm 0.5167$ | $2.7873 \pm 0.0770$ | $31.3195 \pm 0.9929$ |
| | Output $\gamma$-discounted distance via sigmoid $\sigma(\cdot)$ | $\gamma^{\hat{d}(x,y)} \triangleq \sigma(f(x,y))$ | $2.9314 \pm 0.1022$ | $2.2634 \pm 0.1147$ | $\infty \pm$ NaN |
| Unconstrained Nets (Triangle Inequality Regularizer Weight $= 3$) | Output distance | $\hat{d}(x,y) \triangleq f(x,y)$ | $5.2955 \pm 0.5279$ | $3.8060 \pm 0.2908$ | $58.1193 \pm 0.4383$ |
| | Output distance via $\exp(\cdot)$ | $\hat{d}(x,y) \triangleq \exp(f(x,y))$ | $3.5713 \pm 0.2002$ | $212.5421 \pm 416.9256$ | $\infty \pm$ NaN |
| | Output distance via squaring $a \to a^2$ | $\hat{d}(x,y) \triangleq (f(x,y))^2$ | $4.3745 \pm 0.3709$ | $2.9491 \pm 0.2228$ | $53.1119 \pm 5.5452$ |
| | Output $\gamma$-discounted distance | $\gamma^{\hat{d}(x,y)} \triangleq f(x,y)$ | $7.3416 \pm 0.6486$ | $3.5232 \pm 0.1352$ | $26.9200 \pm 0.4697$ |
| | Output $\gamma$-discounted distance via sigmoid $\sigma(\cdot)$ | $\gamma^{\hat{d}(x,y)} \triangleq \sigma(f(x,y))$ | $3.5818 \pm 0.3565$ | $\infty \pm$ NaN | $65.7709 \pm 0.8646$ |
| Asym. Dot Products (without Triangle Inequality Regularizer) | Output distance | $\hat{d}(x,y) \triangleq f(x)^\mathsf{T} g(y)$ | $3.1622 \times 10^{19} \pm$ NaN | $23.4270 \pm$ NaN | $0.1529 \pm$ NaN |
| | Output distance via $\exp(\cdot)$ | $\hat{d}(x,y) \triangleq \exp(f(x)^\mathsf{T} g(y))$ | NaN $\pm$ NaN | NaN $\pm$ NaN | NaN $\pm$ NaN |
| | Output distance via squaring $a \to a^2$ | $\hat{d}(x,y) \triangleq (f(x)^\mathsf{T} g(y))^2$ | $48.1056 \pm 0.0056$ | $2.5195 \times 10^{11} \pm 2.1751 \times 10^{11}$ | $2.6794 \times 10^{11} \pm 2.5398 \times 10^{11}$ |
| | Output $\gamma$-discounted distance | $\gamma^{\hat{d}(x,y)} \triangleq f(x)^\mathsf{T} g(y)$ | NaN $\pm$ NaN | NaN $\pm$ NaN | NaN $\pm$ NaN |
| | Output $\gamma$-discounted distance via sigmoid $\sigma(\cdot)$ | $\gamma^{\hat{d}(x,y)} \triangleq \sigma(f(x)^\mathsf{T} g(y))$ | $48.1073 \pm 0.0112$ | $\infty \pm$ NaN | $\infty \pm$ NaN |
| Asym. Dot Products (Triangle Inequality Regularizer Weight $= 0.3$) | Output distance | $\hat{d}(x,y) \triangleq f(x)^\mathsf{T} g(y)$ | NaN $\pm$ NaN | NaN $\pm$ NaN | NaN $\pm$ NaN |
| | Output distance via $\exp(\cdot)$ | $\hat{d}(x,y) \triangleq \exp(f(x)^\mathsf{T} g(y))$ | NaN $\pm$ NaN | NaN $\pm$ NaN | NaN $\pm$ NaN |
| | Output distance via squaring $a \to a^2$ | $\hat{d}(x,y) \triangleq (f(x)^\mathsf{T} g(y))^2$ | $48.1041 \pm 0.0035$ | $1.9498 \times 10^{11} \pm 7.9641 \times 10^{10}$ | $1.6049 \times 10^{11} \pm 3.7099 \times 10^{10}$ |
| | Output $\gamma$-discounted distance | $\gamma^{\hat{d}(x,y)} \triangleq f(x)^\mathsf{T} g(y)$ | NaN $\pm$ NaN | NaN $\pm$ NaN | NaN $\pm$ NaN |
| | Output $\gamma$-discounted distance via sigmoid $\sigma(\cdot)$ | $\gamma^{\hat{d}(x,y)} \triangleq \sigma(f(x)^\mathsf{T} g(y))$ | $48.1103 \pm 0.0110$ | $\infty \pm$ NaN | $\infty \pm$ NaN |
| Asym. Dot Products (Triangle Inequality Regularizer Weight $= 1$) | Output distance | $\hat{d}(x,y) \triangleq f(x)^\mathsf{T} g(y)$ | NaN $\pm$ NaN | NaN $\pm$ NaN | NaN $\pm$ NaN |
| | Output distance via $\exp(\cdot)$ | $\hat{d}(x,y) \triangleq \exp(f(x)^\mathsf{T} g(y))$ | NaN $\pm$ NaN | NaN $\pm$ NaN | NaN $\pm$ NaN |
| | Output distance via squaring $a \to a^2$ | $\hat{d}(x,y) \triangleq (f(x)^\mathsf{T} g(y))^2$ | $48.1021 \pm 0.0002$ | $2.2986 \times 10^{11} \pm 9.1970 \times 10^{10}$ | $2.5002 \times 10^{11} \pm 1.4464 \times 10^{11}$ |
| | Output $\gamma$-discounted distance | $\gamma^{\hat{d}(x,y)} \triangleq f(x)^\mathsf{T} g(y)$ | NaN $\pm$ NaN | NaN $\pm$ NaN | NaN $\pm$ NaN |
| | Output $\gamma$-discounted distance via sigmoid $\sigma(\cdot)$ | $\gamma^{\hat{d}(x,y)} \triangleq \sigma(f(x)^\mathsf{T} g(y))$ | $58.4894 \pm 23.2224$ | $\infty \pm$ NaN | $\infty \pm$ NaN |
| Asym. Dot Products (Triangle Inequality Regularizer Weight $= 3$) | Output distance | $\hat{d}(x,y) \triangleq f(x)^\mathsf{T} g(y)$ | NaN $\pm$ NaN | NaN $\pm$ NaN | NaN $\pm$ NaN |
| | Output distance via $\exp(\cdot)$ | $\hat{d}(x,y) \triangleq \exp(f(x)^\mathsf{T} g(y))$ | NaN $\pm$ NaN | NaN $\pm$ NaN | NaN $\pm$ NaN |
| | Output distance via squaring $a \to a^2$ | $\hat{d}(x,y) \triangleq (f(x)^\mathsf{T} g(y))^2$ | $48.1031 \pm 0.0020$ | $2.3522 \times 10^{11} \pm 2.6429 \times 10^{11}$ | $1.7025 \times 10^{11} \pm 1.0700 \times 10^{11}$ |
| | Output $\gamma$-discounted distance | $\gamma^{\hat{d}(x,y)} \triangleq f(x)^\mathsf{T} g(y)$ | NaN $\pm$ NaN | NaN $\pm$ NaN | NaN $\pm$ NaN |
| | Output $\gamma$-discounted distance via sigmoid $\sigma(\cdot)$ | $\gamma^{\hat{d}(x,y)} \triangleq \sigma(f(x)^\mathsf{T} g(y))$ | $48.3034 \pm 0.4485$ | $\infty \pm$ NaN | $\infty \pm$ NaN |
| Metric Embeddings | Euclidean space | $\hat{d}(x,y) \triangleq \|f(x) - f(y)\|_2$ | $17.5952 \pm 0.2667$ | $7.5399 \pm 0.0742$ | $53.8500 \pm 3.8430$ |
| | $\ell_1$ space | $\hat{d}(x,y) \triangleq \|f(x) - f(y)\|_1$ | $18.0521 \pm 0.3546$ | $7.1154 \pm 0.1835$ | $66.2507 \pm 3.3308$ |
| | Spherical distance space w/ learnable scale $\alpha$ | $\hat{d}(x,y) \triangleq \alpha \cdot \arccos(\frac{f(x)^\mathsf{T} f(y)}{\|f(x)\|_2 \|f(y)\|_2})$ | $19.2990 \pm 0.2032$ | $6.9545 \pm 0.0887$ | $32.1458 \pm 0.4562$ |
| | Mixing above three spaces w/ learnable weights | | $17.8312 \pm 0.3099$ | $7.3493 \pm 0.1086$ | $51.7481 \pm 3.6248$ |
| DeepNorms | 3-layer 128-width | | $7.0862 \pm 0.3170$ | $2.4498 \pm 0.0617$ | $111.2209 \pm 2.5045$ |
| | 3-layer 512-width | | $5.0715 \pm 0.1348$ | $2.0853 \pm 0.0633$ | $120.0452 \pm 4.3525$ |
| WideNorms | 32-component (each of size 32) | | $3.5328 \pm 0.2120$ | $1.7694 \pm 0.0213$ | $124.6580 \pm 2.8678$ |
| | 48-component (each of size 32) | | $3.6842 \pm 0.2385$ | $1.8081 \pm 0.0680$ | $122.6833 \pm 5.5026$ |
| | 128-component (each of size 32) | | $3.8125 \pm 0.2331$ | $1.8096 \pm 0.0765$ | $128.5427 \pm 5.1412$ |

Table 2: Quasimetric learning on the large-scale *directed* Berkeley-StanfordWebGraph.

| Method Family | Formulation | MSE w.r.t. $\gamma$-discounted distances $(\times 10^{-3})\downarrow$ | L1 Error when true $d < \infty \downarrow$ | Prediction $\hat{d}$ when true $d = \infty \uparrow$ |
|---|---|---|---|---|
| PQEs | PQE-LH | $2.4400 \pm 0.0695$ | $0.6480 \pm 0.0119$ | NaN $\pm$ NaN |
| | PQE-GG | $2.5895 \pm 0.0318$ | $0.6697 \pm 0.0042$ | NaN $\pm$ NaN |
| Unconstrained Nets (without Triangle Inequality Regularizer) | Output distance $\quad \hat{d}(x,y) \triangleq f(x,y)$ | $1.4883 \pm 0.0168$ | $0.5084 \pm 0.0029$ | NaN $\pm$ NaN |
| | Output distance via $\exp(\cdot)\quad \hat{d}(x,y) \triangleq \exp(f(x,y))$ | $1.5223 \pm 0.0160$ | $0.4910 \pm 0.0151$ | NaN $\pm$ NaN |
| | Output distance via squaring $a \to a^2 \quad \hat{d}(x,y) \triangleq (f(x,y))^2$ | $2.2955 \pm 1.1674$ | $0.6185 \pm 0.1409$ | NaN $\pm$ NaN |
| | Output $\gamma$-discounted distance $\quad \gamma^{\hat{d}(x,y)} \triangleq f(x,y)$ | $1.5069 \pm 0.0228$ | $0.4975 \pm 0.0211$ | NaN $\pm$ NaN |
| | Output $\gamma$-discounted distance via sigmoid $\sigma(\cdot)\quad \gamma^{\hat{d}(x,y)} \triangleq \sigma(f(x,y))$ | $1.4802 \pm 0.0197$ | $0.5082 \pm 0.0036$ | NaN $\pm$ NaN |
| Unconstrained Nets (Triangle Inequality Regularizer Weight $= 0.3$) | Output distance $\quad \hat{d}(x,y) \triangleq f(x,y)$ | $1.5009 \pm 0.0208$ | $0.5107 \pm 0.0032$ | NaN $\pm$ NaN |
| | Output distance via $\exp(\cdot)\quad \hat{d}(x,y) \triangleq \exp(f(x,y))$ | $1.5206 \pm 0.0444$ | $0.4935 \pm 0.0098$ | NaN $\pm$ NaN |
| | Output distance via squaring $a \to a^2 \quad \hat{d}(x,y) \triangleq (f(x,y))^2$ | $1.7398 \pm 0.3896$ | $0.5488 \pm 0.0600$ | NaN $\pm$ NaN |
| | Output $\gamma$-discounted distance $\quad \gamma^{\hat{d}(x,y)} \triangleq f(x,y)$ | $1.5005 \pm 0.0148$ | $0.4986 \pm 0.0121$ | NaN $\pm$ NaN |
| | Output $\gamma$-discounted distance via sigmoid $\sigma(\cdot)\quad \gamma^{\hat{d}(x,y)} \triangleq \sigma(f(x,y))$ | $1.4851 \pm 0.0168$ | $0.5089 \pm 0.0026$ | NaN $\pm$ NaN |
| Unconstrained Nets (Triangle Inequality Regularizer Weight $= 1$) | Output distance $\quad \hat{d}(x,y) \triangleq f(x,y)$ | $1.4999 \pm 0.0243$ | $0.5107 \pm 0.0046$ | NaN $\pm$ NaN |
| | Output distance via $\exp(\cdot)\quad \hat{d}(x,y) \triangleq \exp(f(x,y))$ | $1.5224 \pm 0.0376$ | $0.4948 \pm 0.0169$ | NaN $\pm$ NaN |
| | Output distance via squaring $a \to a^2 \quad \hat{d}(x,y) \triangleq (f(x,y))^2$ | $1.8875 \pm 0.5078$ | $0.5692 \pm 0.0683$ | NaN $\pm$ NaN |
| | Output $\gamma$-discounted distance $\quad \gamma^{\hat{d}(x,y)} \triangleq f(x,y)$ | $1.4769 \pm 0.0176$ | $0.4919 \pm 0.0128$ | NaN $\pm$ NaN |
| | Output $\gamma$-discounted distance via sigmoid $\sigma(\cdot)\quad \gamma^{\hat{d}(x,y)} \triangleq \sigma(f(x,y))$ | $1.4846 \pm 0.0115$ | $0.5088 \pm 0.0021$ | NaN $\pm$ NaN |
| Unconstrained Nets (Triangle Inequality Regularizer Weight $= 3$) | Output distance $\quad \hat{d}(x,y) \triangleq f(x,y)$ | $1.4939 \pm 0.0110$ | $0.5099 \pm 0.0018$ | NaN $\pm$ NaN |
| | Output distance via $\exp(\cdot)\quad \hat{d}(x,y) \triangleq \exp(f(x,y))$ | $1.5154 \pm 0.0389$ | $0.4871 \pm 0.0174$ | NaN $\pm$ NaN |
| | Output distance via squaring $a \to a^2 \quad \hat{d}(x,y) \triangleq (f(x,y))^2$ | $2.4747 \pm 1.0850$ | $0.6505 \pm 0.1357$ | NaN $\pm$ NaN |
| | Output $\gamma$-discounted distance $\quad \gamma^{\hat{d}(x,y)} \triangleq f(x,y)$ | $1.4915 \pm 0.0127$ | $0.4983 \pm 0.0160$ | NaN $\pm$ NaN |
| | Output $\gamma$-discounted distance via sigmoid $\sigma(\cdot)\quad \gamma^{\hat{d}(x,y)} \triangleq \sigma(f(x,y))$ | $1.4829 \pm 0.0153$ | $0.5084 \pm 0.0029$ | NaN $\pm$ NaN |
| Asym. Dot Products (without Triangle Inequality Regularizer) | Output distance $\quad \hat{d}(x,y) \triangleq f(x)^{\mathsf{T}}g(y)$ | $2633.7907 \pm$ NaN | $11.3879 \pm$ NaN | NaN $\pm$ NaN |
| | Output distance via $\exp(\cdot)\quad \hat{d}(x,y) \triangleq \exp(f(x)^{\mathsf{T}}g(y))$ | NaN $\pm$ NaN | NaN $\pm$ NaN | NaN $\pm$ NaN |
| | Output distance via squaring $a \to a^2 \quad \hat{d}(x,y) \triangleq (f(x)^{\mathsf{T}}g(y))^2$ | $339.1550 \pm 0.0022$ | $7.8948 \times 10^{11} \pm 7.4010 \times 10^{11}$ | NaN $\pm$ NaN |
| | Output $\gamma$-discounted distance $\quad \gamma^{\hat{d}(x,y)} \triangleq f(x)^{\mathsf{T}}g(y)$ | $2.6920 \pm 1.2655$ | $0.7062 \pm 0.2156$ | NaN $\pm$ NaN |
| | Output $\gamma$-discounted distance via sigmoid $\sigma(\cdot)\quad \gamma^{\hat{d}(x,y)} \triangleq \sigma(f(x)^{\mathsf{T}}g(y))$ | $182.2068 \pm 1.2382$ | $\infty \pm$ NaN | NaN $\pm$ NaN |
| Asym. Dot Products (Triangle Inequality Regularizer Weight $= 0.3$) | Output distance $\quad \hat{d}(x,y) \triangleq f(x)^{\mathsf{T}}g(y)$ | $9.9748 \times 10^5 \pm$ NaN | $8.1867 \pm$ NaN | NaN $\pm$ NaN |
| | Output distance via $\exp(\cdot)\quad \hat{d}(x,y) \triangleq \exp(f(x)^{\mathsf{T}}g(y))$ | NaN $\pm$ NaN | NaN $\pm$ NaN | NaN $\pm$ NaN |
| | Output distance via squaring $a \to a^2 \quad \hat{d}(x,y) \triangleq (f(x)^{\mathsf{T}}g(y))^2$ | $339.1560 \pm 0.0010$ | $6.8658 \times 10^{11} \pm 3.4985 \times 10^{11}$ | NaN $\pm$ NaN |
| | Output $\gamma$-discounted distance $\quad \gamma^{\hat{d}(x,y)} \triangleq f(x)^{\mathsf{T}}g(y)$ | NaN $\pm$ NaN | NaN $\pm$ NaN | NaN $\pm$ NaN |
| | Output $\gamma$-discounted distance via sigmoid $\sigma(\cdot)\quad \gamma^{\hat{d}(x,y)} \triangleq \sigma(f(x)^{\mathsf{T}}g(y))$ | $183.3337 \pm 1.0384$ | $\infty \pm$ NaN | NaN $\pm$ NaN |
| Asym. Dot Products (Triangle Inequality Regularizer Weight $= 1$) | Output distance $\quad \hat{d}(x,y) \triangleq f(x)^{\mathsf{T}}g(y)$ | NaN $\pm$ NaN | NaN $\pm$ NaN | NaN $\pm$ NaN |
| | Output distance via $\exp(\cdot)\quad \hat{d}(x,y) \triangleq \exp(f(x)^{\mathsf{T}}g(y))$ | NaN $\pm$ NaN | NaN $\pm$ NaN | NaN $\pm$ NaN |
| | Output distance via squaring $a \to a^2 \quad \hat{d}(x,y) \triangleq (f(x)^{\mathsf{T}}g(y))^2$ | $339.1552 \pm 0.0021$ | $7.4588 \times 10^{11} \pm 3.7277 \times 10^{11}$ | NaN $\pm$ NaN |
| | Output $\gamma$-discounted distance $\quad \gamma^{\hat{d}(x,y)} \triangleq f(x)^{\mathsf{T}}g(y)$ | NaN $\pm$ NaN | NaN $\pm$ NaN | NaN $\pm$ NaN |
| | Output $\gamma$-discounted distance via sigmoid $\sigma(\cdot)\quad \gamma^{\hat{d}(x,y)} \triangleq \sigma(f(x)^{\mathsf{T}}g(y))$ | $191.0928 \pm 9.7137$ | $\infty \pm$ NaN | NaN $\pm$ NaN |
| Asym. Dot Products (Triangle Inequality Regularizer Weight $= 3$) | Output distance $\quad \hat{d}(x,y) \triangleq f(x)^{\mathsf{T}}g(y)$ | NaN $\pm$ NaN | NaN $\pm$ NaN | NaN $\pm$ NaN |
| | Output distance via $\exp(\cdot)\quad \hat{d}(x,y) \triangleq \exp(f(x)^{\mathsf{T}}g(y))$ | NaN $\pm$ NaN | NaN $\pm$ NaN | NaN $\pm$ NaN |
| | Output distance via squaring $a \to a^2 \quad \hat{d}(x,y) \triangleq (f(x)^{\mathsf{T}}g(y))^2$ | $339.1556 \pm 0.0020$ | $9.0283 \times 10^{11} \pm 6.0203 \times 10^{11}$ | NaN $\pm$ NaN |
| | Output $\gamma$-discounted distance $\quad \gamma^{\hat{d}(x,y)} \triangleq f(x)^{\mathsf{T}}g(y)$ | NaN $\pm$ NaN | NaN $\pm$ NaN | NaN $\pm$ NaN |
| | Output $\gamma$-discounted distance via sigmoid $\sigma(\cdot)\quad \gamma^{\hat{d}(x,y)} \triangleq \sigma(f(x)^{\mathsf{T}}g(y))$ | $228.0300 \pm 37.0632$ | $\infty \pm$ NaN | NaN $\pm$ NaN |
| Metric Embeddings | Euclidean space $\quad \hat{d}(x,y) \triangleq \|f(x) - f(y)\|_2$ | $1.3131 \pm 0.0671$ | $0.4833 \pm 0.0128$ | NaN $\pm$ NaN |
| | $\ell_1$ space $\quad \hat{d}(x,y) \triangleq \|f(x) - f(y)\|_1$ | $3.5993 \pm 1.5986$ | $0.7787 \pm 0.1842$ | NaN $\pm$ NaN |
| | Spherical distance space w/ learnable scale $\alpha \quad \hat{d}(x,y) \triangleq \alpha \cdot \arccos(\frac{f(x)^{\mathsf{T}}f(y)}{\|f(x)\|_2\|f(y)\|_2})$ | $6.7731 \pm 0.1915$ | $1.0829 \pm 0.0177$ | NaN $\pm$ NaN |
| | Mixing above three spaces w/ learnable weights | $2.1014 \pm 0.0685$ | $0.5923 \pm 0.0109$ | NaN $\pm$ NaN |
| DeepNorms | 3-layer 128-width | $8.0192 \pm 0.2476$ | $1.1834 \pm 0.0213$ | NaN $\pm$ NaN |
| | 3-layer 512-width | $5.4366 \pm 0.0855$ | $0.9666 \pm 0.0072$ | NaN $\pm$ NaN |
| WideNorms | 32-component (each of size 32) | $3.0841 \pm 0.0667$ | $0.7272 \pm 0.0068$ | NaN $\pm$ NaN |
| | 48-component (each of size 32) | $3.0438 \pm 0.1322$ | $0.7247 \pm 0.0173$ | NaN $\pm$ NaN |
| | 128-component (each of size 32) | $2.9964 \pm 0.1363$ | $0.7173 \pm 0.0166$ | NaN $\pm$ NaN |

Table 3: Metric learning on the large-scale *undirected* Youtube graph. This graph does not have unreachable pairs so the last column is always NaN.

**Full results.**    Tables 2 and 3 show full results of distance learning on these two graphs. On the *directed* Berkeley-StanfordWebGraph, PQE-LH performs the best (w.r.t. discounted distance MSE). While PQE-GG has larger discounted distance MSE than some other baselines, it accurately predicts finite distances and outputs large values for unreachable pairs. On the *undirected* Youtube graph, perhaps as expected, metric embedding methods have an upper hand, with the best performing method being an Euclidean space embedding. Notably, DeepNorms and WideNorms do much worse than PQEs on this symmetric graph.

### D.2.3    OFFLINE Q-LEARNING

As shown in Proposition A.4 and Remark A.5, we know that a quasimetric is formed with the optimal goal-reaching plan costs in a MDP $\mathcal{M} = (\mathcal{S}, \mathcal{A}, \mathcal{R}, \mathcal{P}, \gamma)$ where each action has *unit cost* (i.e., negated reward). The quasimetric is defined on $\mathcal{X} \triangleq \mathcal{S} \cup (\mathcal{S} \times \mathcal{A})$.

Similarly, Tian et al. (2020) also make this observation and propose to optimize a distance function by Q-learning on a collected set of trajectories. The optimized distance function (i.e., Q-function) is then used with standard planning algorithms such as the Cross Entropy Method (CEM) (De Boer et al., 2005). The specific model they used is an unconstrained network $f\colon (s, a, s') \to \mathbb{R}$, outputting discounted distances (Q-values).

Due to the existing quasimetric structure, we explore using PQEs as the distance function formulation. We mostly follow the algorithm in Tian et al. (2020) except for the following minor differences:

- Tian et al. (2020) propose to sample half of the goal from future steps of the same trajectory, and half of the goal from similar states across the entire dataset, defined by a nearest neighbor search. For simplicity, in the latter case, we instead sample a random state across the entire dataset.
- In Tian et al. (2020), target goals are defined as single states, and the Q-learning formulation only uses quantities distances from state-action pairs $(s, a) \in \mathcal{S} \times \mathcal{A}$ to states $s'$: $\hat{d}((s, a), s')$.

  However, if we only train on $\hat{d}((s, a), s')$, quasimetric embeddings might not learn much about the distance to state-action pairs, or from states, because it may simply only assign finite distances to $\hat{d}((s, a), s')$, and set everything else to infinite. To prevent such issues, we choose to use state-action pairs as target goals, by adding a random action. Then, the embedding methods only need to embed state-action pairs.

  In planning when the target is actually a single goal $s' \in \mathcal{S}$, we use the following distance/Q-function

  $$\hat{d}((s, a), s') \triangleq -1 + \frac{1}{|\mathcal{A}|} \sum_{a' \in \mathcal{A}} \hat{d}((s, a), (s', a')). \tag{205}$$

  Such a modification is used for all embedding methods (PQEs, metric embeddings, asymmetrical dot products). For unconstrained networks, we test both the original formulation (of using single state as goals) and this modification.

**Environment.**    The environment is a grid-world with one-way doors, as shown in of Fig. 15, which is built upon `gym-minigrid` (Chevalier-Boisvert et al., 2018) (a project under Apache 2.0 License). The agent has 4 actions corresponding to moving towards 4 directions. When it moves toward a direction that is blocked by a wall or an one-way door, it does not move. States are represented as 18-dimensional vectors, containing the 2D location of the agent (normalized to be within $[-1, 1]^2$). The other dimensions are always constant in our enviroment as they refer to information that can not be changed in this particular environment (e.g., the state of the doors). The agent always starts at a random location in the center room (e.g., the initial position of the red triangle in Fig. 15). The environment also defines a goal sampling distribution as a random location in one of the rooms on the left or right side. Note that this goal distribution is only used for data collection and evaluation. In training, we train goal-conditional policies using the goal sampling mechanism adapted from Tian et al. (2020), as described above.

**Training trajectories.**    To collect the training trajectories, we use an $\epsilon$-greedy planner with groundtruth distance toward the environment goal, with a large $\epsilon = 0.6$. Each trajectory is capped to have at most 200 steps.

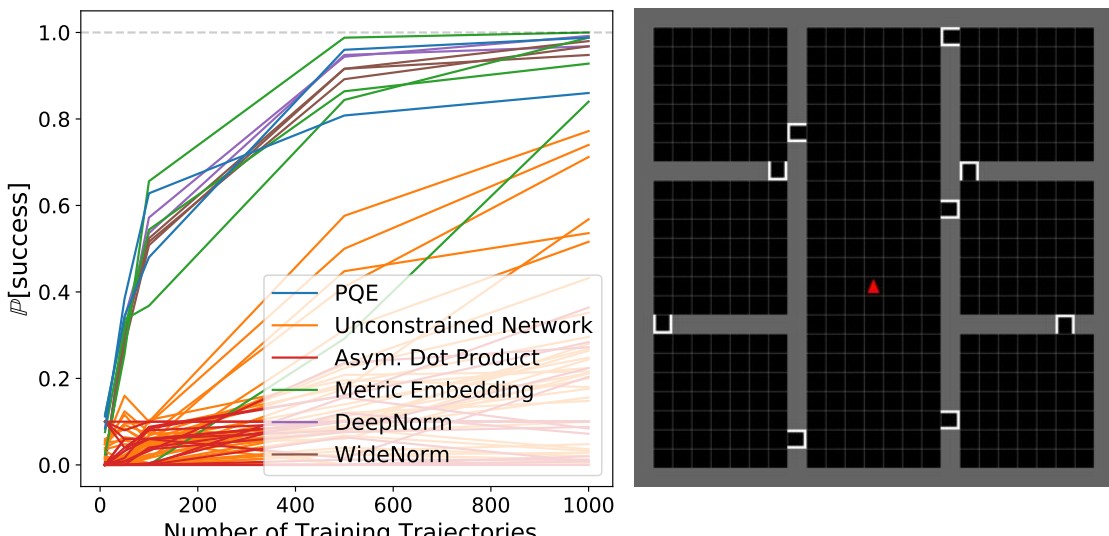

Figure 15: Grid-world offline Q-learning average planning success rates. Right shows the environment.

**Architecture.** All encoder based methods (PQEs, metric embeddings, dot products) use 18-2048-2048-2048-1024 network with ReLU activations and Batch Normalization (Ioffe & Szegedy, 2015) after each activation, mapping a 18-dimensional state to four 256-dimensional latent vectors, corresponding to the embeddings for all four state-action pairs. Unconstrained networks use a similar architecture and take in concatenated 36-dimensional inputs. With the original formulation with states as goals, we use a 36-2048-2048-2048-256-4 network to obtain a $\mathbb{R}^{|\mathcal{A}|}$ output, representing the distance/Q-values from each state-action pair to the goal; with the modified formulation with state-action pairs as goals, we use a 36-2048-2048-2048-256-16 network to obtain a $\mathbb{R}^{|\mathcal{A}| \times |\mathcal{A}|}$ output.

**Training.** We use 1024 batch size with the Adam optimizer (Kingma & Ba, 2014), with learning rate decaying according to the cosine schedule without restarting (Loshchilov & Hutter, 2016) starting from $10^{-4}$ to 0 over 1000 epochs. Since we are running Q-learning, all models are optimized w.r.t. MSE on the $\gamma$-discounted distances, with $\gamma = 0.95$. When running with the triangle inequality regularizer, $341 \approx 1024/3$ triplets are uniformly sampled at each iteration.

**Planning details.** To use the learned distance/Q-function for planning towards a given goal, we perform greedy 1-step planning, where we always select the best action in $\mathcal{A}$ according to the learned model, without any lookahead. In each of 50 runs, the planner is asked to reach a goal given by the environment within 300 steps. The set of 50 initial location and goal states is entirely decided by the seed used, regardless of the model. We run each method 5 times using the same set of 5 seeds.

**Full results.** Average results across 5 runs are shown in Fig. 15, with full results (with standard deviations) shown in Fig. 16. Planning performance across the formulations vary a lot, with PQEs and the Euclidean metric embedding being the best and most data-efficient ones. Using either formulation (states vs. state-action pairs as goals) does not seem to affect the performance of unconstrained networks. We note that the the asymmetrical dot product formulation outputting discounted distance is similar to Universal Value Function Approximators (UVFA) formulation (Schaul et al., 2015); the unconstrained network outputting discounted distance with states as goals is the same formulation as the method from Tian et al. (2020).

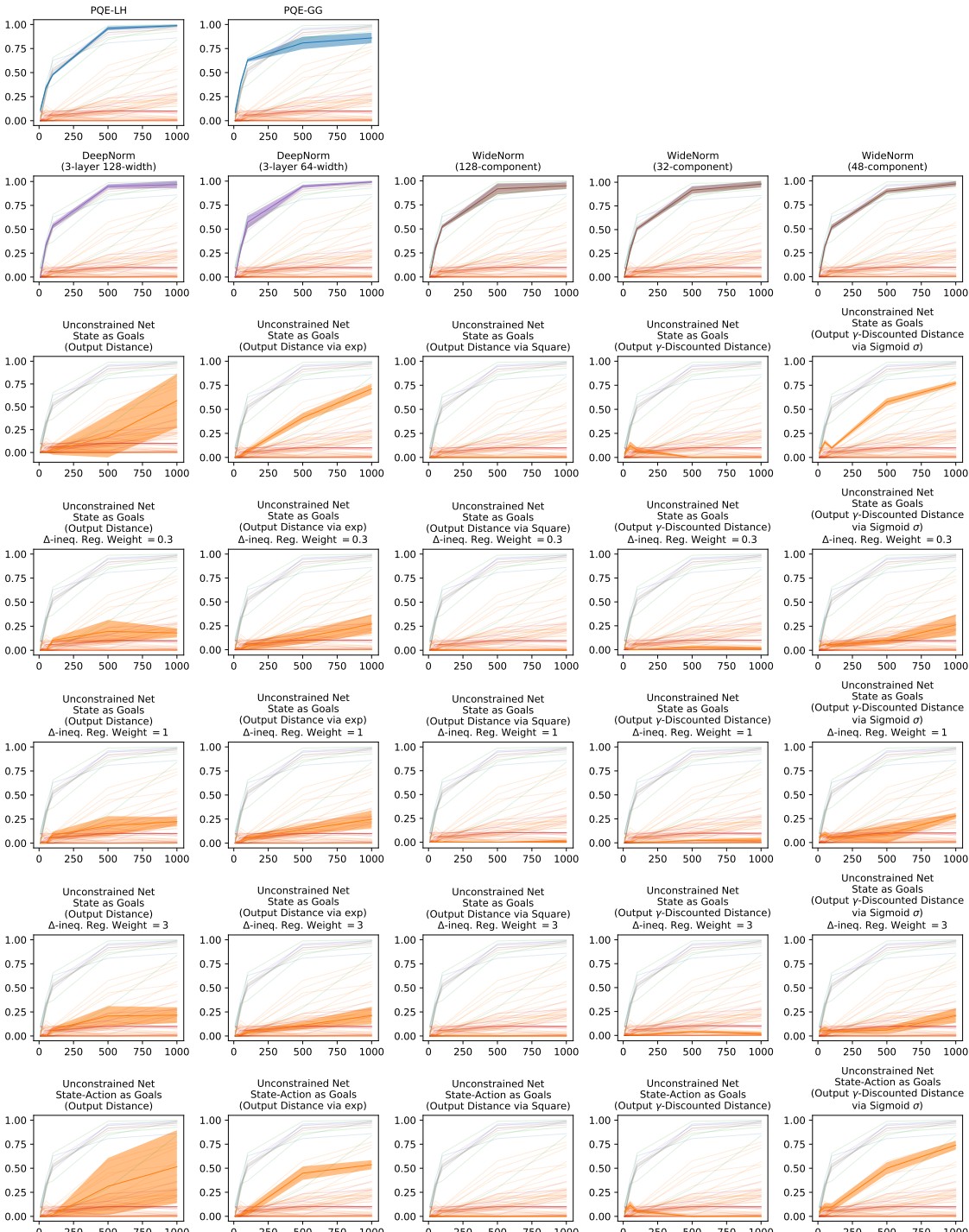

Figure 16: Grid-world offline Q-learning full results. Individual plots on show standard deviations.

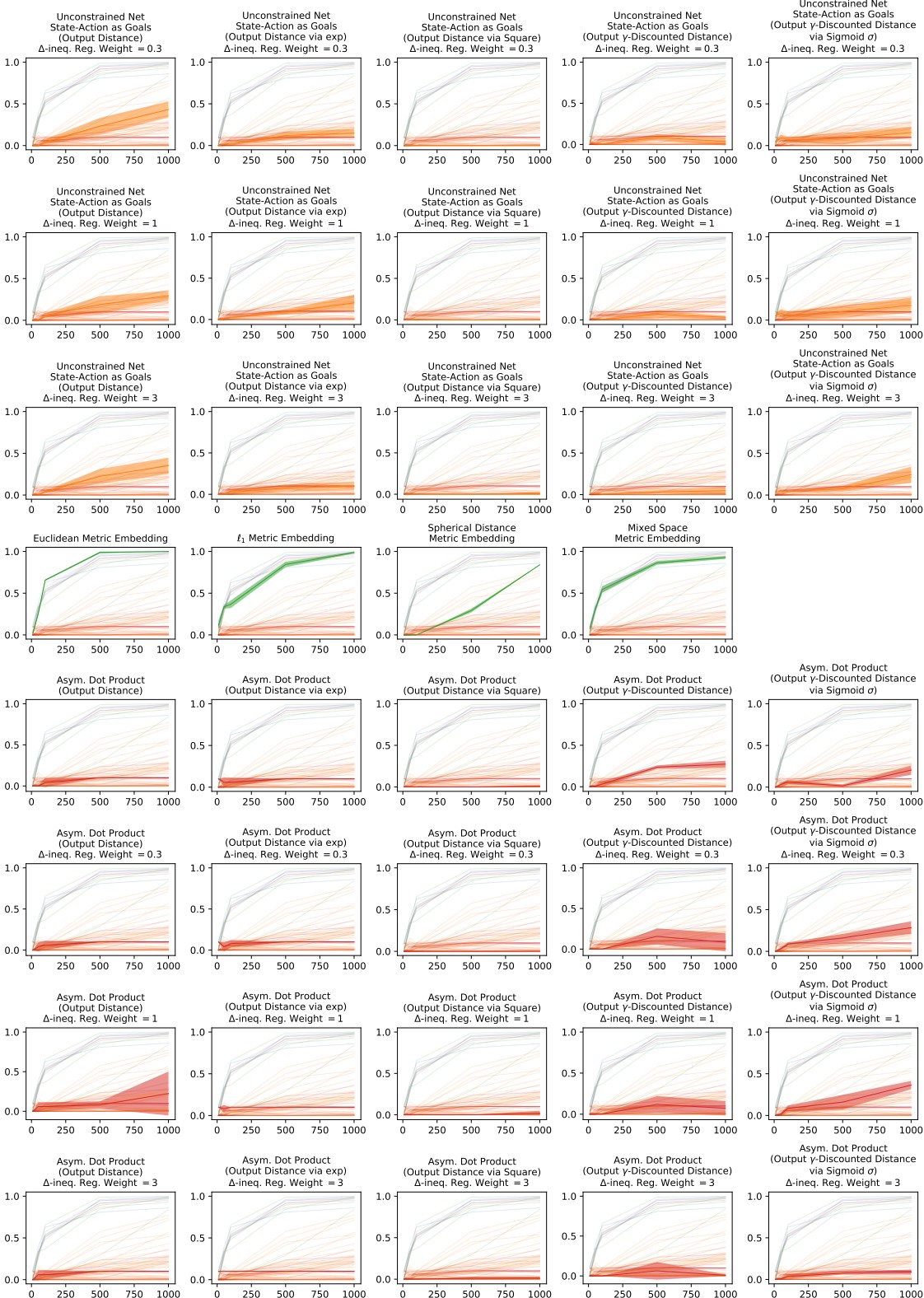

Figure 16: Grid-world offline Q-learning full results (cont.). Individual plots on show standard deviations.

