# OpenReview forum: "On the Learning and Learnability of Quasimetrics"
_ICLR.cc/2022/Conference — ICLR 2022 Poster_

### Official Review · Reviewer_WT3X · 2021-11-01

**Correctness:** 4
**Technical Novelty And Significance:** 3
**Empirical Novelty And Significance:** 3
**Recommendation:** 8
**Confidence:** 2

**Main Review:**

Strength:

- The paper is well written for most of time. The problem of learning quasimetrics is interesting, and the motivation is clear.

- The large scale experiments in Table 2 and Table 3 look promising. It is interesting to see that in directed networks, the proposed method works fairly well compared to the baselines.

Comments:

- It is mentioned in Section 3: "These objectives also indirectly capture generalization because a predictor with bad approximation or that ignores quasimetric constraints must have large error on some possible pairs." Could the authors elaborate on that?

- In particular, is there any relationship between quasimetric violation (Definition 4.2) and the generalization bound?

- If my understanding is correct, the regular (symmetric) metric is no more than a special case of the quasimetric. I wonder if there is any possible explanation for the undirected case (Table 3), in which the proposed method performs worse than some symmetric methods. Is this because of some overfitting on the training network?

- It seems that the acronym NTK is never defined.

- I did not check the proofs.

**Summary Of The Paper:**

The paper proposes a Poisson Quasimetric Embedding (PQE) framework, that can be used to embed data to a quasimetric space. Quasimetrics are similar to distance metrics, except that symmetry property does not need to hold. The proposed PQE approach can be solved using gradient based methods.

In addition, the paper shows that many common learning algorithms provably fail to learn a quasimetric consistent with training data.

**Summary Of The Review:**

Overall an interesting paper with solid theoretic and experimental results. I am not an expert in this area though, hence I cannot say how novel the approach is.

---

> ### Author Response · Authors · 2021-11-13
> **Reply to Reviewer WT3X**
>
> Thank you for your review!
>
> Please don’t hesitate to let us know if our reply below doesn’t fully address your comments.
>
> 1. **Quasimetric violation, distortion and generalization**
>
>    Generally, if the true function $f \in \mathcal{F}$ but the predicted $\hat{f}$ is far away from $\mathcal{F}$, it must also be far away from $f$. In our setting, $\mathcal{F}$ is the set of distance functions with good distortion and violation, containing the true quasimetric. So a predictor with bad distortion or violation should be far away from the truth.
>
>    Specifically to quasimetric learning, bad distortion means a large error on some training pair(s). If the predictor well approximates the training set but badly disrespects quasimetric constraints (large violation), it must be inaccurate on some heldout pair, because the true distance follows such constraints.
>
>    As a toy example, consider a quasimetric space with three elements $\{x,y,z\}$, with the training set containing $d(x,y)= d(y,z)=1$. Suppose the learned predictor $\hat{d}$ approximates these two values well, but violates triangle inequality with $\hat{d}(x,z) >> \hat{d}(x,y)+\hat{d}(y,z) \approx {d}(x,y)+{d}(y,z) $. Then it must predict far away from the truth $\hat{d}(x,z) >> {d}(x,y)+{d}(y,z) > d(x,z)$.
>
>    More concretely, for any quasimetric distance predictor, we have $\mathsf{dis} \geq \max(\mathsf{dis}_S, \sqrt{\mathsf{vio}})$, where
>       + $\mathsf{dis}$ is the distortion over the **entire** quasimetric, capturing **generalization**,
>       + $\mathsf{dis}_S$ is the distortion over training set, capturing **approximation**, and
>       + $\mathsf{vio}$ is the **violation of quasimetric constraints**.
>
>    We have added more discussion, and included this concrete formula in the revision as Theorem 4.6. Please see the orange text in Section 4.1, and proof of the result in Section B.1.
>
> 2. **Underperforming symmetric alternatives in undirected graph**
>
>    In this task, the true target function is symmetrical. We hypothesize that here the symmetric formulations have the correct structural prior and thus learn better. They search in a smaller function class that contains the true target (or at least good approximations to it). Even if all methods fit training data well, the symmetric formulations always find a symmetric solution. But the proposed method (PQEs) may find an asymmetric fit, and thus is less optimal.
>
>    In addition, while unconstrained networks also do very well on this task, they still fall short of the symmetric Euclidean metric embedding.
>
> 3. **Undefined NTK acronym**
>
>    We have revised the introduction to say “Neural Tangent Kernel (NTK)”.

---

### Official Review · Reviewer_RT4G · 2021-11-02

**Correctness:** 3
**Technical Novelty And Significance:** 3
**Empirical Novelty And Significance:** 3
**Recommendation:** 6
**Confidence:** 3

**Main Review:**

The problem of learning Quasi-metrics is an interesting problem with many applications. The proof of the fact that algorithms invariant to orthogonal transforms can not learn quasimetric functions is quite neat. The formulation of the PQEs is also quite nice. The paper is quite well written as well with toy examples which help pedgogically.

One complaint I have is that the empirical results are over just 5 trials.

**Summary Of The Paper:**

This paper considers the problem of learning Quasi-metrics. A quasi-metric is a non-negative function of two variables that satisfies the triangle inequality and that d(x,y) = 0 <=> x=y. Unlike metrics it does not satisfy triangle inequality.

In this paper, the authors define the quasi-metric learning problem. They then proceed to show that algorithms invariant to orthogonal transforms (in a sense they define - such that it includes unconstrained multi-layer perceptron) can not learn transforms satisfy the quasimetric constraints with probability at least 1/2 - o(1). They then define a program called Poisson Quasimetric Embeddings (PQEs) to learn quasimetrics - which come in two related flavours. They conclude with empirical evaluation of PQE and a variety of algorithms.

**Summary Of The Review:**

The paper examines an interesting problem. They propose an interesting program to solve the problem, with some analysis to show a class of algorithms can not be used to learn quasimetrics. It is also well written. Hence I recommend acceptance.

---

> ### Author Response · Authors · 2021-11-13
> **Reply to Reviewer RT4G**
>
> Thank you for your review!
>
> We conducted experiments on 780 random graphs settings, 92 large social graph settings, and 330 Q-learning settings. Each setting consists of a task, a dataset, and a method. We ran each setting for 5 trials, leading to a total of 6010 experiment runs. Such a large number of experiments is both extensive and already stretching our compute capabilities. Additionally, we observe that the relative comparison is often consistent across different tasks and/or datasets.
>
> If you have concerns or questions on specific settings, we are happy to hear your thoughts and provide more details. Please do not hesitate to let us know if we do not fully address your comments.

---

### Official Review · Reviewer_Lnh9 · 2021-11-02

**Correctness:** 3
**Technical Novelty And Significance:** 2
**Empirical Novelty And Significance:** 3
**Recommendation:** 5
**Confidence:** 3

**Main Review:**

Firstly, the topic under consideration is important and timely. Less structured spaces are frequently encountered, and learning algorithms tend to be developed with very restrictive settings in mind.

One highlight of the paper is Lemma 4.4, which rules out some common approaches to distance learning for this problem. The proof seems to me to be relatively straightforward.

My most significant reservation comes from Theorem 5.2, which is the theoretical justification behind the Poisson embedding: There we find a guarantee of distortion O(t log^2 n) for the embedding, and this is exceptionally high. Even the hopeless task of embedding metric space into Euclidean space gives smaller distortion O(logn) by Bourgain's theorem. The authors point to a result of Memoli, but the goal of that paper is quite different than the current one: That paper embeds into an ultrametric - a very restrictive space - and this allows them to  approximate a difficult computational task. So the distortion translates to approximation, which is more reasonable, but even then the dependence on the treewidth is problematic. The task of the current paper is to estimate a distance, and I find that the large distortion undermines this approach.

Some more minor concerns:
- I'm confused by the statement of Theorem 4.5: What's undesirable about S containing only few pairs? And why should a 50% probability be objectionable? Based on this bound alone, the random embedding could be repeated multiple times until finding one that is a good match for the sample.
- Why would use test your method against a metric embedding into Euclidean space? That embedding doesn't seem remotely appropriate for the problem under consideration.

**Summary Of The Paper:**

The paper is concerned with learning asymmetric distance functions (that is, quasimetrics) from a sample. The authors prove that a certain class of algorithms fails to properly learn these spaces, and instead propose the "Poisson quasimetric embedding" as a tool to learn these functions.

**Summary Of The Review:**

The paper has some nice positive contributions, but I have concerns about the overall theoretical foundation of the work.

---

> ### Author Response · Authors · 2021-11-13
> **Reply to Reviewer Lnh9**
>
> Thank you for your comments!
>
> Please let us know if our reply does not fully address your concerns.
>
> 1. **Distortion of Poisson Quasimetric Embeddings (PQEs)**
>
>    Our theoretical analyses (Theorem 4.5) show that many algorithms (including several common ones) attain unbounded distortion/violation. In comparison, $O(t \log^2 n)$ distortion (the best known quasimetric distortion to our knowledge) and optimal $1$ violation are already a huge improvement.
>
>    Our main contribution is the first step in theoretical analysis of quasimetric learning and the first formulation of a bounded-distortion **learnable** embedding. In the experiments, the proposed method achieves good approximations, which also suggest that the bound may be improved in a future work, e.g., by an average-case analysis (since distortion is a worst-case metric).
>
> 2. **Theorem 4.5 and small training set $S$**
>
>    Small $S$ is not undesirable, but does indeed represent an easier task so that it doesn’t expose the failure of these learning algorithms.
>
>    Recall that the goal is to find a quasimetric that fits training distances in $S$. A small training set means fewer requirements and thus is easy. For example, in the extreme case where $|S|=1$, an algorithm can simply always output a quasimetric where all points are equidistant from each other. Such an algorithm would perfectly fit the training distance and be equivariant to any transform (including orthogonal ones), but obviously fails for general $|S|$.
>
> 3. **Theorem 4.5 and 1/2 probability**
>
>    The 1/2 probability is over the random selection of the training set $S$ rather than on algorithm randomness. In other words, there are quasimetrics that are essentially **not learnable** by these algorithms, however many times they are repeated.
>
> 4. **Comparison against metric embedding**
>
>    Our investigation in quasimetric learning draws inspiration from metric learning. So such evaluations are designed to (1) show insufficiency of metric embeddings  and to (2) highlight the underlying asymmetry in the tasks.

---

> ### Author Response · Authors · 2021-11-29
> **Further clarification?**
>
> Thanks again for your review. We tried to address your comments in the previous message. Please do not hesitate to let us know if anything needs further clarification or if you still have any concern about any part of the paper.
>
> We look forward to hearing your thoughts.

---

### Official Review · Reviewer_kCMM · 2021-11-03

**Correctness:** 3
**Technical Novelty And Significance:** 2
**Empirical Novelty And Significance:** 2
**Recommendation:** 8
**Confidence:** 3

**Main Review:**

I am impressed by the long proof of the failure case construction in the appendix. But the statement of the Theorem 4.5 and the base case construction (Figure 3) is confusing. It seems that the orthogonal transformation is not essential in this result. Only thing that matters is that the action is at least nontrivial. Then we can identify different points y ~ y', w ~ w' and do the same trick.

Once we identify those points, the result becomes obvious: We have a quasimetric in one space, but we cannot define the pushforward of the quasimetric directly to a quotient space. It can fail to be a quasimetric. d(x, y) = d(y, w) = d(w, z) = 1 and d(w, z) = c for c large are just two contradictory statements. You can even get a simpler counterexample by considering d(x, y) = 1 and d(x, y') = c. (If the construction in Figure 3 is allowed, I don't see why this easier one is not allowed.) In other words, if we train an equivariant model on a space where there is actually no equivariance, the model will fail. The author can explain more clearly what Theorem 4.5 means and what are the intended implications.

**Summary Of The Paper:**

This paper studies the problem of learning of quasimetrics. It first proves that an orthogonal equivariant model cannot learn quasimetrics reliably, and then proposes the Poisson Quasimetric Embedding model which is both universal and differentiable. To be more precise, the Poisson process (or just a soft modification of Order Embedding) is differentiable and can be used to approximate quasipartitions, and quasipartitions can be used to approximate arbitrary quasimetrics.

The paper also conducts extensive experiments and indeed PQE has better performance for many tasks.

**Summary Of The Review:**

The paper is organized nicely and I have no difficulty following the proof. I think the paper provides a valuable contribution for the learning of quasimetrics and would be useful for many real world applications.

I consider this paper to be marginally above the acceptance threshold.

---

> ### Author Response · Authors · 2021-11-13
> **Reply to Reviewer kCMM**
>
> Thank you for your review and thoughts on our theoretical analysis.
>
> Our reply below is based on our understanding of your comments. Please do not hesitate to let us know if we misinterpret or do not adequately address your concerns.
>
> 1. **Theorem 4.5 can be generalized to algorithms equivariant to other transforms (beyond orthogonal ones)**
>
>    We agree with your comment. In essence, as long as the transform allows some kind of “input feature swapping”, a version of the theorem holds true. The paper specifies orthogonal transforms because it covers many common learning algorithms.  To avoid unnecessary complexity and potential confusion, we only state this version in the paper.
>
>    In the revision, we added a discussion paragraph on this generalization in the appendix, with reference to it in the main text. Please see the orange text in Section B.3.2.
>
> 2. **Simpler counter example**
>
>    We interpret your proposed example as a quasimetric space with elements $\{x, y, y’\}$ and corresponding quasimetric $d$ with $d(x, y) = 1, d(x, y’) = c >> 1$. If this interpretation is incorrect, please let us know!
>
>    This 3-element example would only work if the learning algorithm is **invariant** to the transform that swaps $(x,y) \leftrightarrow (x,y’)$ (so that it would predict **identically** on $(x,y)$ and $(x,y’)$). However, an equivariant algorithm will still solve the task because, on transformed input $(x,y) \leftrightarrow (x,y’)$, it outputs a **transformed** predictor that also swaps predictions on $(x,y)$ and $(x,y’)$.
>
>    The construction in Figure 3 is based on the observation that the triangle inequality is both an upper-bound and a lower-bound on a pair (by considering different triplets). E.g., in Figure 3, we use two triplets ($(x,y,z)$ and $(y,w,z)$) that share a pair ($(y,z)$). Therefore, on the same space, two different quasimetrics can give conflicting bounds on the same pair.  Suppose a transform can manipulate one quasimetric into another, **without changing that pair**. Then equivariance w.r.t. this transform implies identical prediction on that pair (when learned on either quasimetric), and a failure case is found.  Figure 3 is a manifestation of such an idea, with 4 nodes $(x,y,z,w)$ forming two triplets, and additional 2 nodes $(y’,w’)$ for “feature swapping” in equivariance. We are unable to simplify it further without either violating quasimetric constraints or removing the conflicting bounds.

---

> > ### Comment · Reviewer_kCMM · 2021-11-18
> > **Reply to the authors**
> >
> > Thanks for the explanation! My confusion has been cleared up. I agree with your second point. I will raise the score to 8.

---

### Author Response · Authors · 2021-11-13
**Revision**

We thank all reviewers for their detailed comments. Based on feedbacks, we have uploaded a new revision. Only a few changes are made, which we indicate with orange text. Here is an overview:

1. **Formal result that large distortion or violation implies bad generalization** (Reviewer WT3X)

   See Theorem 4.6 in Section 4.1, with proof in Section B.1.

   Note: Despite appearing before many other results, it is numbered after existing theorems to avoid confusion in discussion. If the paper is accepted, the numbering will be fixed in the camera-ready version.

2. **Discussion on generalizing the negative results to other transforms** (Reviewer kCMM)

    See paragraph in Section B.3.2.

3. **Fixed a few formatting issues**, including a missing space, a missing acronym definition.

We look forward to hearing more from reviewers!

---

### Decision · Program_Chairs · 2022-01-20

**Decision:**

Accept (Poster)

**Comment:**

The reviewers agree that the paper is addressing an interesting problem, and provides a valuable contribution for the learning of quasimetrics and would be useful for many real world applications.